# Protein abundance of AKT and ERK pathway components governs cell type-specific regulation of proliferation

Lorenz Adlung[1],[†] (ID), Sandip Kar[2,3,4,†], Marie-Christine Wagner[1,†], Bin She[1,†], Sajib Chakraborty[1], Jie Bao[5], Susen Lattermann[1], Melanie Boerries[5,6,7], Hauke Busch[5,6,7] (ID), Patrick Wuchter[8,9], Anthony D Ho[8], Jens Timmer[10], Marcel Schilling[1] (ID), Thomas Höfer[2,3,*] & Ursula Klingmüller[1,11,**] (ID)

## Abstract

Signaling through the AKT and ERK pathways controls cell proliferation. However, the integrated regulation of this multistep process, involving signal processing, cell growth and cell cycle progression, is poorly understood. Here, we study different hematopoietic cell types, in which AKT and ERK signaling is triggered by erythropoietin (Epo). Although these cell types share the molecular network topology for pro-proliferative Epo signaling, they exhibit distinct proliferative responses. Iterating quantitative experiments and mathematical modeling, we identify two molecular sources for cell type-specific proliferation. First, cell type-specific protein abundance patterns cause differential signal flow along the AKT and ERK pathways. Second, downstream regulators of both pathways have differential effects on proliferation, suggesting that protein synthesis is rate-limiting for faster cycling cells while slower cell cycles are controlled at the G1-S progression. The integrated mathematical model of Epo-driven proliferation explains cell type-specific effects of targeted AKT and ERK inhibitors and faithfully predicts, based on the protein abundance, anti-proliferative effects of inhibitors in primary human erythroid progenitor cells. Our findings suggest that the effectiveness of targeted cancer therapy might become predictable from protein abundance.

**Keywords** 32D-EpoR; BaF3-EpoR; CFU-E; MAPK; PI3K
**Subject Categories** Cell Cycle; Quantitative Biology & Dynamical Systems; Signal Transduction

**Mol Syst Biol. (2017) 13: 904**

## Introduction

Eukaryotic cells use a limited number of signal transduction pathways to integrate information from extracellular stimuli and to regulate cellular decisions such as proliferation, differentiation, and apoptosis. In particular, the PI3K/AKT and Ras/MEK/ERK pathways have been implicated in the control of cell growth and proliferation in many different cell types (Saez-Rodriguez *et al*, 2015). The key components of these pathways are highly conserved, which suggests that they form generic proliferation-control modules that function in a specialized way in different cell types. Indeed, previous studies suggest that the regulation of the AKT and the ERK pathway crucially depends on the cellular context (McCubrey *et al*, 2011).

Both pathways are activated by multiple growth factors and cytokines such as the hormone erythropoietin (Epo), which is the prime regulator of erythropoiesis. Epo is essential for survival, proliferation, and differentiation of erythroid progenitor cells and thereby facilitates continuous renewal of mature erythrocytes (Koury & Bondurant, 1990). The cognate Epo receptor (EpoR) is present on the cell surface of erythroid progenitor cells as a preformed homodimer (Livnah *et al*, 1999). At the stage of

1   Division of Systems Biology of Signal Transduction, German Cancer Research Center (DKFZ), Heidelberg, Germany
2   Division of Theoretical Systems Biology, German Cancer Research Center (DKFZ), Heidelberg, Germany
3   BioQuant Center, University of Heidelberg, Heidelberg, Germany
4   Department of Chemistry, Indian Institute of Technology, Mumbai, India
5   Systems Biology of the Cellular Microenvironment Group, IMMZ, ALU, Freiburg, Germany
6   German Cancer Consortium (DKTK), Freiburg, Germany
7   German Cancer Research Center (DKFZ), Heidelberg, Germany
8   Department of Medicine V, University of Heidelberg, Heidelberg, Germany
9   Institute for Transfusion Medicine and Immunology, University of Heidelberg, Mannheim, Germany
10  Center for Biological Signaling Studies (BIOSS), Institute of Physics, University of Freiburg, Freiburg, Germany
11  Translational Lung Research Center (TLRC), Member of the German Center for Lung Research (DZL), Heidelberg, Germany
    *Corresponding author. Tel: +49 6221 54 51380; Fax: +49 6221 54 51487; E-mail: t.hoefer@dkfz.de
    **Corresponding author. Tel: +49 6221 42 4481; Fax: +49 6221 42 4488; E-mail: u.klingmueller@dkfz.de
    †These authors contributed equally to this work

colony-forming unit-erythroid (CFU-E), primary cells exhibit highest EpoR levels and are thus most responsive to Epo (Wu *et al*, 1995).

Epo-induced signaling comprises besides the activation of STAT5 (Klingmüller *et al*, 1996) the induction of PI3K and MAPK signal transduction (Miura *et al*, 1994; Haseyama *et al*, 1999). The activation of MAPK pathways has been associated with erythroblast enucleation and stress-induced erythropoiesis (Tamura *et al*, 2000; Schultze *et al*, 2012) while PI3K signaling has been shown to control maturation of erythroid progenitor cells by promoting cell survival and proliferation (Myklebust *et al*, 2002; Bouscary *et al*, 2003). The open question remains how these pathways are integrated to control proliferation.

PI3K/AKT is connected to the initiation of translation by the mammalian target of rapamycin (mTOR) in proliferating erythroid progenitor cells (Grech *et al*, 2008). Phosphorylation of the ribosomal protein S6, a downstream target of mTOR, is a crucial step for protein synthesis, and thus cell growth. If cells are treated with the mTOR inhibitor rapamycin, they are considerably smaller than untreated cells (Fingar *et al*, 2002, 2004). Lately, an mTOR-independent S6 activation mechanism through ERK and RSK (Roux *et al*, 2007) was found. S6 activation, and thus protein synthesis, is required for cell growth and proliferation of reticulocytes (Knight *et al*, 2014). Factor-induced proliferation is a complex process that can be divided into two steps (Smith & Martin, 1973). First, cells integrate growth factor signals and grow in G1 phase of the cell cycle if enough nutrients are available (Pardee, 1974). Second, if a critical mass is reached and the restriction point is crossed, cells progress through the cell cycle, synthesize DNA, and finally undergo cytokinesis to double their number (Jones & Kazlauskas, 2000).

It has been shown that Epo regulates proliferation of erythroid progenitor cells and modulates cell cycle regulators (Dai *et al*, 2000; Bouscary *et al*, 2003; Sivertsen *et al*, 2006). Epo stimulation of erythroblasts results in a rapid upregulation of Cyclin-D2 as well as Nupr1, Gstpt1, Egr1, Nab2 and a downregulation of Cyclin-G2 and p27 (Fang *et al*, 2007). The regulation of cell cycle progression is rather complex, as it is regulated by multiple feedforward and feedback loops (Ferrell, 2013), and for example, the negative regulators Cyclin-G2 and p27 do not necessarily act in a coordinated manner (Le *et al*, 2007). Mathematical models of the cell cycle in mammalian cells have been developed that describe the change of cyclins and CDKs with time (Yao *et al*, 2008; Alfieri *et al*, 2009); however, only lately mechanistic and dynamic links from signaling to cell cycle progression were established (Mueller *et al*, 2015).

The ample knowledge on molecular mechanisms contributing to the regulation of erythropoiesis has been facilitated, on the one hand, by the availability of factor-dependent, immortalized hematopoietic cell lines from mice. For example, the interleukin (IL) 3-dependent cell lines BaF3 of lymphoid origin (Palacios & Steinmetz, 1985) and 32D of myeloid origin (Greenberger *et al*, 1983) have been utilized for decades to unravel structure–function relationship of cytokine receptors such as the EpoR (Wang *et al*, 1993; Klingmüller *et al*, 1996). Exogenous expression of the EpoR renders these cell lines responsive to Epo and enables proliferation in the presence of Epo (D'Andrea *et al*, 1989). Due to their growth properties, BaF3 cells are currently widely used in kinase drug discovery and represent a reliable cellular system to access kinase activity (Jiang *et al*, 2005; Moraga *et al*, 2015). On the other hand, primary erythroid progenitor cells from mice (mCFU-E) are readily available

from fetal liver or bone marrow, and methods for their cultivation have been established (Rich & Kubanek, 1976; Landschulz *et al*, 1989). For the human system, a protocol has been devised (Broudy *et al*, 1991; Miharada *et al*, 2006) to expand and differentiate human erythroid progenitor (hCFU-E) cells from CD34$^+$ cells mobilized into the peripheral blood of healthy donors. With this strategy, sufficient material of hCFU-E can be obtained to confirm in functional studies the clinical relevance of observations.

Here, we present a mathematical model that links Epo-induced activation of AKT, ERK, and S6 to cell cycle progression and proliferation in the context of murine erythroid progenitor cells and murine hematopoietic cell lines exogenously expressing the EpoR. We uncover that the cell type-specific protein abundance is sufficient to explain alterations in the dynamics of the signaling pathways. Further, we demonstrate how mathematical modeling can establish a mechanistic connection from signaling to cell growth, cell cycle progression, and proliferation upon Epo stimulation and inhibitor treatment. We show that in murine erythroid progenitor cells, proliferation is primarily controlled by the regulation of cell growth, whereas regulation of cell cycle progression is the major determinant of proliferation of the murine hematopoietic cell lines as well as in human erythroid progenitor cells.

# Results

## Cell type-specific regulation of proliferation and signaling by erythropoietin

To quantitatively assess Epo-induced proliferative responses in murine primary erythroid progenitor cells at the colony-forming unit-erythroid stage (mCFU-E) and in the immortalized murine cell line BaF3 exogenously expressing the EpoR (BaF3-EpoR), we incubated the cells in the presence of different Epo doses and measured as a readout for proliferation DNA synthesis by thymidine incorporation. mCFU-E cells showed a higher sensitivity toward Epo with an EC$_{50}$ of $0.26 \pm 0.02$ U/ml Epo as compared to $0.55 \pm 0.04$ U/ml Epo for BaF3-EpoR cells (Fig 1A). At saturating doses of 50 U/ml Epo, mCFU-E cells doubled their number within 13.1 h and BaF3-EpoR cells within 18.7 h (Fig 1B). We determined the size of unstimulated cells by imaging flow cytometry and observed average diameters of mCFU-E cells and BaF3-EpoR cells of 11 and 13.8 μm, respectively (Fig 1C). mCFU-E cells were smaller and more heterogeneous in size. The smaller average size of mCFU-E cells correlated with a higher sensitivity toward Epo and a shorter doubling time compared with BaF3-EpoR cells.

To investigate the Epo-dependent regulation of cell growth and proliferation, we examined the Epo-induced activation of AKT and ERK pathways in mCFU-E and BaF3-EpoR cells by immunoblotting. In the first step, the protein abundance and dynamics of phosphorylation of EpoR, AKT, and ERK in response to Epo stimulation were qualitatively assessed. To study the expression and phosphorylation of the EpoR and AKT, mCFU-E cells were stimulated with 2.5 U/ml Epo, while BaF3-EpoR cells were stimulated with 5 U/ml Epo to account for the at least twofold difference in sensitivities toward Epo (Fig 1A) and the observation that responses in mCFU-E cells already saturated at 2.5 U/ml Epo, whereas BaF3-EpoR required more than 5 U/ml Epo (Appendix Fig S1). As shown in Fig 1D, top

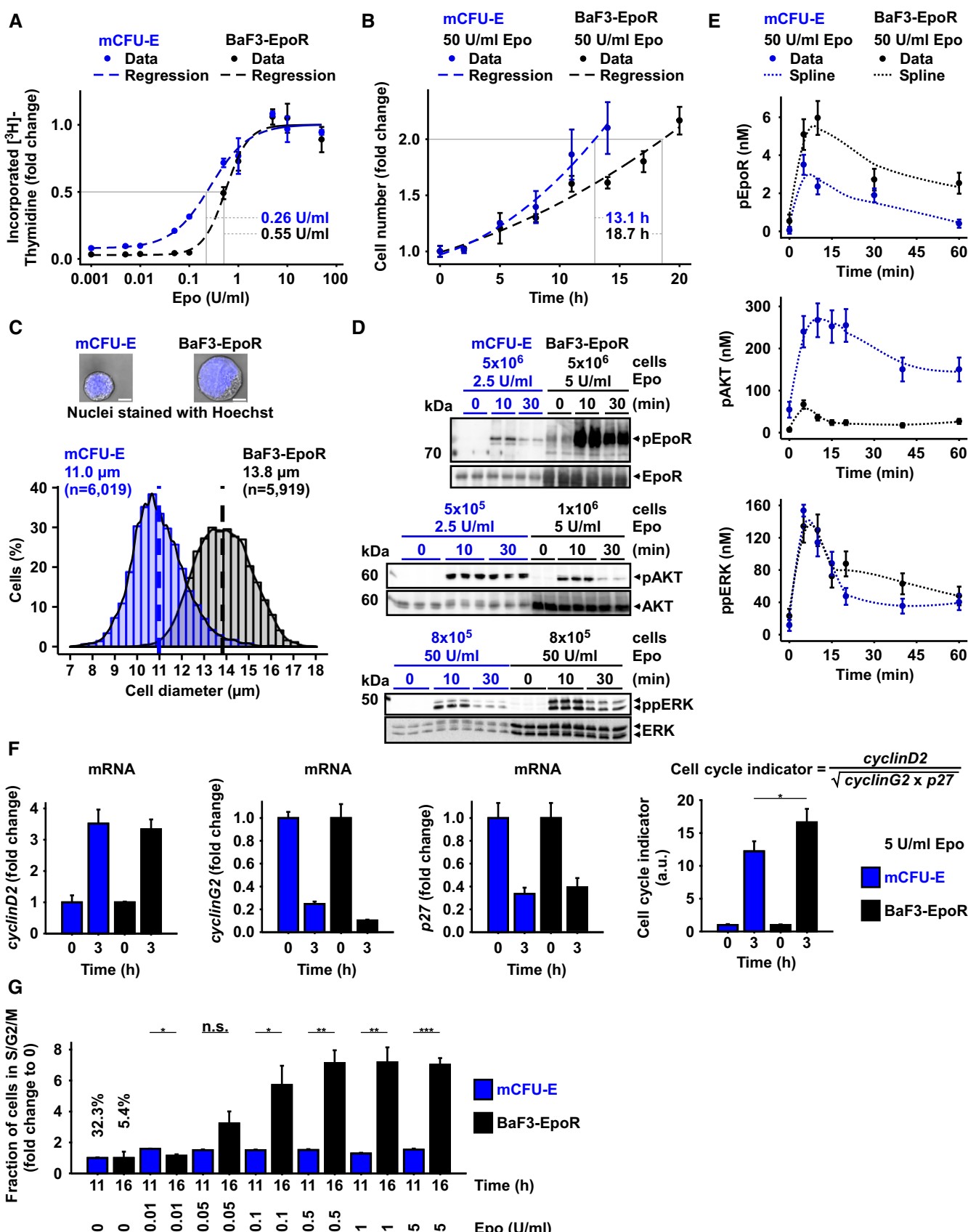

**Figure 1.**

**Figure 1.  Characterization of Epo-induced proliferation and signaling.**

A   DNA content of mCFU-E and BaF3-EpoR cells in response to different Epo concentrations. [³H]-thymidine incorporation was measured after 14 h (mCFU-E) or 38 h (BaF3-EpoR). Data represented as mean $\pm$ standard deviation, $N = 3$. Lines represent sigmoidal regression. $EC_{50}$ values are given.

B   Cell doubling with time in response to 50 U/ml Epo. Cell numbers were determined by manual counting using trypan blue exclusion assay. Data represented as mean $\pm$ standard deviation, $N = 3$. Lines represent exponential regression. Doubling time is indicated.

C   Size determinations of mCFU-E and BaF3-EpoR cells. Exemplary fluorescence microscopy pictures upon Hoechst staining for nucleus visualization with 60× objective. The bar represents 10 µm distance (upper panel). Cell diameter was measured by imaging flow cytometry. Cytoplasm was stained with calcein, and nuclei were stained with DRAQ5. Probability density function of size distribution with indicated mean diameter of mCFU-E and BaF3-EpoR cells. All cells were growth factor-deprived and unstimulated.

D   Epo-induced phosphorylation of EpoR, AKT, and ERK of mCFU-E and BaF3-EpoR cells. Above each panel, the number of growth factor-deprived cells examined per time point is indicated as well as the concentration of Epo applied for stimulation. To the left, the position of the molecular weight marker is indicated in kDa and arrowheads indicate the position of the protein of interest. For the detection of the EpoR, $5 \times 10^6$ mCFU-E cells and $5 \times 10^6$ BaF3-EpoR cells were stimulated with 5 U/ml Epo. Cells were lysed, subjected to immunoprecipitation with anti-EpoR, and were analyzed by immunoblotting using either anti-pTyr (pEpoR) or anti-EpoR (EpoR) antibodies. For the detection of AKT, mCFU-E cells were stimulated with 2.5 U/ml Epo and BaF3-EpoR cells were stimulated with 5 U/ml Epo. Per time point, cellular lysates equivalent to $5 \times 10^5$ mCFU-E cells and $1 \times 10^6$ BaF3-EpoR cells were analyzed by immunoblotting using anti-pAKT and anti-AKT antibodies. For the detection of ERK, cells were stimulated with 50 U/ml Epo. Per time point, cellular lysates equivalent to $8 \times 10^5$ cells were analyzed by immunoblotting. Immunoblot detection was performed with chemiluminescence utilizing a CCD camera device (ImageQuant).

E   Absolute concentrations of pEpoR, pAKT, and ppERK in mCFU-E and BaF3-EpoR cells with time in response to 50 U/ml Epo. Experimental data of a representative experiment are depicted with filled circles, and dashed lines represent splines. Error bars indicate standard deviation estimated by an error model. $N = 1$.

F   Expression of cell cycle indicator genes of mCFU-E and BaF3-EpoR cells in response to 5 U/ml Epo. Genes were selected based on microarray analysis. Experimental data are shown as fold change to unstimulated cells with mean $\pm$ standard deviation, $N = 3$. Welch modified two-sample *t*-test, *$P < 0.05$.

G   Fold change of fractions of cells in S/G2/M phase of the cell cycle with respect to 0 U/ml Epo. Growth factor-deprived cells were stimulated with indicated Epo doses for given time. Fractions in sub-G1, G1, S, G2/M were determined by propidium iodide (PI) staining for DNA content. Data represented as mean $\pm$ standard deviation, $N = 3$. Welch modified two-sample *t*-test, n.s. = not significant, *$P < 0.05$, **$P < 0.01$, ***$P < 0.005$. The percentage of cells in S/G2/M phase in unstimulated cells is additionally indicated.

Source data are available online for this figure.

panel, the total expression level of the EpoR and the extent of EpoR phosphorylation were higher in BaF3-EpoR cells compared with mCFU-E cells. With regard to AKT, we consistently observed much lower levels of Epo-induced AKT phosphorylation in BaF3-EpoR cells. Therefore, to obtain reproducible results for Epo-induced phosphorylation of AKT in both cell types, we examined per time point $1 \times 10^6$ BaF3-EpoR cells and $5 \times 10^5$ mCFU-E cells. These studies showed that although the apparent abundance of total AKT was higher in BaF3-EpoR cells, the Epo-induced phosphorylation of AKT was higher and more sustained in mCFU-E cells compared with BaF3-EpoR cells (Fig 1D, middle panel). As an indicator for AKT activation, we focused on the analysis of Ser473 phosphorylation that is predictive for full kinase activation (Alessi *et al*, 1996; Scheid *et al*, 2002; Sarbassov *et al*, 2005) since we observed that it correlates with Thr308 phosphorylation in an Epo dose-dependent manner (Appendix Fig S2). We previously noted that cytokine receptors activated the MAP kinase pathway to a much lower extent compared with receptor tyrosine kinases (Iwamoto *et al*, 2016) and showed that 50 U/ml Epo is required to achieve an ERK phosphorylation degree of at least 10% in mCFU-E cells (Schilling *et al*, 2009). Therefore, we stimulated mCFU-E cells and BaF3-EpoR cells with 50 U/ml Epo to reliably examine ERK phosphorylation. As depicted in Fig 1D, bottom panel, higher ERK protein levels as well as elevated levels of ERK phosphorylation were observed in BaF3-EpoR cells, but overall both cell types exhibited a comparable transient ERK phosphorylation dynamics.

To quantitatively assess the cell type-specific differences, we determined the specific concentrations of key signaling molecules. We assumed spherical geometry of the cells and calculated the cytoplasmic volumes and cell surface areas by confocal microscopy (Table 1). For each protein of interest, calibrator proteins of known concentration were used to determine the absolute number of molecules per cell for mCFU-E, BaF3-EpoR, and 32D-EpoR cells, a cell line that was used for validation experiments (Table 1). BaF3-EpoR cells exhibited a ten times higher density of EpoR molecules on their cell surface (26 molecules/µm²) than mCFU-E cells (2.6 molecules/µm²). The accurate quantification of total molecules per cell and the correction for the difference in cellular volume showed that indeed the concentration of total ERK was higher in BaF3-EpoR (2964 $\pm$ 166 nM) than in mCFU-E cells (1140 $\pm$ 64 nM), whereas the concentration of total AKT was comparable (510 $\pm$ 62 nM in BaF3-EpoR cells and 407 $\pm$ 16 nM in mCFU-E cells).

To quantitatively examine the dynamics of Epo-induced signal transduction in mCFU-E and BaF3-EpoR cells, we used randomized sample loading in combination with quantitative immunoblotting to determine in a time-resolved manner the phosphorylation of EpoR, AKT, and ERK in both cell types. In our previous studies (Becker *et al*, 2010), at maximum 75% of receptor dimers on the cell surface were bound to Epo. Therefore, we assumed the EpoR phosphorylation degree does not exceed 75% (Fig 1E). We experimentally determined the phosphorylated fraction of AKT by quantitative protein arrays that combine in-spot normalization and binding model-based calibration; 54% of AKT was phosphorylated in mCFU-E cells upon stimulation with 2.5 U/ml Epo for 10 min (Appendix Fig S3). For ppERK, we previously showed by quantitative mass spectrometry that at maximum 10% of ERK1/2 is double-phosphorylated in mCFU-E cells (Schilling *et al*, 2009). These numbers were used to derive the nanomolar concentrations of pEpoR, pAKT, and ppERK in mCFU-E cells. The concentrations of the respective abundance in BaF3-EpoR cells were scaled accordingly as mCFU-E and BaF3-EpoR cells were always analyzed on the same blot. In both cell types, the dynamics of the concentration of pEpoR was transient albeit with higher peak amplitude and steady state level in BaF3-EpoR cells (Fig 1E, top panel), which reflects their much larger total density of EpoR. Despite the higher EpoR activation in BaF3-EpoR cells, pAKT concentrations were higher in mCFU-E cells (Fig 1E, middle panel).

**Table 1.   Cytoplasmic concentration of pathway components of mCFU-E, BaF3-EpoR, and 32D-EpoR cells.**

|  | mCFU-E | BaF3-EpoR | 32D-EpoR |
|---|---|---|---|
| Cytoplasm ($\mu m^3$) | 399 | 1,400 | 1,406 |
| Cell surface area ($\mu m^2$) | 378.5 | 600.3 | 607 |
| EpoR surface (molecules/$\mu m^2$) | 2.6 | 26.1 | 22.7 |
| EpoR (nM) | 4.16 | 18.62 | 16.76 |
| PI3K/AKT |  |  |  |
| AKT (nM) | 407 ± 16.3 | 510 ± 62.1 | 607.7 |
| PI3K (nM) | 12.7 ± 2.5 | 12.4 ± 0.8 | 14.3 |
| SHIP1 (nM) | 15.4 ± 2.5 | 84.2 ± 10.4 | 127.8 |
| PTEN (nM) | 10.4 ± 0.5 | 107.4 ± 7.0 | 96.8 |
| PDK1 (nM) | 545 ± 80 | 763 ± 175 | 1,554.5 |
| Gab1 (nM) | 20.8 ± 2.3 | – | – |
| Gab2 (nM) | – | 30 | 1.1 |
| Ras/ERK |  |  |  |
| Ras (nM) | 3,530 ± 249 | 9,531 ± 790 | 7,855.4 |
| Raf (nM) | 1,340 ± 298 | 3,886 ± 864 | 7,807.3 |
| MEK (nM) | 1,460 | 4,380 | 4,743.9 |
| ERK (nM) | 1,140 ± 64 | 2,964 ± 166 | 2,326.1 |
| S6 (nM) | 5,340 ± 694 | 2,590 ± 270 | 4,531.8 |
| Ratio to mCFU-E: |  |  |  |
| mTOR | 1 | 6.18 | 1.8 |
| Rictor | 1 | 4.57 | 0.19 |
| Raptor | 1 | 3.08 | 2.3 |
| RSK | 1 | 7.4 | 2.8 |

Cytoplasmic volumes were estimated using imaging flow cytometry excluding the nucleus. 1,000 EpoR molecules on the surface of mCFU-E cells were reported (D'Andrea and Zon, 1990). 15,500 EpoR molecules on the surface of BaF3-EpoR cells were determined by a saturation-binding assay with [$^{125}$I]-labeled Epo (Becker *et al*, 2010). Number of EpoR molecules on the surface of 32D-EpoR cells was calculated in comparison to BaF3-EpoR cells using flow cytometry. All other concentrations were determined as fold differences to BaF3-EpoR expression level. "–" indicates the absence of protein. For details, see Appendix G.

This is not simply an effect of AKT expression, which was comparable in both cell types (Table 1). Moreover, the maximum concentration of ppERK was similar, despite the higher abundance of ERK in BaF3-EpoR cells (Table 1). The dynamics of ppERK in BaF3-EpoR and mCFU-E cells was slightly different at time points beyond 15 min of Epo stimulation (Fig 1E, bottom panel). These data indicate that EpoR activation is translated into cell type-specific patterns of activation of the AKT and ERK pathways.

To provide a link between signal transduction and cell cycle progression, transcriptome analysis was performed for up to 18.5 h after stimulation of BaF3-EpoR cells with 1 U/ml Epo (Appendix Fig S4A) and for up to 24 h after stimulation of mCFU-E cells with 0.5 U/ml Epo (Appendix Fig S4B). These analyses revealed that in both cell types several cell cycle regulator genes were differentially expressed upon Epo stimulation (Appendix Fig S4). Prominent among these cell cycle regulators affected by Epo were the activator Cyclin-D2, and the repressors Cyclin-G2 and p27, all of which jointly

control the progression from G1 phase to S phase—the key event for cell cycle entry (Fang *et al*, 2007). On the other hand, other genes involved in the regulation of the cell cycle such as *cyclinE1* (CCNE1) and *cyclinE2* (CCNE2) showed only little regulation in either cell types. To confirm the transcriptomics studies, we examined the selected Epo-responsive cell cycle-regulating genes by quantitative RT–PCR analysis (Fig 1F) and showed that after 3 h of stimulation with 5 U/ml Epo, a saturating Epo dose for proliferation in BaF3-EpoR and mCFU-E cells (Appendix Fig S1), mRNA induction of *cyclinD2 (CCND2),* and mRNA repression of *cyclinG2 (CCNG2)* and *p27 (CDKN1B)* exhibited comparable fold changes in BaF3-EpoR and mCFU-E cells. These results suggested that the quantification of the expression of *cyclinD2*, *cyclinG2*, and *p27* might provide an early quantitative measure to compare Epo-induced cell cycle progression in BaF3-EpoR and mCFU-E cells. To summarize the contribution of the cell cycle activator and the two cell cycle repressors that counteract each other in controlling cell cycle progression, we defined a cell cycle indicator as follows:

$$\frac{[cyclinD2]}{\sqrt{[cyclinG2] \times [p27]}}$$

As evidenced in Fig 1F, after 3 h of Epo addition we observed in BaF3-EpoR and mCFU-E comparatively small changes in the expression of the individual components (e.g. only *cyclinG2*) but a strong increase in the cell cycle indicator, 16-fold for BaF3-EpoR cells and 13-fold for CFU-E cells, respectively. These results underscore that at this early time point the coefficient reflects the complex regulation of cell cycle progression in response to Epo stimulation better than any of its components alone.

Notably, the cell cycle indicator was significantly ($P = 0.04$) higher in BaF3-EpoR cells compared with CFU-E cells (Fig 1F, right panel). In line with this observation, we observed by propidium iodide staining after stimulation with 5 U/ml Epo for 16 h (BaF3-EpoR) or 11 h (mCFU-E) that the fold change of cells in the S/G2/M phase of the cell cycle in response to Epo stimulation was also significantly ($P = 0.002$) higher in BaF3-EpoR cells compared with mCFU-E cells (Fig 1G). This result supports our notion of the cell cycle indicator as an early measure for cell cycle progression and shows that, whereas mCFU-E cells are already committed to cell cycle progression, an increasing fraction of BaF3-EpoR cells enters S/G2/M phase in response to stimulation with increasing Epo doses. The dynamics of EpoR, AKT, and ERK phosphorylation were distinct between the two cell types, which also differed in their Epo sensitivity of the proliferative response and their proliferation rate.

### Influence of cellular protein abundance on Epo-induced signaling dynamics

To understand how the cell type-specific signaling dynamics of AKT and ERK arise, we applied quantitative dynamical pathway modeling. Our mathematical model consists of coupled ordinary differential equations assuming mass action kinetics, and a Hill-type phenomenological term at the receptor level. As depicted in Fig 2A, the model describes preformed dimers of the EpoR to be phosphorylated (pEpoR) upon Epo stimulation. Dephosphorylation of the pEpoR is catalyzed by the pEpoR-activated phosphatase SHP1 and a constitutively active phosphatase. In the model, pEpoR forms

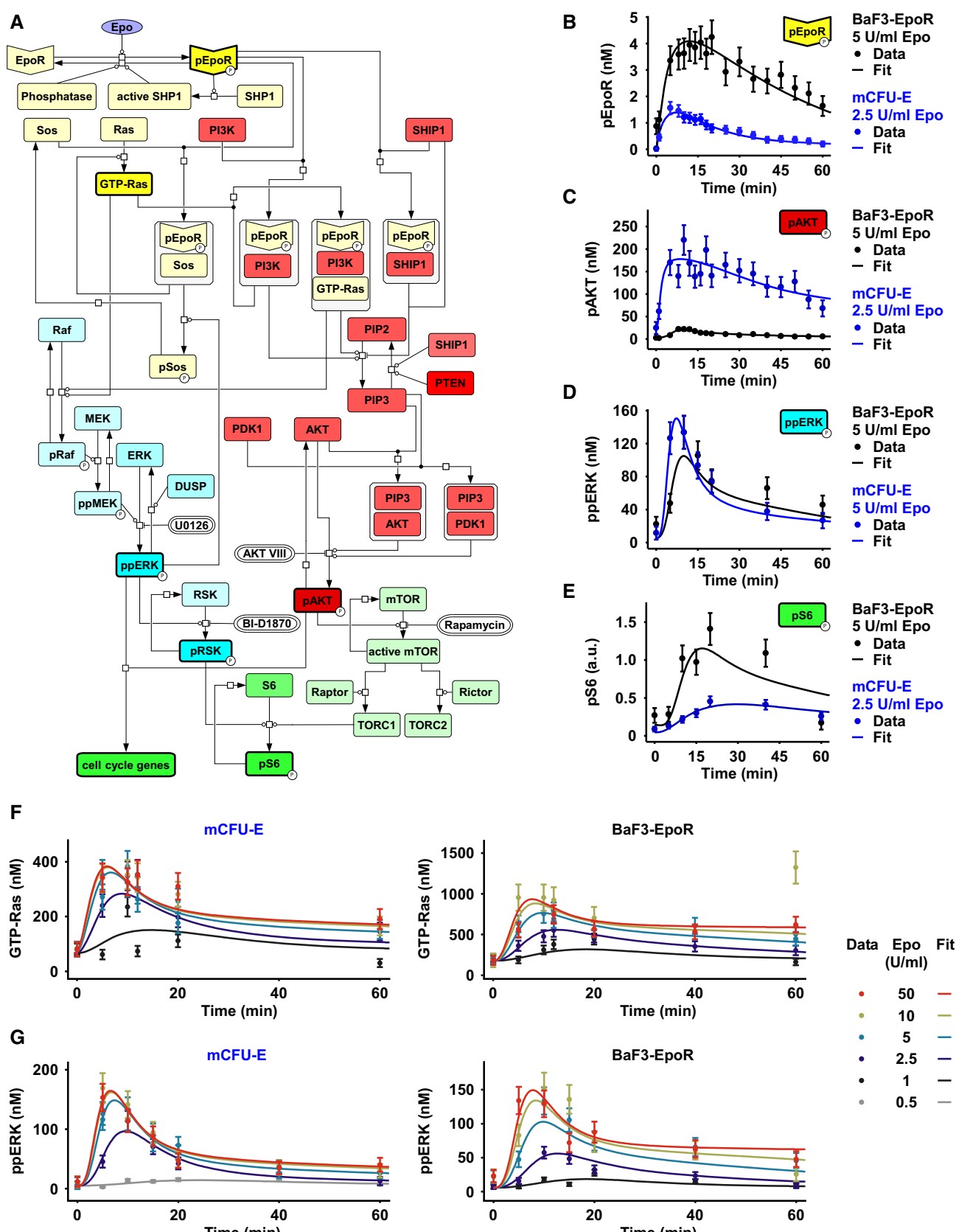

**Figure 2.**

◄

**Figure 2.  Mathematical modeling of the Epo-induced AKT, ERK, and S6 activation.**

A     The model is represented as a process diagram and reactions are modulated by enzyme catalysis (circle-headed lines) or inhibition (bar-headed lines). Prefix "p" represents phosphorylated species. Bold framed species were experimentally measured. White capsules represent inhibitors. Binding of the ligand Epo to its cognate receptor results in the phosphorylation of EpoR. Receptor and associated complexes are depicted in yellow, the AKT pathway is red, the ERK pathway is blue, and cell cycle genes and the S6 network are shown in green.

B–G   Model calibration with time-resolved quantitative immunoblot data of mCFU-E cells in blue and BaF3-EpoR cells in black. Growth factor-deprived mCFU-E cells ($5 \times 10^6$ cells per condition) and BaF3-EpoR cells ($1 \times 10^7$ cells per condition) were stimulated with different Epo doses, and absolute concentrations were determined for pEpoR (B), pAKT (C), ppERK (D). The scale for pS6 (E) was estimated in arbitrary units. GTP-Ras (F) and ppERK were determined upon stimulation with indicated, color-coded Epo doses. pEpoR was analyzed by immunoprecipitation followed by immunoblotting, GTP-Ras was analyzed after pulldown using a fusion protein harboring GST fused to the Ras binding domain of Raf-1 followed by detection by quantitative immunoblotting. For pAkt and ppERK, cellular lysates were subjected to quantitative immunoblotting. Calibrator proteins were used for EpoR, AKT, GTP-Ras, and ERK to facilitate the conversion to nM concentrations. Experimental data are represented by filled circles. Error bars represent standard deviation estimated by an error model. Solid lines represent model trajectories. $N = 1$.

Source data are available online for this figure.

complexes with Sos, PI3K, GTP-Ras, and SHIP1. PI3K generates PIP3 that recruits AKT and PDK1 to the plasma membrane, triggering phosphorylation of AKT. SHIP1 and PTEN dephosphorylate PIP3 and thus inhibit the activation of AKT. The pEpoR-Sos complex induces the formation of GTP-Ras that in turn activates the Raf/MEK/ERK cascade. ppERK is dephosphorylated by dual-specific phosphatases (DUSP). Both pAKT and ppERK regulate the expression of *cyclinD2*, *cyclinG2*, and *p27*. Phosphorylated S6 is linked to the regulation of cell growth and proliferation (Meyuhas & Dreazen, 2009). Phosphorylated AKT catalyzes the activation of mTOR that forms the complexes TORC1 and TORC2 with binding partners Raptor and Rictor, respectively. TORC1 facilitates S6 phosphorylation. ppERK can also trigger the phosphorylation of S6 via phosphorylation of RSK. Thus, pAKT and ppERK signals are integrated in the phosphorylation of S6 through mTOR and RSK (Appendix Fig S5) as well as in the expression of cell cycle regulators. To perturb the system, we used inhibitors acting at different processes (AKTVIII, phosphorylation of AKT; U0126, MEK activity; BID1870, phosphorylation of RSK; rapamycin, mTOR activity). By a systematic model reduction (Appendix F.2), we tested the binding rates of the adaptor proteins Gab1/2 (Sun *et al*, 2008) to the EpoR. We identified that the adapter proteins Gab1/2 may bind either very fast or slow and therefore play a negligible role in the fast equilibrium of receptor–adaptor complex formation. Additionally, we decomposed the enzymatic rate constants (e.g. for phosphatases and kinases) into the product of total enzyme concentration and a biochemical rate constant (also called catalytic efficiency, or turnover, $k_{cat}$). This decomposition enabled us to quantify the biochemical rate constant as a property of the enzyme, which therefore can be assumed to be independent of a given cell type, whereas the enzyme concentration is cell type-specific (Appendix F). For further details on the coupled ordinary differential equations, the dynamic variables, the parameter estimation as well as their annotation (Appendix Table S1), and their sensitivities toward inhibitors, see Appendix F. The full SBML model is available at FAIRDOMHub (https://fairdomhub.org/).

In the mathematical model, cell type-specific and global parameters were distinguished. The total concentrations of all proteins in the signaling network are specific for the particular cell type (Table 1; Appendix Fig S7). By contrast, the rate constants for reversible protein binding and enzymatic catalysis are global parameters independent of cell type. In principle, however, these kinetic parameters might still be affected by further regulatory proteins that have not been included in the model. To test this, the question was whether the different expression levels of the pathway components

we measured for mCFU-E and BaF3-EpoR cells could explain the observed cell type-specific signal processing through the ERK and AKT pathways. To this end, we used the measured protein concentrations as an input and otherwise assumed identical kinetic parameters for both cell types. The mathematical model was fitted first for receptor activation and deactivation to account for the different EpoR and JAK2 levels in the two cell types (Appendix Fig S8; Becker *et al*, 2010; Bachmann *et al*, 2011). In total, 432 data points (the raw data can be viewed as Source Data of the Figures and the Appendix) of Epo-induced pathway activation, measuring pEpoR, pAKT, GTP-Ras, ppERK, and pS6, were used to estimate the 82 global kinetic parameters of the model. The experimental conditions comprised different Epo doses and perturbation by inhibitor treatment or overexpression of negative regulators (Appendix Figs S9–11). We found that the distinct signaling dynamics and dose responses to Epo were captured by the mathematical model (Fig 2B–G; Appendix Fig S13).

In summary, our mathematical analysis indicates that differences in signal processing can be explained by different abundance in signaling proteins in mCFU-E and BaF3-EpoR cells, based on a mathematical model with global kinetic parameters.

## Experimental validation of model predictions for negative regulators

Having established the mathematical model for the activation of the AKT and ERK pathways in CFU-E and BaF3-EpoR cells, the negative regulators of signaling came into focus.

First, the lipid phosphatases SHIP1 and PTEN were overexpressed, and the impact on AKT activation was monitored (Fig 3A). In mCFU-E cells, a strong effect of PTEN overexpression on Epo-induced AKT phosphorylation was experimentally observed, and a weaker effect of a similar overexpression of SHIP1, which were both captured by the model (Fig 3A, Appendix Fig S13). Further, we observed that the Epo-induced induction pAKT in wild-type BaF3-EpoR cells was even lower than in mCFU-E cells with overexpressed PTEN (Fig 3A), which is consistent with the high concentrations of SHIP1 and PTEN in BaF3-EpoR cells (Table 1). The mathematical model calibrated based on these data nevertheless predicted that overexpression of SHIP1 or PTEN would decrease AKT phosphorylation even further in these cells. Indeed in an independent experiment, the Epo-induced dynamics of pAKT in BaF3-EpoR cells overexpressing SHIP1 or PTEN was in agreement with the model trajectories (Fig 3B). Further, we predicted with the model and

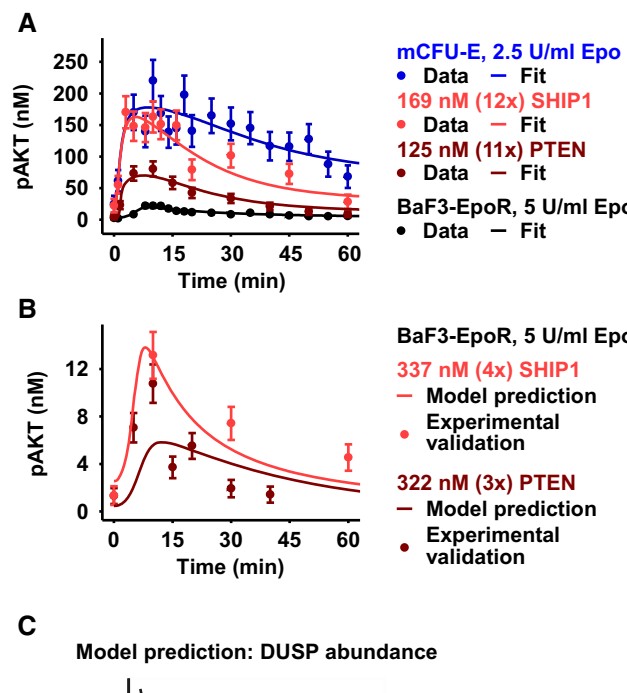

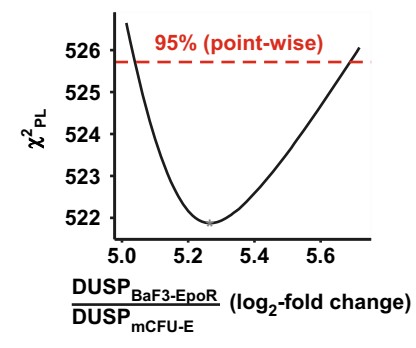

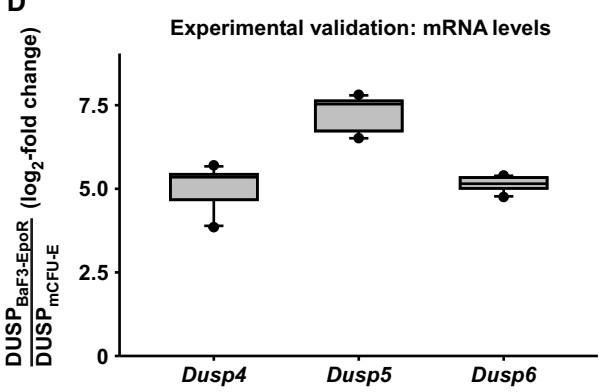

Figure 3.   Model predictions and experimental validations for negative regulators of AKT and ERK signaling.

A   Model calibration with mCFU-E wild-type cells, and mCFU-E cells overexpressing SHIP1 or PTEN, and BaF3-EpoR wild-type cells. Experimental data are represented by filled circles. Error bars represent standard deviation estimated by an error model. Solid lines represent model trajectories. N = 1.

B   Model prediction of pAKT dynamics in BaF3-EpoR cells overexpressing PTEN or SHIP1. Model predictions are represented by solid lines. Experimental validation data obtained by quantitative immunoblotting are represented by filled circles. Error bars represent standard deviation estimated by an error model. A representative experiment is shown. N = 1.

C   Model prediction for the basal expression level of the dual-specific phosphatase (DUSP). DUSP abundance ratio between BaF3-EpoR and mCFU-E cells was identified by the mathematical model. The solid line indicates the profile likelihood. The dashed red line indicates the threshold to assess point-wise 95% confidence interval. The asterisk indicates the optimal parameter value.

D   Experimental validation of basal DUSP expression in mCFU-E and BaF3-EpoR. Quantitative RT–PCR was performed with all samples on the same plate, allowing a direct comparison of mRNA levels between mCFU-E and BaF3-EpoR cells. Data are normalized to the *Rpl32* gene. Ratios of expression in BaF3-EpoR cells compared with mCFU-E cells are shown as box plots. Boxes indicate the interquartile range and whiskers extend to 1.5 × interquartile range. N = 10. For details, see Appendix K.

Source data are available online for this figure.

mass spectrometry analysis of unstimulated mCFU-E, hCFU-E, and BaF3-EpoR cells, Epo-regulated DUSP family members were below the detection limit. Therefore, we used the mRNA expression levels as proxy, assuming at least some correlation with protein expression. The mathematical model predicted a $\log_2$-fold change of 5.27 higher basal expression of DUSP in BaF3-EpoR cells compared with mCFU-E cells (Fig 3C). To experimentally validate this model prediction, we first identified by microarray analysis of mCFU-E cells (Bachmann *et al*, 2011) and BaF3-EpoR cells (Appendix Fig S14) DUSP4, DUSP5, and DUSP6 as family members that are differentially expressed in response to Epo stimulation. The analysis of the basal mRNA expression of these DUSP by quantitative RT–PCR showed that the $\log_2$-fold difference in the basal expression of DUSP4, DUSP5, and DUSP6 in BaF3-EpoR cells compared with the expression in mCFU-E cells (Fig 3D) was in agreement with the prediction by the mathematical model.

In summary, the expression levels of negative regulators of the AKT and ERK pathways are critical for cell type-specific Epo signal processing.

### Cell type-specific information flow through ERK and AKT pathways

To test our observation that the abundance of signal transduction proteins is a key determinant of the dynamics of cell type-specific signal processing, we examined another Epo-responsive hematopoietic cell line, 32D cells, which are derived from the myeloid branch, that exogenously express the EpoR, 32D-EpoR. We determined the abundance of pathway components in 32D-EpoR cells by quantitative immunoblotting (Table 1) and utilized these concentrations in our mathematical model as cell type-specific parameters. Without altering the previously determined global kinetic parameters, we simulated the putative response of pAKT, ppERK, and pS6 at 50 U/ml Epo in 32D-EpoR cells and observed good agreement with

validated experimentally that simultaneous downregulation of SHIP1 and PTEN to their respective concentrations in mCFU-E cells enhanced Epo-induced pAKT levels in BaF3-EpoR cells to the extent observed in mCFU-E cells (Appendix Fig S13).

Second, the DUSPs, a family of phosphatases that negatively regulate ERK signaling, were examined. The analysis of DUSP protein abundance is challenging because multiple isoforms with different functions exist and only very few antibodies, mostly with low specificity, are available. In our proteome-wide quantitative

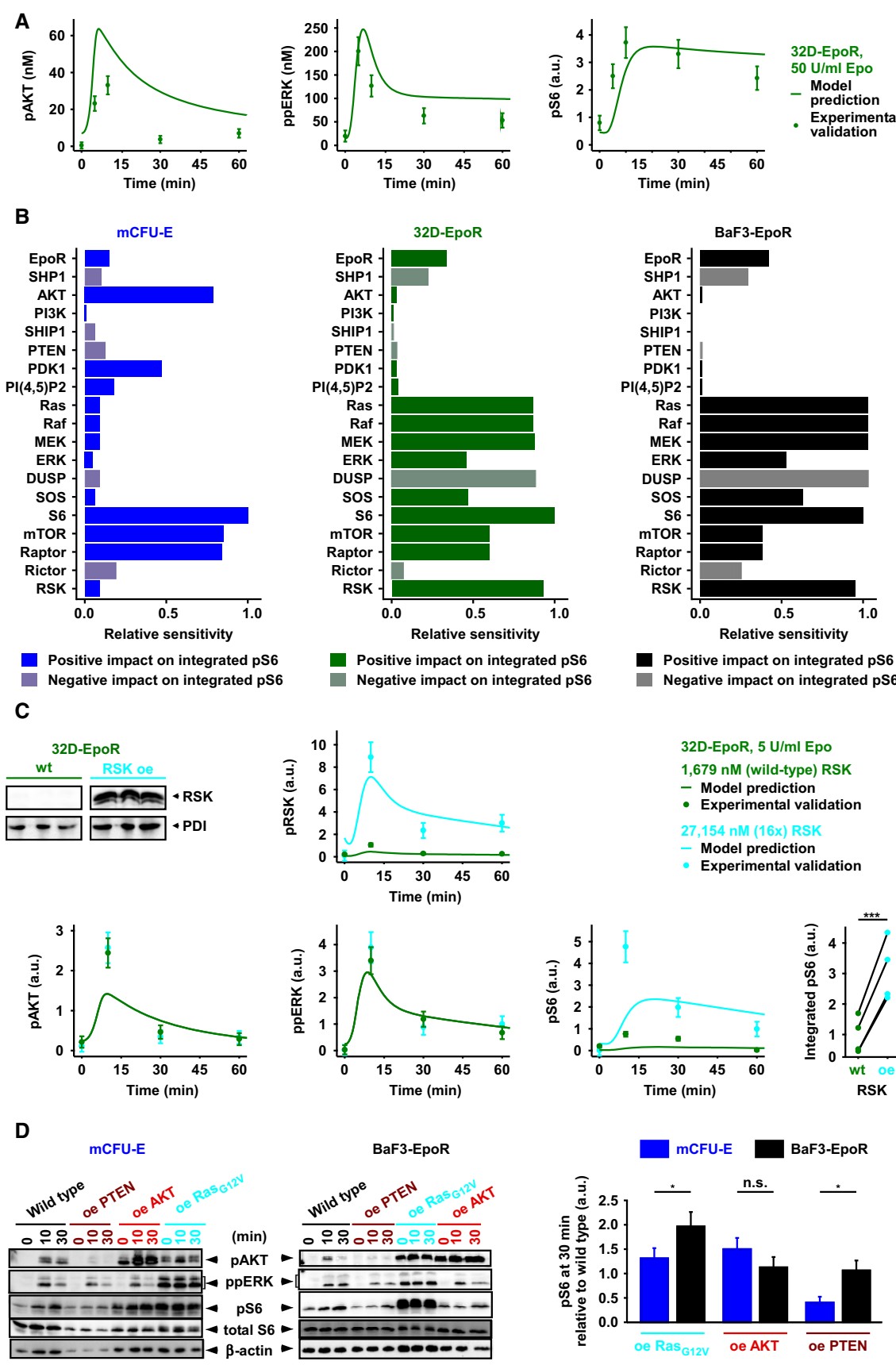

**Figure 4.**

**Figure 4.  Validation of Epo-induced signaling dynamics in 32D-EpoR cells under RSK wild-type conditions and upon overexpression of RSK.**

A    Model prediction and experimental validation for pAKT, ppERK, and pS6 dynamics of 32D-EpoR cells in response to 50 U/ml Epo. Simulations are based on measured, cell type-specific protein abundance of 32D-EpoR cells and global kinetic rates estimated from mCFU-E and BaF3-EpoR cells. Model predictions are represented by solid lines. Experimental validation data obtained by quantitative immunoblotting are represented by filled circles. Error bars represent standard deviation estimated by an error model, $N = 1$.

B    Sensitivity analysis for integrated pS6 in mCFU-E, 32D-EpoR, and BaF3-EpoR cells. Measured, cell type-specific protein abundance and estimated global kinetic rates were taken into account. Proteins were grouped according to the network modules. Integrated pS6 exhibited high sensitivity toward RSK in 32D-EpoR and BaF3-EpoR cells but not in mCFU-E cells.

C    Model prediction and experimental validation for RSK, AKT, ERK, and S6 activation upon RSK overexpression (oe) and 5 U/ml Epo stimulation in 32D-EpoR cells. Simulations are based on measured, cell type-specific protein abundance of 32D-EpoR cells and global kinetic rates estimated from mCFU-E and BaF3-EpoR cells. Model predictions are represented by solid lines. Experimental validation data obtained by quantitative immunoblotting are represented by filled circles. Error bars represent standard deviation estimated by an error model. Integrated pS6 was significantly higher upon RSK overexpression as compared to 32D-EpoR wild-type (wt) cells, $N = 4$ (lower right panel). Two-sample *t*-test, \*\*\*$P < 0.005$.

D    Impact of AKT, Ras, or PTEN overexpression on Epo-induced pS6 dynamics in mCFU-E and BaF3-EpoR cells. Quantitative immunoblotting upon overexpression of PTEN, AKT, a constitutively active Ras protein, or the empty vector control in mCFU-E cells or BaF3-EpoR cells. Growth factor-deprived mCFU-E cells ($5 \times 10^6$ cells per condition) and BaF3-EpoR cells ($1 \times 10^7$ cells per condition) were stimulated with 5 U/ml Epo for indicated time points. Cellular lysates were analyzed by immunoblotting employing sequential reprobing anti-pAKT, anti-ppERK, anti-pS6, anti-S6, and to ensure equal loading with anti-β-actin antibodies. Detection was performed with chemiluminescence using a CCD camera device (ImageQuant). Quantification of pS6 on the right is depicted as fold change to wild-type samples at 30 min after Epo stimulation. Error bars represent standard deviation. oe: overexpression. $N = 3$. Welch modified two-sample *t*-test, n.s. = not significant, \*$P < 0.05$.

Source data are available online for this figure.

the experimental data for ppERK and pS6 (Fig 4A). For pAKT, the peak time and signal duration were correctly predicted, while the model overestimated the peak amplitude and steady state of pAKT. Further, given the similarities in the protein abundance of 32D-EpoR and BaF3-EpoR cells, we assumed similarities in the dynamics of pathway activation. However, model simulations in line with experimental data used for model calibration under those conditions (Fig 2F and G; Appendix Fig S12) indicated that differences in the peak amplitude, signal duration, and steady state existed between these two cell types (Appendix L). Further we showed that the goodness of fit of a mathematical model calibrated with data from mCFU-E and BaF3-Epor cells is superior to predict the dynamic activation of AKT, ERK, and S6 in response to 50 U/ml Epo stimulation in 32D-EpoR cells when adapted to the cell type-specific protein abundance compared with mathematical models calibrated with data obtained from mCFU-E cells or BaF3-EpoR cells alone (Appendix L).

To systematically characterize the signal flow through the underlying molecular network (see Fig 2A) in mCFU-E, BaF3-EpoR, and 32D-EpoR cells and to identify possible similarities or differences, the relative sensitivities (response coefficients) of the network output were computed with respect to the expression levels of these components. As output, the integrated response of pS6 was used as the most downstream molecular component that integrates the signals coming from the AKT and ERK pathway (Fig 4B). This analysis showed a remarkable difference between mCFU-E cells and BaF3-EpoR cells. The integrated response of pS6 in mCFU-E cells was primarily sensitive to the AKT pathway (Fig 4B, left column), whereas in BaF3-EpoR the influence of the Ras/MEK/ERK cascade dominated (Fig 4B, right column). The 32D-EpoR cells were similar to BaF3-EpoR cells but with a slightly higher impact of AKT pathway components AKT, PI3K, SHIP1, PTEN, PDK1, and PI(4,5)P2 on integrated pS6 (Fig 4B, middle column).

The high sensitivities in the specific cell types were overall associated with high abundance of the signaling proteins and pathway activities (Table 1; Figs 1E and 2B–D). For example, the AKT pathway was most active in mCFU-E cells and exhibited highest sensitivities there (Fig 4B, third row), while the Ras/MEK/ERK pathway

was most active in BaF3-EpoR cells having highest sensitivities there (Fig 4B). This result might seem counterintuitive, as high sensitivity is typically associated with network components that occur at low, limiting concentration. However, the distributions of sensitivities can be rationalized based on the fact that S6 is an integration node of signals from the Ras/MEK/ERK and AKT pathways. The more active a pathway is in a given cell type (e.g. AKT in CFU-E cells; Ras/MEK/ERK in BaF3-EpoR cells), the more it controls S6 phosphorylation and, hence, changes in such a pathway will have greater effects (higher sensitivities) than changes in quantitatively less important pathways if the system is not saturated. Therefore, the distribution of sensitivities for the integrated response of pS6 provides a quantitative measure for the differential signal flow along the Ras/MEK/ERK and AKT pathways in the different cell types.

The sensitivity analysis indicated, for example, that the RSK abundance (Fig 4B, bottom line) exhibits a high impact on the integrated pS6 response in BaF3-EpoR and 32D-EpoR cells but not in mCFU-E cells. These model-based insights are consistent with the high sensitivities obtained for the Ras/MEK/ERK pathway in the former two cell types, as RSK is downstream of ERK. Although wild-type 32D-EpoR cells already exhibited 2.8-fold higher levels of RSK than mCFU-E cells (Table 1), the sensitivity analysis taking this protein abundance into account suggested that RSK overexpression in 32D-EpoR cells would result in an increase in integrated pS6 in response to Epo stimulation. To test this counterintuitive model prediction, RSK was overexpressed in 32D-EpoR cells. Utilizing the amount of RSK experimentally detected in wild-type 32-EpoR cells as well as the amount of RSK present in the cells overexpressing RSK, the mathematical model predicted a major increase in pS6 in response to Epo stimulation, whereas pAKT and ppERK remain rather unaffected. In line with this model prediction, experimental overexpression of RSK had no effect on the Epo-induced dynamics of the upstream components pAKT and ppERK in 32D-EpoR cells but strongly increased the Epo-stimulated phosphorylation level of S6 (Fig 4C). The mathematical model correctly predicted the effect of RSK overexpression in 32D-EpoR cells on the Epo-induced dynamics of ppERK and pRSK, but the peak amplitude of pAKT and pS6 were underestimated. In four independent experiments, the

**Figure 5.  Evaluation of the effects of AKT inhibitor AKT VIII and MEK inhibitor U0126 on signaling and cell cycle.**

A   Epo-induced signaling upon AKT or MEK inhibitor treatment. Growth factor-deprived cells were pretreated for half an hour with AKT VIII or U0126, respectively, and subsequently stimulated with 5 U/ml Epo. PDI served as loading control.

B   Model calibration with integrated pAKT, ppERK, and pS6 data upon inhibitor treatment and 5 U/ml Epo stimulation of mCFU-E, 32D-EpoR, and BaF3-EpoR cells. Area under curve from time-resolved, quantitative immunoblotting data was calculated and is represented by filled circles. *N* = 3. Error bars represent standard deviation estimated by an error model. Solid lines represent model trajectories.

C   Analysis of cell cycle indicator genes upon AKT VIII and U0126 treatment. Growth factor-deprived cells were pretreated for half an hour with indicated doses of a single inhibitor, followed by stimulation with 5 U/ml Epo for 0 and 3 h. The expression of *cyclinD2*, *cyclinG2*, and *p27* was measured by quantitative RT–PCR and normalized to the *Rpl32* gene. Genes were selected based on microarray analysis. Experimental data are shown as fold change to unstimulated cells with mean ± standard deviation, *N* = 3. Welch modified two-sample *t*-test, n.s. not significant, *$P < 0.05$, **$P < 0.01$, ***$P < 0.005$.

Source data are available online for this figure.

integrated pS6 response was increased (Fig 4C, bottom right panel), validating the prediction of high RSK sensitivity of pS6 in this cell type. In further agreement with the sensitivity analysis, the observed experimental overexpression of constitutively active Ras resulted in comparison with wild-type cells in a significantly ($P = 0.04$) stronger elevation of Epo-induced S6 phosphorylation in BaF3-EpoR than in mCFU-E cells, whereas the overexpression of PTEN significantly ($P = 0.01$) diminished Epo-stimulated S6 phosphorylation more strongly in mCFU-E than in BaF3-EpoR cells (Fig 4D).

Taken together, our results show that the abundance of the network components directs the signal flow differentially through the AKT and Ras/MEK/ERK pathways. In mCFU-E cells, signaling to S6 occurs primarily through the AKT axis and in BaF3-EpoR cells primarily through the ERK pathway. In 32D-EpoR cells signaling to S6 is similar to BaF3-EpoR cells, but with slightly higher sensitivity toward AKT.

### Effects of AKT and ERK inhibition on Epo signaling depend on cell type

Having established how the protein abundance of the network components controls the information flow through the AKT/ERK/S6 network from the EpoR, the question was how this network controls Epo-induced proliferation in a cell type-specific manner. To this end, inhibitors of specific network nodes in the AKT and ERK pathways were employed: AKT VIII, an inhibitor of AKT phosphorylation (Lindsley *et al*, 2005), and U0126, an inhibitor of ERK phosphorylation (Favata *et al*, 1998). The dynamics of AKT, ERK, and S6 phosphorylation upon inhibitor treatment and stimulation with 5 U/ml Epo were monitored in all three cell types by quantitative immunoblotting. Specifically, the levels of pAKT, ppERK, and pS6 were determined in the absence or presence of 0.05, 0.5 and 5 μM of each inhibitor after 0, 10, 30, and 60 min of Epo stimulation.

In mCFU-E cells, AKT VIII reduced the pAKT amplitude. In 32D-EpoR and BaF3-EpoR cells, higher AKT VIII doses reduced the steady state level of pAKT, which therefore became more transient (Fig 5A, upper panel). The cell type-specific impact of inhibitors on signaling dynamics was even more pronounced for U0126 treatment. The duration of the ppERK signal decreased with higher U0126 doses in mCFU-E cells, whereas the signal was overall reduced in BaF3-EpoR cells and to lesser extent also in 32D-EpoR cells (Fig 5A, lower panel). Surprisingly, none of the three cellular systems studied here exhibited a significant cross talk between the AKT and the ERK axes. Under wild-type conditions, AKT VIII reduced pAKT but not ppERK, and U0126 decreases ppERK but not

pAKT. S6 phosphorylation was primarily influenced by AKT VIII in mCFU-E cells, whereas U0126 more strongly diminished the pS6 response in BaF3-EpoR. 32D-EpoR cells occupied an intermediate position, showing both AKT VIII and U0126 effects on pAKT and ppERK, respectively. This might, however, depend on protein abundance because overexpression of PTEN, AKT, or Ras could shift the information flow (Fig 4D). The cell type-specific inhibitor effects are consistent with the signal flow in the network in the three cell types (cf. Fig 4B).

To account for the cell type-specific dynamics upon inhibitor treatment and for the fact that signaling components such as AKT, ERK, and S6 integrate information (Schneider *et al*, 2012), the integrated response within 1 h was calculated based on the experimentally determined data points. The values of these integrals were simulated with our mathematical model. Subsequently, the cell type-specific model parameters for the strength of the inhibitors were estimated. The experimentally observed effects of the two inhibitors on integrated pAKT, ppERK, and pS6 responses (Fig 5A) were reproduced by the mathematical model, except for a slight underestimation of the AKT VIII effect on pS6 in 32D-EpoR cells (Fig 5B).

To quantify the impact of the two inhibitors, AKTVIII and U0126, on the regulation of cell cycle progression, the expression levels of *cyclinD2*, *cyclinG2*, and *p27* in response to 5 U/ml Epo stimulation for 3 h and inhibitor treatment in all three cell types were determined by quantitative RT–PCR. The observed expression pattern of the individual genes *cyclinD2*, *cyclinG2*, and *p27* was complex (Appendix Fig S16). However, the cell cycle indicator, as a coefficient which summarizes the influence of the individual components, showed a graded alteration to the doses of the two inhibitors (Fig 5C). Specifically, the cell cycle indicator was significantly reduced already at low doses of AKT VIII in mCFU-E cells, at intermediate AKT VIII doses in 32D-EpoR cells, and only at high AKT VIII doses in BaF3-EpoR cells (Fig 5C). The effect of U0126 dose on the cell cycle indicator was graded in a similar manner for the three cell types (Fig 5C).

Taken together, these data show that the effect of inhibition of the AKT and ERK pathways depends on the cellular context, and the main determinant is protein abundance.

### Linking Epo-induced signal processing to cell proliferation

Next, the molecular activity of the AKT-ERK signaling network was linked to cell proliferation. The integrated pS6 response and the cell cycle indicator quantify key cellular activities contributing to proliferation upon Epo stimulation and inhibitor treatment. On the one hand, pS6 serves as an indicator of the activity of the ribosomal

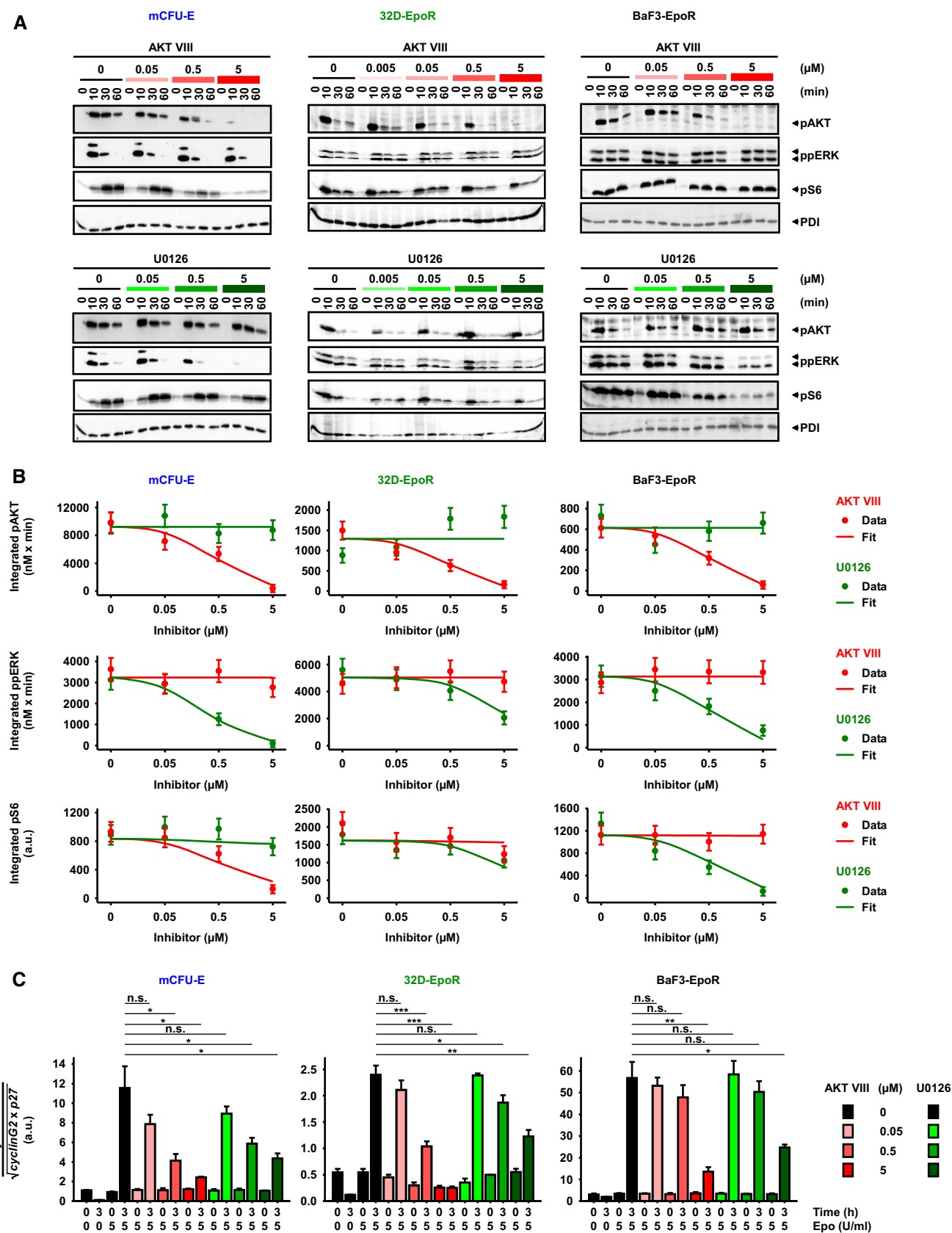

Figure 5.

protein S6 kinase, which is a pivotal regulator of protein synthesis and thus cell growth (Ruvinsky *et al*, 2006). On the other hand, the cell cycle regulator quantifies the balance of positive and negative regulators that control the entry of cells into the S phase of the cell cycle. Although the details of size regulation of mammalian cells remain poorly understood (Kafri *et al*, 2013), it is plausible that cellular context, such as protein abundance, will determine how protein synthesis and thus cell growth versus the G1-S transition rate control factor-induced proliferation. Cells with a long G1 phase will have sufficient time to grow, so that primarily the regulators of the G1-S transition, which are quantified by the cell cycle indicator, should control proliferation. Conversely, proliferation in cells with a short G1 phase should be more strongly controlled by growth as a necessary precondition for cell cycle progression.

Linear regression was applied to link the activity of AKT and Ras/MEK/ERK pathways with the cell cycle indicator (Fig 6A). The mathematical model of the signaling network was used to evaluate the integrated pAKT response and the integrated ppERK response in the absence or presence of 0.05, 0.5 or 5 μM of inhibitor upon 1-h stimulation with 5 U/ml Epo for mCFU-E, BaF3-EpoR, and 32D-EpoR cells. The measured values of the integrated pAKT response, the integrated ppERK response, and the cell cycle indicator for mCFU-E, BaF3-EpoR, and 32D-EpoR cells yielded high correlation, indicating that the effects of the inhibitor treatment on the cell cycle indicator are explained very well by changes in the integrated pAKT response and the integrated ppERK response with only slight deviation at the highest inhibitor doses (Fig 6B; Appendix N).

To quantitatively connect the integrated pS6 response and the cell cycle indicator to cell proliferation upon Epo stimulation and inhibitor treatment (Fig 6A), the proliferation of mCFU-E, BaF3-EpoR, and 32D-EpoR cells in response to 5 U/ml Epo and in the absence or presence of 0.005, 0.05, 0.5 and 5 μM of AKT VIII or U0126 was measured. The analysis shown in Appendix O (Fig 6C) revealed that both variables are not correlated and therefore are likely to be regulated independently. Multiple linear regression analysis was performed to link the integrated pS6 response and/or the cell cycle indicator with proliferation in mCFU-E, BaF3-EpoR, and 32D-EpoR cells upon Epo stimulation and inhibitor treatment, and the best model was selected based on Akaike's information criterion (Burnham & Anderson, 2002). For mCFU-E cells, Epo-induced proliferation under inhibitor treatment was described best as a function of the integrated pS6 response only ($R^2 = 0.89$), whereas for BaF3-EpoR cells and 32D-EpoR cells, the proliferation data were best described based on the cell cycle indicator ($R^2 = 0.86$ and $0.81$, respectively; Fig 6D). Noteworthy, the models with contribution of both the integrated pS6 response and the cell cycle indicator to proliferation were not significantly more informative than the models of individual contributions (Appendix O) but predicting proliferation in mCFU-E, BaF3-EpoR, and 32D-EpoR cells upon Epo stimulation and inhibitor treatment similarly well (Appendix Fig S26).

In summary, the quantitative dynamical pathway model of Epo-induced signaling was linked to the phenotypic parameter proliferation rate by linear regression. The dependence of mCFU-E proliferation on pS6 indicates that for these rapidly proliferating cells, protein synthesis and cell growth primarily control proliferation. By contrast, for BaF3-EpoR and 32D-EpoR cells, the cell cycle indicator was best predictive for proliferation upon Epo stimulation and inhibitor treatment.

## Combinatorial effects of AKT and ERK inhibitors predicted by the integrative model

To validate the quantitative link between Epo-induced signaling and proliferation, the integrative mathematical model (Fig 6A) was used, which was established for a single Epo dose, to predict proliferation in response to a broad range of Epo concentration and in response to overexpression of the negative regulators of AKT signaling, SHIP1, and PTEN (Fig 3A and B) for mCFU-E and BaF3-EpoR cells. Overall these phenotype predictions by the mathematical model (Fig 7A, upper panels) were in good agreement ($R^2 = 0.88$; Appendix Fig S19) with the experimental data (Fig 7A, lower panels). In agreement with the experimental data, the mathematical model predicted that there was no effect of overexpression of SHIP1 or PTEN on the $EC_{50}$ of Epo-induced proliferation of BaF3-EpoR, whereas in mCFU-E cells a small effect was detectable and overexpression of PTEN consistently gave rise to the highest $EC_{50}$ values. The $EC_{50}$ values estimated for the experimentally measured proliferative responses of the wild-type mCFU-E ($0.27 \pm 0.05$) and BaF3-EpoR ($0.68 \pm 0.46$) cells were in line with our initial observations (see Fig 1A). At very low Epo concentrations, the mathematical model predicted an elevated baseline proliferation for wild-type and SHIP1-overexpressing mCFU-E cells that was not detected in the experiment. At these low Epo concentrations, residual phosphorylation of signaling components is detectable and this information was utilized for calibration of the mathematical model. However, in the experiments the activation of signal transduction below a certain threshold apparently was not sufficient to elicit proliferation and therefore baseline proliferation is absent. Yet, in line with the experimental observations the mathematical model predicted that overexpression of PTEN decreased the proliferative response of mCFU-E cells and BaF3-EpoR cells the most. Further, the mathematical model correctly predicted that BaF3-EpoR and 32D-EpoR cells showed comparable Epo dose-dependent proliferation, an observation that was experimentally validated (Appendix Fig S20).

The integrative mathematical model was used to predict the proliferation of mCFU-E, BaF3-EpoR, and 32D-EpoR cells upon stimulation with 5 U/ml Epo and cotreatment with AKT VIII and U0126. Note that these model predictions (Fig 7B, upper panels) were based on the results of the multiple linear regression analysis with the treatment of single inhibitors only. For mCFU-E and 32D-EpoR cells, AKT inhibition was predicted to control Epo-induced cell proliferation in a dose-dependent manner, without or negligible combined effect of MEK inhibition. Only for BaF3-EpoR cells, the model indicated that Epo-induced cell proliferation is strongly inhibited by increasing doses of both AKT VIII and U0126, resulting in a combined effect of both drugs together (Fig 7B, upper right panel). These model predictions were experimentally validated (Fig 7B, lower right panel) and imply that cellular context determines whether molecularly targeted inhibitors of proliferation have combined effect or not.

To highlight that the protein abundance governs cell type-specific regulation of Epo-induced proliferation and as a consequence the sensitivity toward inhibitors, we prepared human CFU-E cells from CD34[+] cells mobilized into the peripheral blood of three healthy donors. By means of mass spectrometry, we quantified 6,925 proteins, of which 5,912 proteins were shared among the hCFU-E cells from the three independent donors (Fig 8A). Next, we applied

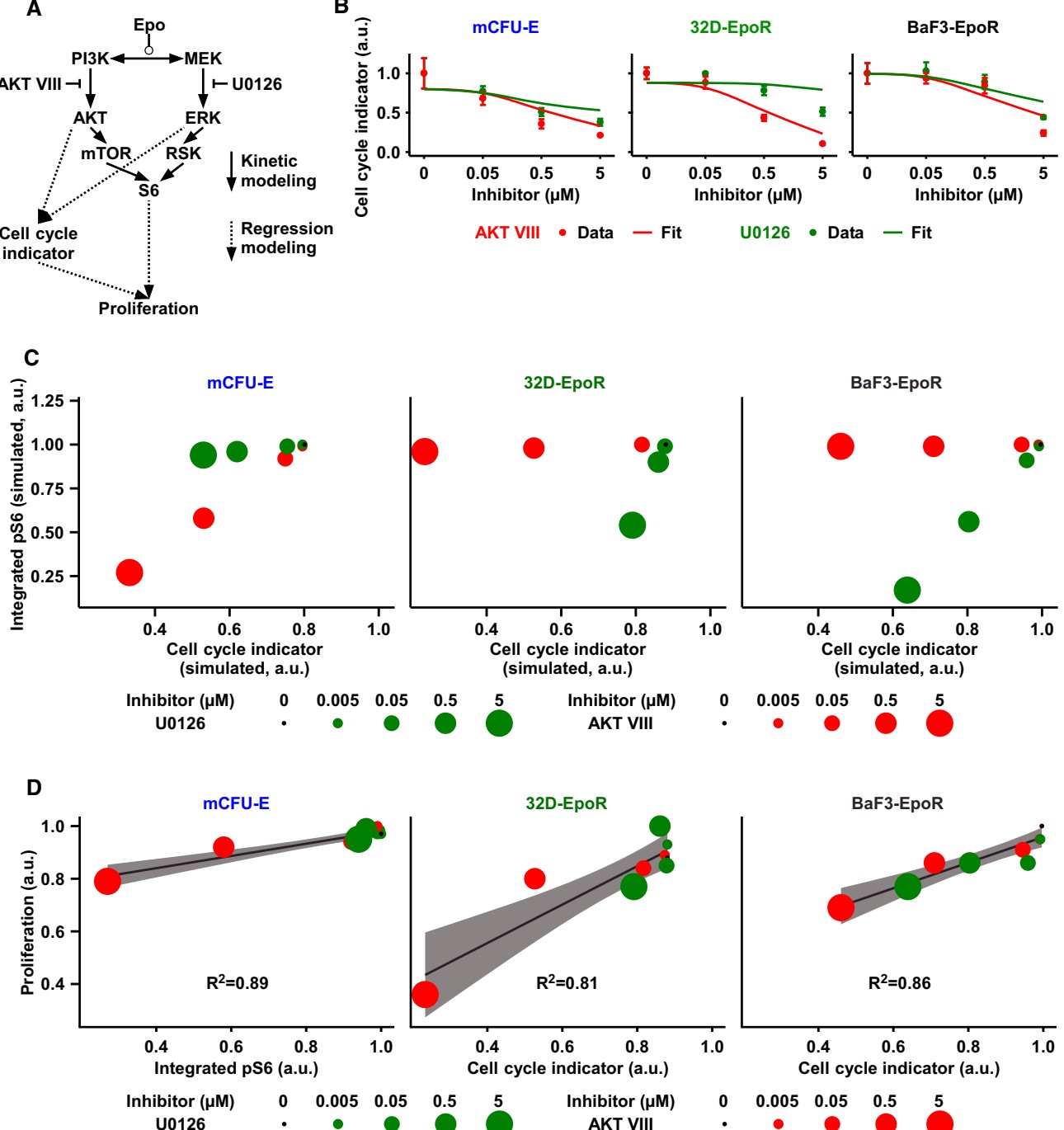

**Figure 6. Linking cell cycle and integrated pS6 to proliferation.**

A   Scheme of the mathematical model. Integrated pAKT and ppERK obtained from our kinetic model were linked by linear regression analysis to measured cell cycle indicator. Similarly, cell cycle indicator and integrated pS6 were linked to measured proliferation by linear regression analysis and model selection.

B   Linear regression fit for the respective cell cycle indicator of experimental data. Data points represent mean ± standard deviation, *N* = 3. Solid lines represent the linear regression fit.

C   Simulated integrated pS6 response versus simulated cell cycle indicator of mCFU-E, BaF3-EpoR, and 32D-EpoR cells upon 5 U/ml Epo stimulation and single inhibitor treatment at indicated doses.

D   Linear regression modeling and model selection revealed that while proliferation of mCFU-E cells is best described by integrated pS6 only, for 32D-EpoR and BaF3-EpoR cells proliferation correlates mainly with cell cycle indicator. Proliferation was measured, and integrated pS6 and cell cycle indicator were simulated with our mathematical model. The linear regression model selection is based on Akaike's information criterion. The respective correlation is given. For details, see Appendix N and O.

Source data are available online for this figure.

**Figure 7. Prediction of cell type-specific proliferation.**

A   Upper panel: Model prediction of Epo-dependent proliferation in mCFU-E and BaF3-EpoR cells overexpressing SHIP1 or PTEN. Solid lines represent model trajectories. Lower panel: Experimental validation of PTEN and SHIP1 overexpression effects on Epo-dependent proliferation. Proliferation was assessed using [³H]-thymidine incorporation 14 h (mCFU-E) or 38 h (BaF3-EpoR) after retroviral transduction with PTEN or SHIP1 construct for overexpression. Data are represented as mean ± standard deviation, $N = 3$. $EC_{50}$ values are given.

B   Upper panel: Model prediction of proliferation upon combined inhibitor treatment and 5 U/ml Epo stimulation. Maximum proliferation was scaled to 1. Lower panel: Experimental validation of proliferation upon combined inhibitor treatment and 5 U/ml Epo stimulation. Proliferation was measured as cell numbers after 14 h (mCFU-E) or 38 h (BaF3-EpoR, 32D-EpoR) with Coulter Counter. Maximum was scaled to 1.

Source data are available online for this figure.

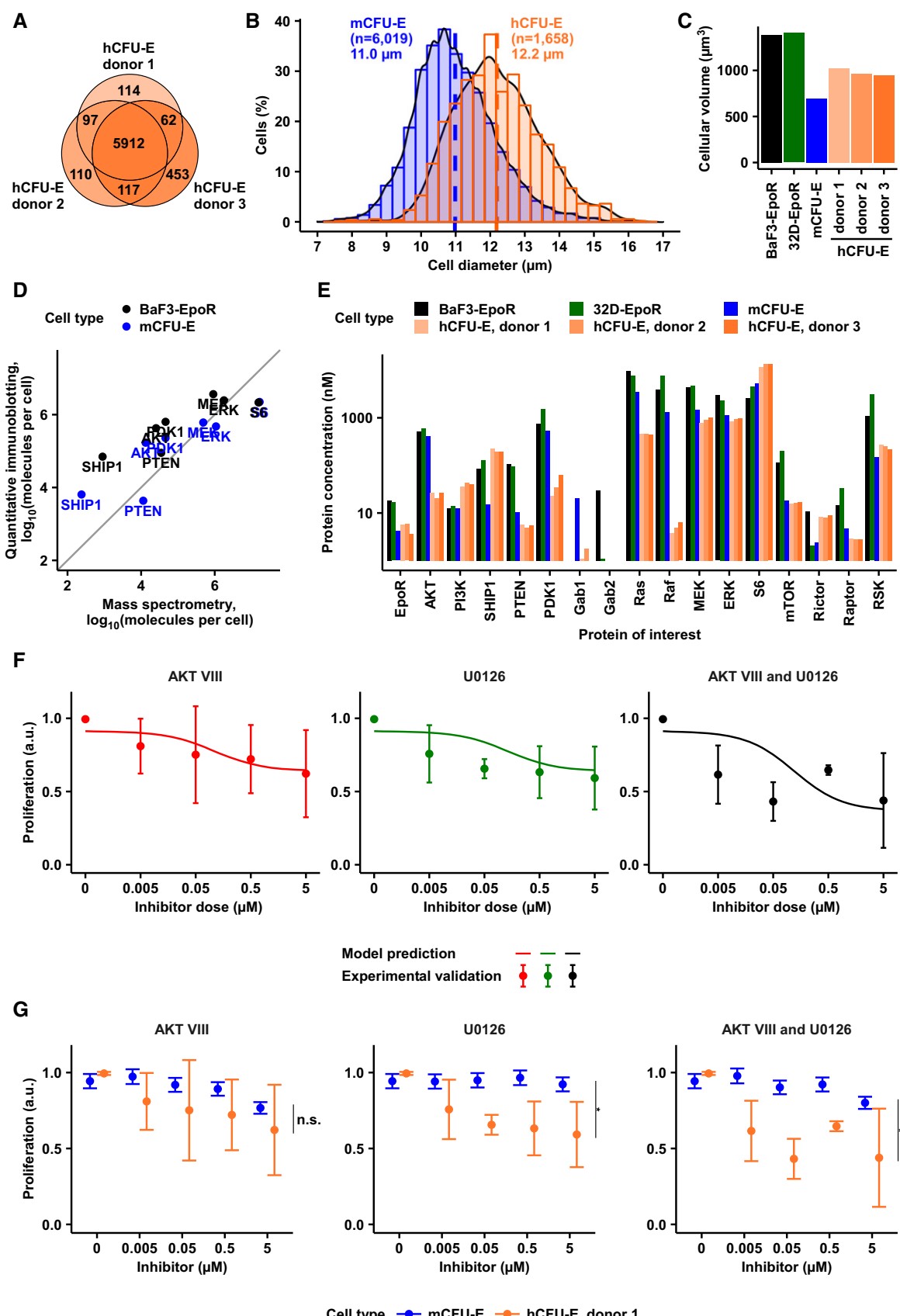

Figure 8.

◀

**Figure 8.  Predicting combined effects of inhibitor treatment on Epo-induced proliferation of hCFU-E cells solely based on protein abundance.**
Human CFU-E cells were prepared from CD34[+] cells mobilized into the peripheral blood of three independent stem cell donors.

A   Venn diagram representing overlap of hCFU-E proteome from three independent healthy donors. In total, 6,925 proteins were quantified.

B   Size determinations of hCFU-E compared to mCFU-E cells. Cell diameter was measured by imaging flow cytometry. Cytoplasm was stained with calcein, and nuclei were stained with DRAQ5. Probability density function of size distribution with indicated mean diameter of hCFU-E and mCFU-E cells. All cells were growth factor-deprived and unstimulated.

C   Cytoplasmic volumes of BaF3-EpoR, 32D-EpoR, mCFU-E, and the hCFU-E cells. All volumes were determined using imaging flow cytometry as described in (B).

D   Correlation between protein molecules per cell for key signaling components determined by quantitative immunoblotting or quantitative mass spectrometry and the "proteomic ruler" method (Wiśniewski *et al*, 2014) in BaF3-EpoR and mCFU-E cells. Diagonal line as guide to the eye. Spearman's rank-based coefficient of correlation ρ = 0.88.

E   Protein abundance of key signaling components was determined by mass spectrometry of whole cell lysates of hCFU-E cells. Copy numbers of proteins per cell were obtained by the "proteomic ruler" method and converted to cytoplasmic concentrations with the volumes calculated from (B) and shown in (C).

F   Model prediction and experimental validation of proliferation upon single or combined inhibitor treatment with AKT VIII and U0126 upon 5 U/ml Epo stimulation. Maximum proliferation was scaled to 1. Proliferation was measured as cell numbers, counted with a hemocytometer after 96 h by trypan blue exclusion assay. Solid lines represent model trajectories. Experimental data are represented as mean ± standard deviation. N = 3.

G   Comparison of the effect of single or combinatorial inhibitor treatment on Epo-induced proliferation of murine and human CFU-E cells. Proliferation was measured upon single or combined inhibitor treatment with AKT VIII and U0126 upon 5 U/ml Epo stimulation. Numbers of mCFU-E cells were determined with the Coulter Counter after 14 h, whereas numbers of hCFU-E cells were determined with a hemocytometer after 96 h by the trypan blue exclusion assay. Maximum proliferation was scaled to 1. Data are represented as mean (N = 3), and error bars indicate standard deviation. Data of mCFU-E cells are presented in Figs 6D and 7B, and data of hCFU-E cells are taken from Fig 8F. Tukey multiple comparison of means, n.s. = not significant, *P < 0.05.

Source data are available online for this figure.

the "proteomic ruler" method (Wiśniewski *et al*, 2014) to calculate the copy numbers of individual proteins per cell. To convert these numbers into cytoplasmic concentrations, we measured the average cell size by imaging flow cytometry (Fig 8B) and calculated the cytoplasmic volume from these data. The average cytoplasmic volume of the hCFU-E of the three donors was comparable but considerably larger than the volume of the mCFU-E cells (Fig 8C). To relate the results obtained with the proteomic ruler method to the previous measurements in mCFU-E, BaF3-EpoR, and 32D-EpoR utilizing quantitative immunoblotting and recombinant protein standards, we performed additional mass spectrometric measurements for mCFU-E and BaF3-EpoR cells. As shown in Fig 8D, for the key signaling components in the two cell types, the number of molecules per cell determined with the different techniques showed a good correlation (Spearman's rank-based coefficient of correlation ρ = 0.88), validating our determinations by quantitative immunoblotting and confirming that the snapshot measurement by mass spectrometry yielded reliable results. Surprisingly, the protein abundance of the key signaling proteins determined for hCFU-E cells showed very low variance between the three independent donors but were highly distinct from the values obtained for mCFU-E cells as well as for the murine cells lines BaF3-EpoR and 32D-EpoR (Fig 8E). For example, Ras, Raf, and AKT were present at much lower levels in hCFU-E compared to mCFU-E, BaF3-EpoR, and 32-EpoR cells, whereas for PI3 kinase elevated levels were observed in hCFU-E cells. As expected, the levels of the EpoR were comparable in hCFU-E and mCFU-E and elevated to the same extent in Ba3-EpoR and 32D-EpoR cells. To link the signaling layer to proliferation, we applied our concept that larger (see Fig 8B) and more slowly dividing (Appendix Fig S28) cells regulate proliferation primarily by the control of G1-S progression (Appendix Q). Since the measured protein abundance of the key signaling proteins was highly comparable in the hCFU-E cells of the three donors, we used the average concentrations of these proteins to predict with our mathematical model the impact of the AKT inhibitor and MEK inhibitor on Epo-supported cell proliferation in hCFU-E cells. Without any further information (Appendix S), the mathematical model predicted the effect of the AKT inhibitor, the MEK inhibitor, and a combination

thereof on Epo-induced proliferation in hCFU-E cells (Fig 8F). To validate this model prediction experimentally, we quantified the numbers of hCFU-E cells after 96 h of stimulation with 5 U/ml Epo and individual or combined treatment with AKT VIII and U0126. Since hCFU-E cells from donor 3 responded most strongly to Epo, showing a significantly higher fold change in the numbers of cells compared with donor 1 (P = 0.038) and donor 2 (P = 0.009) after 3 days of subcultivation (Appendix Fig S31) and thus a advantageous signal-to-noise ratio, we focused the analyses of proliferation inhibition on hCFU-E cells from this donor in biological triplicates (Fig 8F). We observed that the Epo-induced proliferation of hCFU-E is impaired upon treatment with both inhibitors individually and their combination. Pearson's coefficient of correlation $R^2$ = 0.88 of the results obtained in two independent experiments confirmed the reproducibility of our experimental observations (Appendix Fig S32). When the impact of AKT VIII and U0126 and the combination thereof on the Epo-induced proliferation in hCFU-E and in mCFU-E was compared, we observed that the experimental data on proliferation (Fig 8G) were in line with the model predictions for hCFU-E (Fig 8F) and mCFU-E cells (Figs 6D and 7B), respectively. The results depicted in Fig 8G show that the impact of AKT VIII on Epo-induced proliferation was comparable in hCFU-E and mCFU-E (P = 0.3), whereas MEK inhibition (P = 0.03) and the combinatorial inhibitor treatment (P = 0.03) exhibited significantly larger effects on Epo-promoted proliferation in hCFU-E cells (Fig 8G).

These findings showcase that protein abundance can be reliably measured from snapshot data of human material. Based on these data, our integrative mathematical model allows to evaluate the impact of inhibitors *in silico* and thus may serve to improve the treatment of proliferative disorders such as tumors driven by exacerbated growth factor signaling.

# Discussion

By a combination of quantitative measurements with mathematical modeling, we show that proliferation upon Epo stimulation and inhibitor treatment of mCFU-E cells is well predicted by integrated

pS6 as a proxy for cell growth, whereas integrated pAKT and ppERK regulating cell cycle progression described proliferation upon Epo stimulation and inhibitor treatment of hCFU-E, BaF3-EpoR, and 32D-EpoR cells best. Importantly, the experimentally observed differences in the dynamics of Epo-induced activation of AKT, ERK, and S6 in mCFU-E, BaF3-EpoR, and 32D-EpoR cells are primarily due to cell type-specific abundance of key signaling components.

In principle, the link from Epo-induced signaling to cell proliferation could be established through cell cycle progression or cell growth or a combination of both. To investigate the connection of Epo-induced AKT and ERK pathway activation to proliferation, we linearly connected the integrated responses of pAKT and ppERK to cell cycle progression and/or the integrated pS6 response reflecting cell growth.

Since early measurements can be indicative of the outcome of cell decisions (Shokhirev *et al*, 2015), we analyzed the expression of three cell cycle genes after 3 h of Epo stimulation and calculated the cell cycle indicator as a coefficient to quantify cell cycle progression. The three cell cycle genes *cyclinD2*, *cyclinG2*, and *p27* considered were identified from microarray data as differentially regulated genes (Appendix Fig S4). At saturating Epo doses, the individual genes *cyclinD2*, *cyclinG2*, and *p27* were expressed to similar extent in mCFU-E and BaF3-EpoR cells (Fig 1F). However, treatment with AKT inhibitor had only a slight impact on the expression of *cyclinD2* in mCFU-E cells (Appendix Fig S18), but due to upregulation of the cell cycle repressors *cyclinG2* and *p27* resulted in a strong reduction of the cell cycle indicator (Fig 6B). Treatment with MEK inhibitor had only mild effects on the expression of *cyclinG2* and *p27* alone (Appendix Fig S16), but the alterations in the expression level of both genes together in the denominator of the cell cycle indicator explained the observed effects of inhibitor treatment on Epo-induced proliferation in BaF3-EpoR cells (Fig 7B). Therefore, the cell cycle indicator as a coefficient summarizing alterations in the expression of three genes involved in the control of cell cycle progression is more informative than alterations in the expression of individual genes alone. To quantify cell growth, we utilized the integrated pS6 response. It was shown that embryonic fibroblasts from mutant mice, which cannot phosphorylate S6, are reduced in cell size but accelerated in cell division (Ruvinsky *et al*, 2005). Work in chicken erythroblasts suggested that the length of the G1 phase of the cell cycle ensures proper balancing between growth and cell cycle progression rates (Dolznig *et al*, 2004). We showed that at saturating Epo concentrations the doubling time, which is considered as a function of cell growth controlled by integrated pS6, correlates with difference in size of mCFU-E and BaF3-EpoR as well as 32D-EpoR cells (Appendix Q). In line with this assumption, hCFU-E cells, which are considerably larger than mCFU-E cells (Fig 8B), also doubled their number more slowly than mCFU-E cells (Fig 1B; Appendix Fig S28).

We observed that the impact of AKT or MEK inhibitors on Epo-induced proliferation in BaF3-EpoR cells and 32D-EpoR cells is explained by the sensitivity of the cell cycle indicator, whereas for mCFU-E the impact on the integrated pS6 response is most informative. To ensure sufficient oxygen supply, the oxygen-carrying capacity of mature erythrocytes has to be tightly controlled (Hawkey *et al*, 1991). Therefore, the size of erythroid progenitor cells is connected to a physiological function and is decisive to maintain functionality. On the contrary, the hematopoietic cell lines BaF3-EpoR and 32D-EpoR can proliferate unlimited in the presence of Epo

without fulfilling additional tasks. Therefore, here rapid cell cycle progression is key and the step controlled by integrated pS6 is no longer rate-limiting. This is in line with reports proposing that cell lines evolve toward rapid cell growth (Pan *et al*, 2008), whereas in primary cell maintenance of a specific function that depends on cell size can be key. Therefore, the specific link from factor-induced cell signaling to proliferation can be cell type specific.

In general, linear models are a simple and robust quantification of the input–output function. However, it is more difficult to rationalize mechanistically how the different variables are related. Therefore, it is still under debate to which detail signaling mechanisms should be modeled and how they can be connected to phenotypic behavior (Saez-Rodriguez *et al*, 2009; Birtwistle *et al*, 2013). This study investigates three layers of the cellular response to growth factors, which operate on distinct time scales: signal transduction, gene expression, and cell proliferation (Appendix Fig S27). While it is recognized that signal processing occurs across these layers and time scales (Klamt *et al*, 2006), so far data-based models of growth factor signaling have focused—with few exceptions (Kirouac *et al*, 2013)—on the fast (< 1 h) signal transduction layer. This focus has, at least in part, been due to the fact that the molecular details of signal transduction are overall better understood than those of the downstream layers. Consequently, data-based mathematical models that connect signal transduction, gene expression, and cell cycle regulation present a critical challenge that as yet is largely unmet (Gonçalves *et al*, 2013). To address this challenge, we developed a modular modeling approach that links mechanism-based models of signal transduction with conceptually simple but effective, linear regression models for the downstream layers. A practical rationale for this approach is that many targeted drugs for cancer therapy address signal transduction (reviewed in Saez-Rodriguez *et al*, 2015). Hence, our approach describes the immediate action of the drugs on signal transduction in mechanistic detail and, in turn, infers from the signaling dynamics the proliferative behavior of the cells by a linear regression model. This approach succeeded in quantitatively predicting proliferation inhibition by combinations of AKT and ERK pathway inhibitors in hCFU-E, mCFU-E, BaF3-EpoR, and 32D-EpoR cells.

Signal transduction networks, such as the AKT and ERK pathways, have a common core topology irrespective of cell type. Dynamic pathway models have usually been developed for specific cells by estimating a set of kinetic parameters (mainly enzymatic rate constants) from measurements of model variables (e.g. phosphorylation states) in a given cell (Kim *et al*, 2010). The concept of protein abundance determining the utilization of connections and the dynamics of cellular signal transduction (Appendix F) is not limited to hematopoietic cell types and can be extended to other cells (Merkle *et al*, 2016). It was shown that the abundance of growth factor receptors correlates with growth factor responses and AKT/ERK bias in diverse breast cancer cell lines (Niepel *et al*, 2014). We show here that the mere abundance of a cytokine receptor such as the EpoR is not sufficient to explain proliferative responses. Whereas mCFU-E and hCFU-E cells harbor comparable levels of the EpoR, the abundance of key signaling molecules is very distinct and culminates in major difference in the sensitivity of their Epo-induced proliferative responses toward the AKT and MEK inhibitor. Further refinements might become necessary, as the signaling network topology implemented here involves

simplifications and neglects presence of different isoforms such as Gab1 in hCFU-E and mCFU-E cells and Gab2 in BaF3-EpoR and 32D-EpoR cells (Table 1, Fig 8E; Appendix F), which might need to be included in a larger context as these low-abundant proteins can be the bottlenecks of signaling (van den Akker *et al*, 2004; Shi *et al*, 2016). However, even the cell type-specific wiring of feedback loops in signal transduction (Klinger *et al*, 2013; D'Alessandro *et al*, 2015; Stites *et al*, 2015) ultimately depends on protein expression and can thus be captured by the conceptual framework proposed here. By combining the analysis of protein abundance and structure data, Kiel *et al* (2013) showed that the abundance of signaling components determined the cell context-specific topology of the ErbB signaling network. Further, predicting signaling dynamics from (comparatively simple) static measurements of protein abundance may become of practical use for prognosis, as shown for the JNK network in neuroblastoma (Fey *et al*, 2015). However, for cancer cells, oncogenic mutations that affect enzymatic activities of specific proteins or their binding interactions might also require adjustment of selected biochemical parameters or network topologies (Kiel & Serrano, 2009; AlQuraishi *et al*, 2014).

To determine the abundance of signaling components, we used in our approach quantitative immunoblotting (Schilling *et al*, 2005) and quantitative mass spectrometry in combination with the "proteomic ruler" method that is based on the determination of total protein concentrations relative to the abundance of histones (Wiśniewski *et al*, 2014). The results shown in Fig 8C demonstrate that the abundance of the signaling components determined by quantitative immunoblotting is very comparable to the results obtained by quantitative proteome-wide mass spectrometric measurements (Kulak *et al*, 2014; Hein *et al*, 2015). Protein abundance in tumor material can be quantified using a label-free approach as described above or a super-SILAC approach employing a labeled reference cell line (Zhang *et al*, 2014). It has been suggested that the abundance of proteins is characteristic for different cell types (Wilhelm *et al*, 2014) and can even be used to separate subtypes of cancer cell lines (Deeb *et al*, 2012). Even if not all proteins can be quantified, mathematical modeling linked with sensitivity analysis provides a hierarchy of important network components as it is widely applied to investigate parametric dependence of model properties (Schilling *et al*, 2009; Maiwald *et al*, 2010; Perumal & Gunawan, 2011). In our approach, the relative impact of protein abundance on integrated pS6 was calculated and it was shown that in mCFU-E cells, integrated pS6 was controlled mainly by AKT, whereas in 32D-EpoR and BaF3-EpoR cells, integrated pS6 mainly depended on Ras, Raf, MEK, ERK, and RSK (Table 1; Appendix Fig S27). These findings suggest that at points of signal integration (such as pS6), the more highly expressed signaling pathway(s) dominates signal processing. This is in stark contrast to metabolic regulation where the least abundant enzymes usually control the metabolic flux (Heinrich & Schuster, 1996).

We propose that differences in the protein abundance of signaling components can also explain differential sensitivity to inhibitor treatment. While the information flow in 32D-EpoR cells is similar to BaF3-EpoR cells, their proliferation behavior under inhibitor treatment is akin to mCFU-E cells. Profile likelihoods for the inhibitor parameters suggested that 32D-EpoR cells are significantly less sensitive to U0126 than mCFU-E and BaF3-EpoR cells (Appendix Fig S22). 32D-EpoR cells exhibit higher abundance of MEK than mCFU-E and BaF3-EpoR cells (Table 1). MEK is the target of U0126 and therefore the elevated MEK levels probably buffer the inhibitor's effect (Appendix Fig S27). This is in line with the observation that overexpression of activated MEK1 conferred U0126 resistance in HepG2 cells (Huynh *et al*, 2003).

In summary, the integrative mathematical model provides new insights into cell type-specific mechanisms regulating Epo-induced proliferation in primary erythroid progenitor cells and hematopoietic cell lines. We dissect the cell type-specific contribution of pAKT, ppERK, and pS6 on cell growth and cell cycle progression and thereby establish an important basis for rational interference of cellular information processing and the effects of inhibitor treatment on Epo-induced proliferation. Our study demonstrates that the determination of the abundance of signaling components is sufficient to adapt the integrative mathematical model and predict sensitivities for individual inhibitors or combinations thereof and thereby opens new possibility to test and verify therapeutic interventions.

# Materials and Methods

### Primary cell and cell line cultures

All animal experiments were approved by the governmental review committee on animal care of the state Baden-Württemberg, Germany (reference number DKFZ215). To obtain primary human erythroid progenitor CFU-E cells, CD34$^+$ cells, mobilized into the peripheral blood of healthy donors after written consent, were sorted by autoMACS (CD34-Multisort Kit, Miltenyi Biotech). Sample collection and data analysis were approved by the Ethics Committee of the Medical Faculty of Heidelberg. CD34$^+$ cells were expanded using Stem Span SFEM II (StemCell Technology) supplemented with Stem Span CC100 (StemCell Technology). After 7 days of expansion, cells were differentiated. For differentiation, cells were cultivated in Stem Span SFEM II supplemented with 10 ng/ml mIL-3 (R&D Systems), 50 ng/ml mSCF (R&D Systems), and 6 U/ml Epo alfa (Cilag-Jansen) as reported previously (Miharada *et al*, 2006). After 3 days of cultivation, human CFU-E cells were employed to perform experiments.

Primary murine erythroid progenitor CFU-E cells were prepared from fetal livers of E13.5 Balb/c mice. Fetal liver cells (FLC) were treated with Red Blood Cell Lysing Buffer (Sigma-Aldrich) to remove erythrocytes. For negative depletion, FLC of 40 livers were incubated with rat antibodies against the following surface markers: GR1, CD41, CD11b, CD14, CD45, CD45R/B220, CD4, CD8 (all BD Pharmingen), Ter119 (eBioscience), and with YBM/42 (gift from Suzanne M. Watt, Oxford, UK) for 60 min at 4°C. After washing, cells were incubated for 30 min at 4°C with anti-rat antibody-coupled magnetic beads (Miltenyi Biotech) and negative sorted with MACS columns according to manufacturer's instructions. Sorted mCFU-E cells were cultured for 14 h in Panserin 401 (Pan Biotech) and 50 μM β-mercaptoethanol supplemented with 0.5 U/ml Epo (Cilag-Jansen).

BaF3 and 32D cells were cultured in RPMI 1640 (Invitrogen) including 10% WEHI conditioned medium as a source of IL3 and

supplemented with 10% FCS (Invitrogen), penicillin (100 U/ml) and streptomycin (100 mg/ml).

## Plasmids and retroviral transduction

Retroviral expression vectors were pMOWS-puro-MCS/M2 (Ketteler *et al*, 2002). For stable transfection of BaF3 and 32D cells with murine EpoR, pMOWS-Kz-HA-EpoR was generated in our laboratory (Becker *et al*, 2008). The murine SHIP1, AKT1, RasV12G, RSK1, or human PTEN cDNAs were cloned into pMOWSnr-MCS/M2, in which the puromycin resistance gene was replaced by the LNGFR cDNA (Miltenyi Biotech) that allows magnetic bead selection of transduced cells. Transient transfection of Phoenix-eco cells with retroviral expression vectors was performed by calcium phosphate precipitation. To ensure an efficient uptake of DNA, Phoenix cells were incubated for 6 h in DMEM medium (Invitrogen) supplemented with 25 μM chloroquine and 10% FCS. Subsequently, the medium was replaced by IMDM (Invitrogen) containing 50 μM β-mercaptoethanol and 30% FCS and incubated for another 18 h. Each 250 μl of the filtered retroviral supernatant of pMOWS-Kz-HA-EpoR supplemented with 8 μg/ml polybrene was then used to transduce $1 \times 10^5$ BaF3 or 32D cells, which were centrifuged for 2 h at $340 \times g$ and 37°C. Selection with 1.5 μg/ml puromycin (Sigma-Aldrich) started 48 h after transduction resulting in BaF3-EpoR and 32D-EpoR cells. Surface expression of EpoR in BaF3 and 32D cells was verified by flow cytometry using an antibody against HA (Roche) and a Cy-5-labeled anti-rat antibody (Jackson Immuno Research).

For overexpression experiments, $5 \times 10^6$ cells were transduced using 4.5 ml retroviral supernatant supplemented with 8 μg/ml polybrene in a six-well plate and centrifuged for 3 h at $340 \times g$ and 37°C. Following spin infection, cells were cultured for 14–16 h in the standard media. Positively transduced cells were selected using MACSelect LNGFR selection kit (Miltenyi Biotech) according to manufacturer's instructions. Cells were either used immediately for experiments or further cultivated. Level of overexpression was always verified at the day of experiment using immunoblotting.

## Time course experiments, cell lysis, quantitative immunoblotting, and mass spectrometry

Murine CFU-E cells were washed three times with Panserin 401 and were growth factor-deprived in the medium supplemented with 50 μM β-mercaptoethanol for 1–2 h while BaF3-EpoR cells were washed with RPMI 1640 and deprived for 4–5 h and 32D-EpoR cells for 3 h, in the medium supplemented with 1 mg/ml BSA (Sigma-Aldrich) at 37°C depending on type of experiment. Subsequently, cells were stimulated with 0.5–50 U/ml Epo (Cilag-Jansen) or cells were first pretreated with Akt inhibitor VIII (Millipore) and MEK1/2 inhibitor U0126 (Cell Signaling) for 30 min before Epo stimulation. For each time point, $0.4–1 \times 10^7$ cells were taken from the pool of cells and lysed by adding 2× 1% Nonidet P-40 lysis buffer as described elsewhere (Becker *et al*, 2010). Immunoprecipitation was performed by adding the respective antibody, protein A or G sepharose (GE Healthcare) to the lysates or the calibrator protein. Sample loading on SDS–PAGE was randomized and corrected with a spline-based normalization strategy to avoid correlated blotting errors (Schilling *et al*, 2005). Blots were

developed using ECL Western Blotting Reagents (GE Healthcare) and subsequently detected on a Lumi-Imager F1™ (Roche Diagnostics). Quantification of immunoblots was performed using Image Quant Software (GE). Antibodies were removed by treating the blots with stripping buffer as described previously (Klingmüller *et al*, 1995).

The following antibodies were used: anti-pTyr (4G10; Upstate/Millipore), anti-Ras (Calbiochem), anti-PDI (Stressgen), anti-β-actin (Sigma), and anti-PI3K p85 (N-SH2), anti-Gab1 (all from Upstate), and anti-EpoR (M20), anti-SHIP1, anti-Gab2, anti-Raf (all from Santa Cruz), and anti-Akt, anti-pAkt Ser473, anti-pAkt Thr308, anti-phospho-p44/p42 MAPK (Thr201/Tyr204), anti-p44/p42 MAPK, anti-PTEN, anti-S6, anti-pS6 (Ser235/236), anti-pS6 (Ser240/244), anti pRSK (Thr359/Ser363), anti-RSK, anti-mTOR, anti-Rictor, anti-Raptor, anti-PDK1 (all from Cell Signaling) as well as secondary horseradish peroxidase-coupled antibodies (Amersham Biosciences/Dianova).

For determination of protein concentrations of lysates of respective cells and calibrators for the respective protein, they were subjected to quantitative immunoblotting. Calibrators were either commercially available [Ras, Raf, PDK1, S6 (Abnova), ERK (Invitrogen)], or self-made.

Human CFU-E cells were washed three times with IMDM GlutaMAX and growth factor-deprived in IMDM GlutaMAX supplemented with 1 mg/ml BSA at 37°C for an hour. $2.5 \times 10^6$ cells were lysed by adding 2× RIPA buffer (100 mM Tris pH 7.4, 300 mM NaCl, 2 mM EDTA, 2 mg/ml deoxycholate, 1 mM $Na_3VO_4$, 5 mM NaF). Similarly, $1 \times 10^7$ growth factor-deprived BaF3-EpoR and mCFU-E cells were lysed. Whole cell lysates were sonicated and protein yield was determined by BCA assay (Thermo).

Each 75 μg of protein lysates of hCFU-E cells, 100 μg of BaF3-EpoR protein lysates, or 30 μg of mCFU-E protein lysates was fractionated by 10% 1D SDS–PAGE for 105 min. Gels were then stained with Coomassie (Invitrogen), each lane was divided into five segments, and each segment was cut into smaller pieces. In-gel digestion was performed as previously described (Boehm *et al*, 2014). Samples were analyzed by liquid chromatography, nanoelectrospray ionization, and tandem mass spectrometry with a Q-Exactive Plus (Thermo). Raw files obtained were then analyzed by MaxQuant, version 1.5.3.30, as described elsewhere (Cox & Mann, 2008) by MaxQuant (version: 1.5.0.12). MaxLFQ algorithm (Cox *et al*, 2014) was employed for the quantification purpose. Protein copy numbers per cell were obtained by applying the "proteomic ruler" method (Wiśniewski *et al*, 2014).

Cytoplasmic volumes of mCFU-E and BaF3-EpoR cells were determined by confocal microscopy with a Zeiss LSM 710. The cytoplasmic volume of 32D-EpoR cells was determined relatively to BaF3-EpoR cells by imaging flow cytometry. To determine cellular volumes of hCFU-E cells, the cytoplasm was stained with calcein (eBioscience) and DNA was stained with DRAQ5 (Cell Signaling). Imaging flow cytometry was performed on an Amnis ImageStream$^X$ (Merck Millipore), and data were analyzed with the IDEAS Application v5.0 (Merck Millipore).

The raw data of all qualitative and quantitative immunoblots of this work can be accessed as Source Data for the respective figure. Data of quantified immunoblots have also been uploaded through Excemplify (Shi *et al*, 2013) to the SEEK platform (http://seek.sbepo.de/; Wolstencroft *et al*, 2015) and FAIRDOMHub (https://fairdomhub.org/).

The mass spectrometry proteomics data have been deposited to the ProteomeXchange Consortium via the PRIDE (Vizcaíno *et al*, 2016) partner repository with the dataset identifier PXD004816.

## Microarray analysis

After washing with serum-free medium, BaF3-EpoR cells were growth factor-deprived for 5 h and resuspended in RPMI supplemented with 1 mg/ml BSA. RNA samples were taken at 0, 1, 2, 3, 4, 5, 7, 18 h after 1 U/ml Epo stimulation.

Per time point, total RNA from $4 \times 10^6$ BaF3-EpoR cells was isolated using the RNeasy Mini Plus Kit (Qiagen). Gene expression analysis was conducted using Affymetrix Mouse Genome 2.0 Gene-Chip Arrays (Affymetrix).

Normalization was performed in the R environment together with the Bioconductor toolbox (http://www.bioconductor.org). Arrays were normalized via the Robust Multichip Analysis (Gautier *et al*, 2004). Subsequent probe annotation was handled with the Affymetrix mouse4302 annotation package (R package version 3.1.3). If multiple probes mapped to the same Gene ID, the one with the largest test interquartile range among all time points was selected. The expression data were deposited in the GEO database under accession number http://tinyurl.com/GSE72317.

## Quantitative RT–PCR

To generate cDNA of mCFU-E, BaF3-EpoR, 32D-EpoR cells, 1–2 μg of total RNA was transcribed with the QuantiTect Reverse Transcription Kit (Qiagen). Quantitative RT–PCR was performed using a LightCycler 480 in combination with the hydrolysis-based Universal Probe Library (UPL) platform (Roche Diagnostics). Crossing point (CP) values were calculated using the Second Derivative Maximum method of the LightCycler 480 Basic Software (Roche Diagnostics). PCR efficiency correction was performed for each PCR setup individually based on a dilution series of template cDNA. Relative concentrations were normalized using *HPRT* or *RPL32* as reference genes. UPL probes and primer sequences were selected with the Universal ProbeLibrary Assay Design Center (Roche Diagnostics).

| Gene name | Primer | | UPL No |
|---|---|---|---|
| | Forward | Reverse | |
| *CCND2* | CTGTGCATTTACACCGAC AAC | CACTACCAGTTCCCACTCCAG | 45 |
| *CCNG2* | CCACGCGATTGTATTTT GTC | AGCTGCGCTTCGAGTTTATC | 15 |
| *CDKN1B* | GAGCAGTGTCCAGGGAT GAG | TCTGTTCTGTTGGCCCTTTT | 62 |
| *DUSP4* | GTACCTCCCAGCACCAA TGA | GAGGAAAGGGAGGATTTCCA | 17 |
| *DUSP5* | GATCGAAGGCGAGAGA AGC | GGAAGGGAAGGATTTCAACC | 102 |
| *DUSP6* | TGGTGGAGAGTCGGT CCT | TGGAACTTACTGAAGCCACCT | 66 |
| *RPL32* | GCTGCCATCTGTTTTA CGG | TGACTGGTGCCTGATGAACT | 12 |
| *HPRT* | TCCTCCTCAGACCGCTTTT | CCTGGTTCATCATCGCTAATC | 95 |

## Cell proliferation assays

For proliferation assays, growth factor-deprived murine CFU-E cells were plated at a density of $20 \times 10^4$ cells/well growing for 14–20 h, while BaF3-EpoR and 32D-EpoR cells were plated at a density of $5 \times 10^4$ cells/well growing for 62 h, or $10 \times 10^4$ cells/well growing for 38 h, respectively. Growth factor-deprived human CFU-E cells were plated at a density of $8.75 \times 10^4$ cells/well growing for 96 h. Cells were cultured in their individual mediums supplemented with different doses of Epo (Cilag-Jansen), AKT inhibitor VIII (EMD Millipore), and MEK1/2 inhibitor U0126 (Cell Signaling). Cells were pre-incubated with the inhibitors for 30 min and subsequently stimulated with Epo. Human CFU-E cells were counted by trypan blue exclusion assay using a hemocytometer.

For Coulter Counter assay, cells were plated with appropriate densities in 24-well plates. After respective days, cell numbers were determined using a Coulter Counter Z2 (Beckman, particle size 5.00–12.00 μm for mCFU-E cells and 4.00–17.35 μm for BaF3-EpoR and 32D-EpoR cells).

[³H]-thymidine incorporation assay was performed as follows: murine CFU-E or BaF3-EpoR cells were plated in 96-well plates. After 4-h incubation, 1 μCi/well ³H-thymidine was added and cells were cultivated for respective hours. Cells were collected and the incorporated radioactivity was measured using a scintillation counter. To quantify the proliferation assay, regression lines were calculated with a four-parameter Hill regression ($y = y_0 + ax^b / (c^b + x^b)$). As the logarithmic transformation is a monotonic transformation, the sigmoidality of the curve is also true for a linear axis (Schilling *et al*, 2009).

For the propidium iodide (PI) staining, $2 \times 10^6$ cells were permeabilized with 70% ethanol at −20°C. Cells were washed with 0.3% BSA/PBS and incubated with ribonuclease reaction mixture at 23°C. After an additional washing step, fluorescence was measured by flow cytometry using a FACSCalibur (Becton Dickinson). Data analysis was performed with the MultiCycle (Phoenix Flow Systems) software.

## Mathematical modeling

Quantitative dynamic modeling was performed in MATLAB (Mathworks) using the D2D software package from http://www.data2dynamics.org (Raue *et al*, 2015). For parameter estimation, a deterministic derivative-based optimization with a multi-start strategy based on Latin hypercube sampling was applied (Raue *et al*, 2013). For further details on mathematical modeling, see Appendix F.

As an error model for experimental data, 10% relative error plus 5% absolute error of the highest data point under this condition were assumed.

The relative sensitivity $S_p^X$ shows the change in variable $X$ with infinitesimal small changes in parameter $p$, scaled by the respective values:

$$S_p^X = \frac{\partial X}{\partial p} \frac{p}{X}$$

As variable, we used pS6 integrated for 1 h: $\int_{t=0}^{t=60\ \mathrm{min}} pS6(t)\mathrm{d}t$ Parameters were protein abundance of pathway components (Table 1).

Linear regression analysis and statistical testing was performed with R (http://www.r-project.org). Linear regression model selection was based on Akaike's information criterion (Burnham & Anderson, 2002). For further details on linear regression analyses, see Appendix N and O.

## Data availability

The mass spectrometry proteomics data have been deposited to the ProteomeXchange Consortium via the PRIDE (Vizcaíno *et al*, 2016) partner repository with the dataset identifier PXD004816. The expression data were deposited in the GEO database under accession number http://tinyurl.com/GSE72317. The raw data of all qualitative and quantitative immunoblots have been provided as Source Data for the respective figures.

The models are provided in SBML code as Model EV1 (BaF3-EpoR), Model EV2 (mCFU-E), and Model EV3 (32D-EpoR). Curated model files have also been made available at JWS online: https://jjj.bio.vu.nl/models/adlung1/ (BaF3-EpoR model), https://jjj.bio.vu.nl/models/adlung2/ (mCFU-E model), and https://jjj.bio.vu.nl/models/adlung3/ (32D-EpoR model).

**Expanded View** for this article is available online.

## Acknowledgements

The authors thank Angela Lenze for autoMACS sorting of CD34[+] cells, Verena Lang for help with the imaging flow cytometry, Marvin Wäsch, Nora Schuhmacher, and Klara Zwadlow for excellent technical assistance, and Melania Barile, Helge Hass, and Bernhard Steiert for fruitful discussions about the mathematical model, as well as Aurelio Teleman and Kathrin Thedieck for their advice on the role of mTOR regulation. This work was supported by the SBCancer Network in the Helmholtz Alliance on Systems Biology as well as by the German Federal Ministry of Education and Research (BMBF)-funded CancerSys network LungSysII, the e:Bio network SBEpo, by the Helmholtz International Graduate School for Cancer Research at the German Cancer Research Center (DKFZ), the German Center for Lung Research (DZL) and the German Cancer Consortium (DKTK).

## Author contributions

LA conducted all experiments with human CFU-E cells, flow cytometry, and cell doubling experiments. LA and SK developed the mathematical model together with JT, MS, and TH. M-CW conducted inhibitor experiments, proliferation dose–response experiments and cell cycle indicator and overexpression experiments. BS generated all the other experimental data for model calibration. SC performed mass spectrometric measurements of human CFU-E cells. SL assisted proliferation assays. JB, MB, and HB analyzed microarray data. PW and ADH provided the human material. LA, SK, M-CW, BS, MS, PW, ADH, JT, TH, and UK designed the project. LA, M-CW, TH, and UK wrote the manuscript with comments from SK, MS, and JT. All authors approved the paper.

## Conflict of interest

The authors declare that they have no conflict of interest.

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
