## [Review Process File · Molecular Systems Biology]

Protein abundance of AKT and ERK pathway components governs cell type-specific regulation of proliferation

Lorenz Adlung, Sandip Kar, Marie-Christine Wagner, Bin She, Sajib Chakraborty, Jie Bao, Susen Lattermann, Melanie Boerries, Hauke Busch, Patrick Wuchter, Anthony D. Ho, Jens Timmers, Marcel Schilling, Thomas Höfer, Ursula Klingmüller

Corresponding author: Ursula Klingmueller & Thomas Höfer, German Cancer Research Center DKFZ

Review timeline:

First Submission:	27 January 2016
Editorial Decision:	26 February 2016
Second Submission:	19 August 2016
Editorial Decision:	28 September 2016
Revision received:	25 October 2016
Editorial Decision:	11 November 2016
Revision received:	21 November 2016
Accepted:	06 December 2016

Editor: Maria Polychronidou

Transaction Report:

1st Editorial Decision

26 February 2016

Thank you again for submitting your work to Molecular Systems Biology. We have now heard back from the three referees who agreed to evaluate your manuscript. As you will see below, the reviewers raise substantial concerns on your work, which, I am afraid to say, preclude its publication in Molecular Systems Biology.

Referee #1 is an expert in cell biology and cell signaling, referee #2's primary expertise is modeling and reviewer #3 is an expert in cell cycle. The reviewers appreciate that analyzing cell-type specific responses is an interesting topic. While reviewers #2 and #3 are cautiously positive, reviewer #1 (cell signaling expert) raises substantial concerns regarding the biological significance of the main findings. In particular, s/he mentions that training the model on two cells lines (one endogenously and one exogenously expressing EpoR) and validating it using data from a third cell line (exogenously expressing EpoR), does not provide convincing support for the predictive value of protein abundance patterns. While the presented results seem encouraging, further analyses i.e. using a wider panel of cell lines would be required to convincingly support the main conclusions of the study and their broader relevance. Moreover, this reviewer raises several methodological issues related to the robustness and quantification of the data and mentions a number of figure panels (i.e. Figures 1, S9, S14) that seem to have been assembled from disparate images. Addressing these rather substantial concerns raised by reviewer #1 would require extensive further experimentation

and analyses with unclear outcome that would go beyond the scope of a major revision.

Under these circumstances and considering the overall rather low level of support, I see no other choice than to return the manuscript with the message that we cannot offer to publish it. In any case, thank you for the opportunity to examine your work. I hope that the points raised in the reports will prove useful to you and that you will not be discouraged from submitting future work to Molecular Systems Biology.

REFEREE REPORTS

Reviewer #1:

Adlung et al.
Mol. Syst. Biol.
Manuscript # MSB 16-6819

This study by Adlung et al. aims to use quantitative methods to understand the role of protein abundance in the AKT and ERK pathways on proliferation response during Epo treatment. The authors use cell-line specific ODE models to suggest that differences in protein abundance are sufficient to explain the divergence in response between cell lines of distinct lineages. Rate parameters of the model are trained on two of the three cell lines (CFU-E and BaF3-EpoR) and then testing the models fit to experimental data on the third cell line (32D-EpoR) without changing any parameters and only adding the protein abundance values, calculated by quantitative immunoblotting, of the third cell line. The authors observe good agreement between the model and experimental data and then hypothesize that protein abundance will dictate the response to specific AKT or ERK inhibitors. The ODE models predict that S6 activation sensitivity is determined by protein abundance of the ERK and AKT pathway members. Experimentally, S6 activation is not perturbed in the CFU-E cell line when an ERK inhibitor is added, but S6 phosphorylation is disrupted when AKT inhibitor is added. The opposite is true for the other two cell lines, 32D-EpoR and BaF3-EpoR. Cell line specific behavior is also observed with a set of cell cycle regulators agglomerated as a "cell cycle indicator". The conclusion is that cell type specific protein abundance patterns determine the signal flow through the ERK and AKT pathways in response to Epo treatment and that the downstream effectors of these pathways are distinct in their impact on proliferation.

The premise of this paper - that protein abundance leads to differential responses in a cell-type specific manner - follows from previous work showing that growth factor response of genetically diverse cancer cell lines can be predicted by receptor abundance (PMID: 24655548) and that protein abundance yields context specific responses in the ErbB signaling network (PMID: 24345680). The hypothesis is interesting, but the study design is not ideal for answering the overarching question of cell specificity. The cell lines they use are vastly different from each other in more than just cell type: some overexpress the receptor, some are transformed, etc. Further, the quantitative emphasis of the manuscript is not supported by the data shown; for example, many of the western blots are low quality, inconsistent, and altered such that quantification from them would be unreliable. The authors need to correct their cell system with more consistent cell sources as well as take better care with their quantification methods that form the crux of the paper.

Major Points

1. While it is true that different cell types will not share exact protein abundances, the main breakdown of the model system presented here is that the cells themselves come from different sources and the 32D and BaF3 cell lines do not endogenously express the Epo receptor (EpoR). The fact that the CFU-E cell line is a primary line from mouse while the other two are established cell lines is problematic, because the cell lines may differ simply because of their transformation. Furthermore, the 32D and BaF3 established cell lines do not endogenously express EpoR, so the authors overexpress the receptor. This presents a problem for multiple reasons. First, the exogenously expressed EpoR is expressed at much higher abundance than the endogenously expressed receptor on CFU-E cells, creating an apples-to-oranges comparison for a study where protein abundance is proposed to be critical. In fact, all the cell type specific responses they claim to find can be alternatively explained as differences between cells that express and do not express

EpoR with the appropriate downstream signaling "circuitry".

Second, when the authors stimulate with Epo, they do not stimulate with stoichiometrically consistent concentrations for each cell line. This would be passable if they demonstrated that the amount they were stimulating with was at saturation. The study would have been on much firmer footing if it had used either all primary cell lines from isogenic mice or all established cell lines. The cell lineages should be either endogenously expressing EpoR at the same abundance or all exogenously expressing EpoR at the same abundance.

2. Considering the extreme precision required for the bulk of this work, it is troubling that the immunoblot data shown does not convey a careful approach. There are several examples of low quality western blots throughout the manuscript ranging from general 'waviness' to improper image exposure (Figure S9 panel A, Raf-RDB). These problems alone call into question the quantitative nature of these data, but more worrying is that there is evidence of improper image manipulation. In figure S14, panel A the BaF3-EpoR AKT over expression conditions are very clearly pasted in from another image. Additionally, in figure S9 panel A, top half of BaF3-EpoR; there are 19 bands for GTP-Ras, and 20 for Raf-RDB. In main figure 1, panel D it appears that the blot data does not visually match the quantification given in panel E. Panel E suggests that the largest fold change in pEpoR is ~2 while the blot data suggests a much higher fold change of at least 5; while it is less obvious, there also appears a strange edge at the boundary between the 0 and 10 minute conditions BaF3-EpoR pEpoR lanes. If the 10 and 30 minute section of the blot was altered independent of the rest of the image, this could account for the crisp edge as well as the apparent miss-match of image data shown and quantification provided. The quantitative inconsistencies, image errors, and image manipulation together are sufficient to exclude the manuscript from further consideration.

Minor Points

1. The cell cycle indicator formula is not justified. This formula seems arbitrary and would either need to be explained or the authors would need to show that different versions of the cell cycle indicator formula yield the same results. Alternatively the authors can use other direct experimental readouts of proliferation such as BrdU incorporation.
2. Statistics need to be done on much of the data to support the authors' claims that certain conditions are different or the same (for example: Figure 5C).
3. Many of the conclusions in this paper are either overstated based on the data presented or obvious. For example, many model predictions are presented as findings before there are experimental tests of those predictions.
4. Experiments done in Figure 4 should also be done for the other cell lines.
5. Loading controls in blots often show uneven loading of samples making quantification difficult.
6. Figure S12. EpoR-GST runs smaller than EpoR, which under reducing blot conditions does not make sense as GST should add ~25kDa to the EpoR protein.
7. Figure S25. The upper middle panel looks like a reproduction of the top panel.

Reviewer #2:

This manuscript from Höfer, Klingmüller and co-workers (with four co-first authors, Adlung, Kar, Wagner, and She) reports findings from a modeling-experimental study, in which three cell type-specific mathematical models were developed and analyzed with the goal of better understanding erythropoietin (Epo) receptor (EpoR) signaling. Two major information transmission pathways leading to activation of proliferation (Raf/Mek/Erk and PI3K/Akt) were considered, with coarse representation of molecular mechanisms, in each of the models. The three models, which each consist of ordinary differential equations (ODEs), have the same structure and identical rate constants but differ with respect to total protein abundances. The authors estimated total abundances of key signaling proteins in the different cell types of interest on the basis of quantitative immunoblotting data and other considerations, including assumptions. Model parameter settings were adjusted to reproduce time-course measurements of phosphorylation (of EpoR, Erk, Akt, and S6). Inputs and perturbations considered in the study included different doses of Epo, kinase inhibitor treatments (affecting Akt and Mek, for example), and overexpression. The cell types considered were hematopoietic in origin: primary CFU-E cells, belonging to the myeloid-erythroid branch, from E13.5 Balb/c mice; a murine IL-3-dependent pro-B cell line (BaF3); and a murine IL-3-independent myeloid cell line (32D). These cell types all express EpoR and respond differently to

Epo stimulation; they are commonly used to study EpoR signaling. The authors find that differences in protein abundances across the three cell types largely explain observed differences in their proliferative responses to Epo stimulation. The authors conclude that CFU-E cells activate proliferation primarily through the PI3K/Akt axis, BaF3 cells activate proliferation mainly through the Raf/Mek/Erk axis, and 32D cells use both axes of information flow. These conclusions are based on analysis of the models, including a sensitivity analysis, and experimental tests of model predictions.

The study has some weaknesses. Absolute protein abundances were not comprehensively quantified, which is possible with mass spectrometry-based proteomics (at least for the cell lines). See Hein et al. (2015) [Cell 163:712-723] and Kulak et al. (2014) [Nat Methods 11:319-324]. Models were parameterized on the basis of population-averaged measurements (vs. single-cell measurements). Gene expression, as determined through microarray analysis, was equated with protein expression, which is dubious. Time-series data are correlated, so the data used for calibration of parameter values are not as informative as one might expect from a simple count of data points. The model has only coarse mechanistic resolution, which is intentional and somewhat justified, but there was not a rigorous reduction of model resolution from fine to coarse, which leaves room for doubt about the validity of the model structure, which is a common concern about biological models. Annotation/justification of the model structure is minimal. The D2D software package and the model analyses performed with it rely on commercial software (MATLAB), which may be a hindrance to some researchers desiring to repeat the analyses. There are no easy ways to address these weaknesses and they are offset by considerable strengths. Overall the study is impressive. My only advice is that the authors might want to consider adding more cautionary discussion of limitations, which could be helpful for some readers, although the discussion of limitations is acceptable in current form.

An especially innovative aspect of this study is the integration of a mechanistic model with a regression model, which is impressive. I expect to see more examples of this type of approach in the future.

The datasets generated in this study and used in modeling may be valuable to other researchers, especially modelers. I would encourage the authors to provide the data in as many different commonly used formats as possible, to make the data as easily accessible as possible.

The authors should acknowledge earlier related modeling work, where cell line-specific measurements of protein abundances were used to obtain cell line-specific models [Stites EC et al. (2015) Biophys J 108:1819-1829] and where patient-specific data were used to obtain patient-specific models [Fey D et al. (2015) Sci Signal 8: ra130].

I think this manuscript is significant because it sheds light on the importance of knowing the characteristic protein abundance profile of a cell type when one is attempting to predict cellular responses to signals and perturbations.

Reviewer #3:

In the presented paper by Adlung et al., the authors quantitatively measure the role of multiple components of the signaling network on cell growth, cell-cycle progression and proliferation in three cell lines of hematopoietic origin.

By measuring the dynamics of numerous signaling proteins they demonstrate that different relative inputs of two cell proliferation signaling pathways (ERK and AKT) are different in different cell lines and can be explained by the difference in signaling protein abundances. By measuring these profiles as well as responses to various stimuli in two cell lines the authors fit a mathematical model that was then able to predict the response of the third cell line to the stimuli based solely on the protein abundance profile measurement in that line. This part of the work is interesting and pretty convincing.

In the second part of the paper the authors claim that two different processes can play a dominant

role in cell proliferation control in different cell types: protein synthesis is rate-limiting for faster cycling cells, while G1-S control is limiting for slower cycling cells. Although this idea seems logical, I think the authors need to provide some stronger data to support this claim. All in all, when revised, this work will be suitable for publication in *Molecular Systems Biology*.

Major points:

1) The "cell-cycle indicator" chosen by the authors, which combines the transcription levels of three cell cycle genes (*cycD2*, *cycG2*, and *p27*), needs a more thorough validation to confirm that it actually reflects the rate of the cell cycle progression. Current justification that these genes demonstrate a strong expression change at 3 hours after EpoR stimulation and therefore can be used to estimate the G1-S transition rate is not sufficient. Comparison with other combinations of known cell cycle genes (such as *e2f1*, *cycE*, ...) would be more informative, as well as some measurements of these parameters in the cells cycling with a different rate (e.g., stimulated with different concentrations of growth factors, or treated with inhibitors).

2) The authors use their mathematical model that was fit based on the data from the two cell lines (CFU-E and BaF3-EpoR) to predict the response of the third cell line (32D-EpoR). However, it is not clear how good their prediction is. E.g., in Fig 4A it would be nice to show besides 32D-EpoR predictions and actual measurements, also the responses of the CFU-E and BaF3-EpoR cells to the same stimulus. This would demonstrate how close the prediction for 32D-EpoR is to the experimental data, compared to the difference between different cell lines. Given how similar the protein expression profiles are between BaF3-EpoR and 32D-EpoR (Table 1), one might expect that the 32D-EpoR response curve could be well approximated simply by the BaF3-EpoR curve. So, the authors at least need to show that their model, which integrates the data from both CFU-E and BaF3-EpoR cell lines, gives a better prediction than BaF3-EpoR data alone.

3) Authors use the pS6 levels and the cell-cycle indicator measurements to distinguish whether it is the cell growth or the levels of G1-S regulators that is most critical for determining the proliferation rate for different cell lines. However, these two parameters (growth rate and G1-S transition) are not independent as cell size also feeds into the G1-S regulatory network. To show that these parameters can be separated and considered as at least quasi-independent in the context of this study, the authors should test the correlation between the two (e.g., at different levels of Epo stimulation and pathway inhibitors).

4) The authors make a strong statement that while slower cycling cells (BaF3-EpoR, 32D-EpoR) are controlled at the level of G1-S progression, faster cycling cells (CFU-E) are controlled by protein synthesis rates. In my opinion, this conclusion is not supported strongly by the data provided. Fig. 6C shows that 2.5-fold change in pS6 levels causes only a ~10% change in proliferation rate in CFU-E cells. The cell-cycle indicator has approximately the same small effect on proliferation for this cell line (Fig. S24). And also, as I already mentioned, the cell-cycle indicator needs some further validation.

5) In the last part of the results section the authors show that their mathematical model predicts the combined effect of MEK and AKT pathway inhibitors on the proliferation of the studied cells. The result looks nice but might be trivial: according to Fig. 6C, U0126 does not have any effect on CFU-E and 32D-EpoR cell lines and only decreases the proliferation rate in BaF3-EpoR cells, while AKT VIII has some affect in all three lines. So, predicting the combined effect of the two drugs one of which does not have an effect on its own looks trivial (CFU-E, 32D-EpoR). So, the only interesting prediction is a cooperative effect of the two inhibitors on BaF3-EpoR. This result would look stronger if the authors could demonstrate an opposite example finding two inhibitors, each of which would affect the cell proliferation rate but together they would not show a combined effect.

Minor points:

1) In Fig 1C, the legend mentions cells "stained with Calcein (upper right panel)" that are actually not shown on the figure. Also the legend says that the cell diameter was measured by bright-field microscopy, while in the Results section is said that confocal microscopy was used for this. Was the cytoplasmic volume used for the calculation of protein concentrations measured including or

excluding the nucleus? The data from the table in Fig. 1C is already shown in Table 1, so there is no need to duplicate.

2) There is an inconsistency in what concentration of Epo is used for saturating stimulation. In the beginning of the paper it is 50 U/ml, then it becomes 5 U/ml, then for microarray 1 U/ml was used. Please comment on this.

3) On p.12 the authors conclude that "CFU-E cells signaling to S6 primarily through the AKT axis, BaF3-EpoR cells primarily through the ERK pathway and 32D-EpoR cells using both pathways", but earlier in the same section (Fig. 4B) they have shown that actually 32D-EpoR are very similar to BaF3-EpoR in that influence of the Ras/MEK/ERK cascade has a dominant effect on pS6 with a only a slight impact of AKT pathway.

4) On p.16 the authors compare the experimental and predicted effects of MEK and AKT pathway inhibitors on the cell-cycle indicator levels (Fig. 6B, S17) and conclude that "effects of the inhibitor treatment on the cell-cycle indicator are explained very well by changes in the integrated pAKT and ppERK responses". Please comment on why the model always underestimates the effect of the inhibitors (most obvious - in case of U0126 in 32D-EpoR).

5) When describing the results of the regression analysis of whether the pS6 levels or the cell-cycle indicator better predict the proliferation rate in each cell line (p.17), it's worth mentioning explicitly that the combination of the two was also tested and found less predictive than the best one alone. Does this suggest that one of these two parameters already contains the information about another one (so, they are not independent)? How was the combination of the two parameters exactly calculated?

6) When predicting the combined effect of the two inhibitors on cell proliferation, the authors say that "For CFU-E and 32D-EpoR cells, AKT inhibition was predicted to control cell proliferation in a dose-dependent manner, without combined effect with ERK inhibition" (p. 18). However, actually from Fig. 7B for 32D-EpoR it seems like these the two inhibitors are predicted to have a combined effect, although the U0126 effect is relatively small.

Second submission

19 August 2016

Report continues on next page.

Reviewer #1:

Adlung et al.
Mol. Syst. Biol.
Manuscript # MSB 16-6819

This study by Adlung et al. aims to use quantitative methods to understand the role of protein abundance in the AKT and ERK pathways on proliferation response during Epo treatment. The authors use cell-line specific ODE models to suggest that differences in protein abundance are sufficient to explain the divergence in response between cell lines of distinct lineages. Rate parameters of the model are trained on two of the three cell lines (CFU-E and BaF3-EpoR) and then testing the models fit to experimental data on the third cell line (32D-EpoR) without changing any parameters and only adding the protein abundance values, calculated by quantitative immunoblotting, of the third cell line. The authors observe good agreement between the model and experimental data and then hypothesize that protein abundance will dictate the response to specific AKT or ERK inhibitors. The ODE models predict that S6 activation sensitivity is determined by protein abundance of the ERK and AKT pathway members. Experimentally, S6 activation is not perturbed in the CFU-E cell line when an ERK inhibitor is added, but S6 phosphorylation is disrupted when AKT inhibitor is added. The opposite is true for the other two cell lines, 32D-EpoR and BaF3-EpoR. Cell line specific behavior is also observed with a set of cell cycle regulators agglomerated as a "cell cycle indicator". The conclusion is that cell type specific protein abundance patterns determine the signal flow through the ERK and AKT pathways in response to Epo treatment and that the downstream effectors of these pathways are distinct in their impact on proliferation. The premise of this paper - that protein abundance leads to differential responses in a cell-type specific manner - follows from previous work showing that growth factor response of genetically diverse cancer cell lines can be predicted by receptor abundance (PMID: 24655548) and that protein abundance yields context specific responses in the ErbB signaling network (PMID: 24345680). The hypothesis is interesting, but the study design is not ideal for answering the overarching question of cell specificity. The cell lines they use are vastly different from each other in more than just cell type: some overexpress the receptor, some are transformed, etc. Further, the quantitative emphasis of the manuscript is not supported by the data shown; for example, many of the western blots are low quality, inconsistent, and altered such that quantification from them would be unreliable. The authors need to correct their cell system with more consistent cell sources as well as take better care with their quantification methods that form the crux of the paper.

We take the concerns raised by the reviewer very seriously and would like to address them in the following.

Major Points

1. While it is true that different cell types will not share exact protein abundances, the main breakdown of the model system presented here is that the cells themselves come from different sources and the 32D and BaF3 cell lines do not endogenously express the Epo receptor (EpoR). The fact that the CFU-E cell line is a primary line from mouse while the other two are established cell lines is problematic, because the cell lines may differ simply because of their transformation. Furthermore, the 32D and BaF3 established cell lines do not endogenously express EpoR, so the authors overexpress the receptor. This presents a problem for multiple reasons. First, the exogenously expressed EpoR is expressed at much higher abundance than the endogenously expressed receptor on CFU-E cells, creating an apples-to-oranges comparison for a study where protein abundance is proposed to be critical. In fact, all the cell type specific responses they claim to find can be alternatively explained as differences between cells that express and do not express EpoR with the appropriate downstream signaling "circuitry".

We entirely agree with the reviewer that this needed clarification.

To better introduce the cell systems we used, we now included a new paragraph in the introduction that summarizes key properties of the cellular systems (page 4,5):

"The ample knowledge on molecular mechanisms contributing to the regulation of erythropoiesis has been facilitated, on the one hand, by the availability of factor-dependent, immortalized hematopoietic cell lines from mice. For example the interleukin (IL)3-dependent cell lines BaF3 of lymphoid origin (Palacios & Steinmetz, 1985) and 32D of myeloid origin (Greenberger *et al*, 1983) have been utilized for decades to unravel structure-function relationship of cytokine receptors such as the EpoR (Klingmüller *et al*, 1996; Wang *et al*, 1993). Exogenous expression of the EpoR renders these cell lines responsive to Epo and enables proliferation in the presence of Epo (D'Andrea *et al*, 1989). Due to their growth properties, BaF3 cells are currently widely used in kinase drug discovery and represent a reliable cellular system to access kinase activity (Jiang *et al*, 2005; Moraga *et al*, 2015). On the other hand, primary erythroid progenitor cells from mice (mCFUE) are readily available from fetal liver or bone marrow, and methods for their cultivation have been established (Landschulz *et al*, 1989; Rich & Kubanek, 1976). For the human system, a protocol has been devised (Broudy *et al*, 1991; Miharada *et al*, 2006) to expand and differentiate human erythroid progenitor (hCFU-E) cells from CD34⁺ cells mobilized into the peripheral blood of healthy donors. With this strategy sufficient material of hCFU-E can be obtained to confirm in functional studies the clinical relevance of observations."

We calibrated the mathematical based on data obtained with murine erythroid progenitor cells at the colony-forming unit-erythroid stage (mCFU-E) and with the murine factor-dependent, immortalized cell line BaF3 exogenously expressing the erythropoietin receptor (BaF3-EpoR). Since the primary material is very limited, a systematic, quantitative comparison to responses in an Epo-responsive cell line is of value. For example to obtain sufficient material of mCFU-E to perform immunoprecipitation coupled with immunoblotting analysis at 14 time points, we need to breed 120 mice resulting in 20 mice with timed pregnancy from which we can extract about 100 fetal livers to purify 7×10^7 mCFU-E cells. To improve clarity, we modified the first paragraph of the results section (page 5,6):

"To quantitatively assess Epo-induced proliferative responses in murine primary erythroid progenitor cells at the colony-forming unit-erythroid stage (mCFU-E) and in the immortalized murine cell line BaF3 exogenously expressing the EpoR (BaF3-EpoR), we incubated the cells in the presence of different Epo doses and measured as a readout for proliferation DNA synthesis by thymidine incorporation."

In the previous version of the manuscript we successfully applied the mathematical model to predict based on determinations of protein abundance the impact of inhibitors on the dynamics of Epo-induced signalling and on proliferation in another factor-dependent, immortalized cell line from mice, 32D cells, in which we exogenously expressed the EpoR by the same expression vector as it was utilized for BaF3-EpoR cells.

To further strengthen our conclusions, we now performed new experiments with human erythroid progenitor cells (hCFU-E) differentiated from CD34⁺ cells from three healthy donors. We determined by quantitative mass spectrometry in combination with the "proteomic ruler" method the abundance of the selected signalling proteins, and showed that the abundance of the selected proteins is highly conserved in hCFU-E cells of the three donors yet surprisingly distinct from the protein abundance identified for mCFU-E cells and the murine immortalized hematopoietic cell lines. Based on these determinations the mathematical model predicted that, in contrast to mCFU-E cells, hCFU-E cells require co-treatment with the MEK and AKT inhibitor to efficiently inhibit Epo-stimulated proliferation. We experimentally verified this model prediction.

These new results are now presented in the new Fig. 8 and the text in the result section was adapted accordingly (page 21, 22):

"To highlight that the protein abundance governs cell-type-specific regulation of Epo-induced proliferation and as a consequence the sensitivity towards inhibitors, we prepared human CFU-E cells from CD34⁺ cells mobilized into the peripheral blood of three healthy donors. By means of mass spectrometry, we quantified 6,925 proteins of which 5,912 proteins were shared among the hCFU-E cells from the three independent donors (Fig. 8A).

Next, we applied the "proteomic ruler" method (Wiśniewski et al, 2014) to calculate the copy numbers of individual proteins per cell. To convert these numbers into cytoplasmic concentrations, we measured the average cell size by imaging flow cytometry (Fig. 8B) and calculated the cytoplasmic volume from these data. The average cytoplasmic volume of the hCFU-E of the three donors was comparable but considerably larger than the volume of the mCFU-E cells (Fig. 8C). To relate the results obtained with the proteomic ruler method to the previous measurements in mCFU-E, BaF3-EpoR and 32D-EpoR utilizing quantitative immunoblotting and recombinant protein standards, we performed additional mass spectrometric measurements for mCFU-E and BaF3-EpoR cells. As shown in Fig. 8D, for the key signaling components in the two cell types, the number of molecules per cell determined with the different techniques showed a good correlation ($R^2=0.82$), validating our determinations by quantitative immunoblotting and confirming that the snapshot measurement by mass spectrometry yielded reliable results. Surprisingly, the protein abundance of the key signaling proteins determined for hCFU-E cells showed very low variance between the three independent donors but were highly distinct from the values obtained for mCFU-E cells as well as for the murine cell lines BaF3-EpoR and 32D-EpoR (Fig. 8E). For example Ras, Raf and AKT were present at much lower levels in hCFU-E compared to mCFU-E, BaF3-EpoR and 32-EpoR cells, whereas for PI3 kinase elevated levels were observed in hCFU-E cells. As expected, the levels of the EpoR were comparable in hCFU-E and mCFU-E and elevated to the same extent in Ba3-EpoR and 32D-EpoR cells. To link the signaling layer to proliferation, we applied our concept that larger (see Fig. 8B) and more slowly dividing (Fig. EV30) cells regulate proliferation primarily by the control of G1-S progression (Expanded View information R). Since the measured protein abundance of the key signaling proteins was highly comparable in the hCFU-E cells of the three donors, we used the average concentrations of these proteins to predict with our mathematical model the impact of the AKT inhibitor and MEK inhibitor on Epo-supported cell proliferation of hCFU-E cells. Without any further information (Expanded View information T) the mathematical model predicted that hCFU-E cells are less sensitive to the individual inhibitors but, distinct from mCFU-E cells that are primarily sensitive to the AKT inhibitor, rather require treatment with the combination of both, AKT VIII and U0126, to effectively inhibit Epo-induced proliferation (Fig. 8F). To validate this model prediction experimentally, we quantified the numbers of hCFU-E cells after 96 hours of stimulation with 5 U/ml Epo and single or combined treatment with AKT VIII and U0126. Since hCFU-E cells from donor 3 showed the most robust proliferative response (Fig. EV31), we analyzed proliferation of hCFU-E cells from this donor in biological triplicates (Fig. 8F). Qualitatively the same effects on proliferation were observed for the hCFU-E cells from donor 1 and donor 2 (Fig. EV32). As shown in Fig. 8E, in line with the model prediction, Epo-induced proliferation of hCFU-E was much more sensitive towards combinatorial treatment with AKT VIII and U0126 compared to the treatment with the individual inhibitors.

These findings showcase that protein abundance can be reliably measured from snapshot data of human material. Based on these data, our integrative mathematical model allows to evaluate combinatorial application of inhibitors in silico and thus may serve to improve the treatment of proliferative disorders such as tumors driven by exacerbated growth-factor signaling."

Second, when the authors stimulate with Epo, they do not stimulate with stoichiometrically consistent concentrations for each cell line. This would be passable if they demonstrated that the amount they were stimulating with was at saturation.

To address the issue raised by the reviewer, we calculated for the dose-response experiment in Fig. 1A the stoichiometric concentrations (U/EpoR). The results shown in the new Expanded View information section A demonstrate that from 2.5 U/ml Epo in mCFU-E cells and from 5 U/ml Epo in BaF3-EpoR cells onwards the Epo-induced proliferative response is saturated. As evidenced in Fig. EV13 the activation of signal transduction at 50 U/ml is saturating in mCFU-E and BaF3-EpoR cells. The text in the results section was modified accordingly (page 6):

“In a first step the protein abundance and dynamics of phosphorylation of EpoR, AKT and ERK in response to stimulation with a saturating Epo concentration (50 U/ml, see Expanded View information A, Fig. EV1, Fig. EV13) was qualitatively assessed.”

The study would have been on much firmer footing if it had used either all primary cell lines from isogenic mice or all established cell lines. The cell lineages should be either endogenously expressing EpoR at the same abundance or all exogenously expressing EpoR at the same abundance.

As detailed above we now use in our study primary erythroid progenitor cells from mice and humans. Studies with knockout mice of the hormone Epo and of the EpoR revealed that the system is non-redundant and absolutely essential for the formation of mature erythrocytes. During erythropoiesis, cells at the colony-forming unit-erythroid stage (CFU-E) express the highest level of the EpoR and are most responsive to Epo. Since primary material is scarce, it is valuable to study molecular mechanisms in factor-dependent, immortalized cell lines. The most widely used cell lines to study structure-function relationship of the EpoR are the BaF3 and 32D cell lines. Upon exogenous expression of the EpoR these cells become responsive to Epo and are capable to proliferate in the presence of Epo. As described in the method section, we utilized the same retroviral vector to stably express the EpoR in BaF3 and 32D cells. The new quantitative mass spectrometric analysis reveals that indeed mCFU-E and hCFU-E cells show comparable levels of the EpoR and both, BaF3-EpoR and 32D-EpoR cells, also show comparably elevated levels of the EpoR. Despite these different levels of the EpoR in the primary cells and the cell lines, our integrative model correctly predicts based on the protein abundance of the key signalling molecules that mCFU-E and 32D-EpoR cells are particularly sensitive towards the AKT inhibitor, whereas in hCFU-E and BaF3-EpoR cells Epo-induced proliferation is most effectively inhibited by a co-treatment with both inhibitors.

This is now included in the revised text of the result part (page 21):

“As expected, the levels of the EpoR were comparable in hCFU-E and mCFU-E and elevated to the same extent in Ba3-EpoR and 32D-EpoR cells.”

2. Considering the extreme precision required for the bulk of this work, it is troubling that the immunoblot data shown does not convey a careful approach. There are several examples of low quality western blots throughout the manuscript ranging from general 'waviness' to improper image exposure (Figure S9 panel A, Raf-RDB). These problems alone call into question the quantitative nature of these data, but more worrying is that there is evidence of improper image manipulation. In figure S14, panel A the BaF3-EpoR AKT over expression conditions are very clearly pasted in from another image. Additionally, in figure S9 panel A, top half of BaF3-EpoR; there are 19 bands for GTP-Ras, and 20 for Raf-RDB. In main figure 1, panel D it appears that the blot data does not visually match the quantification given in panel E. Panel E suggests that the largest fold change in pEpoR is ~2 while the blot data suggests a much higher fold change of at least 5; while it is less obvious, there also appears a strange edge at the boundary between the 0 and 10 minute conditions BaF3-EpoR pEpoR lanes. If the 10 and 30 minute section of the blot was altered independent of the rest of the image, this could account for the crisp edge as well as the apparent miss-match of image data shown and quantification provided. The quantitative inconsistencies, image errors, and image manipulation together are sufficient to exclude the manuscript from further consideration.

We took the criticism very serious and collected all original immunoblots utilized in the study in a structured file accessible at the following link for detailed inspection:

<http://bit.ly/1RPddTD>.

The immunoblots in the subfolder "All_JPEG" are displayed with the same exposure and contrast as used for the figures and align to the original raw data in the subfolder "All_TIFF". Of these data 20 immunoblots are of qualitative nature to visualize effects. This was necessary since for the quantitative data generation for model calibration we used randomized sample loading according to our previously published workflow (Schilling et al.,

2005), which is difficult to inspect visually. Altogether 432 data points were generated from 123 quantitative immunoblots for the parameterization of the mathematical model. The distinction between qualitative data and quantitative data is now clearly indicated in the text (page 6,7):

“In a first step the protein abundance and dynamics of phosphorylation of EpoR, AKT and ERK in response to stimulation with a saturating Epo concentration (50 U/ml, see Expanded View information A, Fig. EV1, Fig. EV13) was qualitatively assessed.”

“To quantitatively examine the dynamics of Epo-induced signal transduction in mCFU-E and BAF3-EpoR cells, we used randomized sample loading in combination with quantitative immunoblotting to determine in a time-resolved manner the phosphorylation of EpoR, AKT and ERK in both cell types.”

Revised EV11 (Figure S9 panel A, Raf-RDB)

We thank the reviewer for spotting the error and apologize that we by accident swapped reprobes of two experiments. This mistake has been corrected in the revised Fig. EV11.

Revised EV17 (S14, panel A the BaF3-EpoR AKT)

The samples for overexpression of AKT, Ras or PTEN in BaF3-EpoR cells were analysed on the same SDS-PAGE. However, since the order was different compared to mCFU-E cells, we aimed to ease comparison and re-arranged in the display the order. We corrected this mistake in the revised Fig. EV17 and show the samples as analyzed. Again the entire blots can be inspected using the link <http://bit.ly/1RPddTD>.

As now emphasized in the text, these data served merely as qualitative support of the sensitivity analysis and were not used for model calibration (page 16):

“In further agreement with the sensitivity analysis, the observed experimental overexpression of constitutively active Ras qualitatively resulted in a stronger elevation of S6 phosphorylation in BaF3-EpoR than in mCFU-E cells whereas overexpression of AKT had a qualitative stronger effect in mCFU-E cells, and overexpression of PTEN qualitatively diminished S6 phosphorylation more strongly in mCFU-E than in BaF3-EpoR cells (Fig. EV17).”

Figure 1, panel D

To demonstrate that the immunoblot was neither cut nor otherwise manipulated, we now show a larger section of the immunoblot and the entire image with the same exposure time and contrast can be viewed using the link <http://bit.ly/1RPddTD>.

As indicated above and now explicitly stated in the text of the results section, in a first step we qualitatively assessed the dynamics of Epo-induced signal transduction in mCFU-E cells and BaF3-EpoR cells. For these qualitative studies as indicated in the figure legend equal amounts of cells (5×10^6) were analyzed. Since BaF3-EpoR cells are larger than mCFU-E cells and express higher EpoR levels, a much stronger EpoR signal is to be expected. However, in the quantitative analysis shown in Fig. 1E absolute concentrations in nM were calculated and here the difference in cell volume is corrected for and therefore the difference in signal intensity is smaller.

Minor Points

1. The cell cycle indicator formula is not justified. This formula seems arbitrary and would either need to be explained or the authors would need to show that different versions of the cell cycle indicator formula yield the same results. Alternatively the authors can use other direct experimental readouts of proliferation such as BrdU incorporation.

We thank the reviewer for this suggestion. We followed the advice and performed PI staining experiments for mCFU-E and BaF3-EpoR cells to confirm the validity of the cell cycle

indicator as a coefficient to quantify Epo-induced G1-S progression. The results are now shown in a new Fig. EV5.

2. Statistics need to be done on much of the data to support the authors' claims that certain conditions are different or the same (for example: Figure 5C).

We thank the reviewer for this suggestion and revised Fig. 1F and Fig. 5C to include statistical tests.

To justify the cell-cycle indicator and to refer to the statistical tests, we modified the text in the result section (page 10):

"These results suggested that the quantification of the expression of *cyclinD2*, *cyclinG2* and *p27* might provide a good measure to compare Epo-induced cell-cycle progression in BaF3-EpoR and mCFU-E cells. To summarize the contribution of the cell-cycle activator and the two cell-cycle repressors that counteract each other in controlling cell-cycle progression, we defined a cell-cycle indicator as follows:

$$\frac{[\textit{cyclinD2}]}{\sqrt{[\textit{cyclinG2}] \times [\textit{p27}]}}$$

This coefficient reflects complex regulation of cell-cycle progression in response to Epo stimulation better than individual components (e.g. only *cyclinG2*) and was strongly increased after 3 hours of Epo addition in mCFU-E and BaF3-EpoR cells (Fig. 1F, right panel).

Notably, the cell-cycle indicator was significantly ($p=0.04$) higher in BaF3-EpoR cells (Fig. 1F, right panel), which is in agreement with the results of the propidium iodide staining experiment (Fig. EV5) showing that, whereas mCFU-E cells are already committed to cell-cycle progression, an increasing fraction of BaF3-EpoR cells enters S/G2/M phase in response to stimulation with increasing Epo doses."

3. Many of the conclusions in this paper are either overstated based on the data presented or obvious. For example, many model predictions are presented as findings before there are experimental tests of those predictions.

To address this point we carefully went through the manuscript and rephrased the wording of several paragraphs in the Expanded View Information and the main text (page 12,13, 15, 16):

"First, the lipid phosphatases SHIP1 and PTEN were overexpressed, and the impact on AKT activation was monitored (Fig. 3A). In mCFU-E cells, a strong effect of PTEN overexpression on Epo-induced AKT phosphorylation was experimentally observed, and a weaker effect of a similar overexpression of SHIP1, which were both captured by the model (Fig. 3A, Fig. EV14). Further, we observed that the Epo-induced induction pAKT in wild-type BaF3-EpoR cells was even lower than in mCFU-E cells with overexpressed PTEN (Fig. 3A), which is consistent with the high concentrations of SHIP1 and PTEN in BaF3-EpoR cells (Tab. 1). The mathematical model calibrated based on these data nevertheless predicted that overexpression of SHIP1 or PTEN would decrease AKT phosphorylation even further in these cells. Indeed in an independent experiment the Epo-induced dynamics of pAKT in BaF3-EpoR cells overexpressing SHIP1 or PTEN was in agreement with the model trajectories (Fig. 3B). Further, we predicted with the model and validated experimentally that simultaneous down-regulation of SHIP1 and PTEN to their respective concentrations in mCFU-E cells enhanced Epo-induced pAKT levels in BaF3-EpoR cells to the extent observed in mCFU-E cells (Fig. EV14).

Second, the DUSPs, a family of phosphatases that negatively regulate ERK signaling, were examined. The analysis of DUSP protein abundance is challenging because multiple isoforms with different functions exist and only very few antibodies, mostly with low specificity, are available. In our proteome-wide quantitative mass spectrometry analysis of

unstimulated mCFU-E, hCFU-E and BaF3-EpoR cells, Epo-regulated DUSP family members were below the detection limit. Therefore, we used the mRNA expression levels as proxy, assuming at least some correlation with protein expression. The mathematical model predicted a \log_2 -fold change of 5.27 higher basal expression of DUSP in BaF3-EpoR cells compared to mCFU-E cells (Fig. 3C). To experimentally validate this model prediction, we first identified by microarray analysis of mCFU-E cells (Bachmann et al, 2011) and BaF3-EpoR cells (Fig. EV15) DUSP4, DUSP5 and DUSP6 as family members that are differentially expressed in response to Epo stimulation. The analysis of the basal mRNA expression of these DUSP by quantitative RT-PCR showed that the \log_2 -fold difference in the basal expression of DUSP4, DUSP5 and DUSP6 in BaF3-EpoR cells compared to the expression in mCFU-E cells (Fig. 3D) was in agreement with the prediction by the mathematical model."

"Utilizing the amount of RSK experimentally detected in 32-EpoR cells as well as the amount of RSK present in the cells overexpressing RSK, the mathematical model predicted a major increase in pS6 whereas pAKT and ppERK remain rather unaffected. In line with this model prediction experimental overexpression of RSK had no effect on the dynamics of the upstream components pAKT and ppERK in 32D-EpoR cells but strongly increased the phosphorylation level of S6 (Fig. 4C). The mathematical model correctly predicted the effect of RSK overexpression in 32D-EpoR cells on the dynamics of ppERK and pRSK, but the peak amplitude of pAKT and pS6 were underestimated. In four independent experiments the integrated pS6 response was increased (Fig. 4C bottom right panel), validating the prediction of high RSK sensitivity of pS6 in this cell type."

4. Experiments done in Figure 4 should also be done for the other cell lines.

We followed the suggestion of the reviewer and introduced a new section within the Expanded View information L where we compare the responses of the mCFU-E, BaF3-EpoR and 32D-EpoR cells under the given conditions and adapted the results part accordingly (page 14):

"We determined the abundance of pathway components in 32D-EpoR cells by quantitative immunoblotting (Tab. 1) and utilized these concentrations in our mathematical model as cell-type-specific parameters. Without altering the previously determined global kinetic parameters, we simulated the putative response of pAKT, ppERK and pS6 at 50 U/ml Epo in 32D-EpoR cells and observed good agreement with the experimental data for ppERK and pS6 (Fig. 4A). For pAKT, the peak time and signal duration were correctly predicted, while the model overestimated the peak amplitude and steady state of pAKT. Further, given the similarities in the protein abundance of 32D-EpoR and BaF3-EpoR cells, we assumed similarities in the dynamics of pathway activation. However, model simulations in line with experimental data used for model calibration under those conditions (Fig. EV13) indicated that differences in the peak amplitude, signal duration and steady state existed between these two cell types (Expanded View information L)."

5. Loading controls in blots often show uneven loading of samples making quantification difficult.

For uneven loading, we correct with a spline-based normalization strategy (Schilling et al., 2005). For this normalization, we established housekeeper proteins and generated recombinant calibrator proteins that are spiked into immunoprecipitation samples, differ in size from the endogenous proteins, and harbour the epitope recognized by the antibody used for immunoprecipitation. The materials and methods section was revised accordingly (page 30):

"Sample loading on SDS-PAGE was randomized and corrected with a spline-based normalization strategy to avoid correlated blotting errors (Schilling *et al*, 2005)."

6. Figure S12. EpoR-GST runs smaller than EpoR, which under reducing blot conditions does not make sense as GST should add ~25kDa to the EpoR protein.

We thank the reviewer for pointing this out. To determine the EpoR molecules per cell an EpoR-GST fusion protein was used as calibrator protein that has been described in Bachmann et al., 2011. The fusion protein only contains the cytoplasmic domain of the EpoR (26.3 kDa) that harbours the epitope of the antibody used for immunoprecipitation and immunoblot detection. The entire fusion protein consisting of the GST purification tag and the cytoplasmic domain of the EpoR is approximately 52.3 kD and therefore runs at a different position compared to the EpoR. The figure legend of Fig. EV10 was revised accordingly.

7. Figure S25. The upper middle panel looks like a reproduction of the top panel.

The top two panels of Fig. S25, now revised Fig. EV28, showed the best two regression models and therefore the differences are rather subtle. The top panel shows the model prediction for the model that considers both, the integrated pS6 response and the cell-cycle indicator, as contributing to Epo-supported proliferation. The upper middle panel represents the model predictions of the best model. In this model the integrated pS6 response contributes to Epo-induced proliferation in mCFU-E cells whereas in BaF3-EpoR and 32D-EpoR cells only the cell-cycle indicator contributes to proliferation. As given in Tab. EV4, these two models perform similarly well. However, when comparing the utmost right columns and here the top and the very right tiles it becomes evident that the upper middle panel is not a mere reproduction of the top panel. We have added a paragraph in the results section to point out similarities and differences (page 19):

"Noteworthy, the models with contribution of both, the integrated pS6 response and the cell-cycle indicator, to proliferation were not significantly more informative than the models of individual contributions (Expanded View information P) but predicting proliferation in mCFU-E, BaF3-EpoR and 32D-EpoR cells upon Epo stimulation and inhibitor treatment similarly well (Fig. EV28)."

Reviewer #2:

This manuscript from Höfer, Klingmüller and co-workers (with four co-first authors, Adlung, Kar, Wagner, and She) reports findings from a modeling-experimental study, in which three cell type-specific mathematical models were developed and analyzed with the goal of better understanding erythropoietin (Epo) receptor (EpoR) signaling. Two major information transmission pathways leading to activation of proliferation (Raf/Mek/Erk and PI3K/Akt) were considered, with coarse representation of molecular mechanisms, in each of the models. The three models, which each consist of ordinary differential equations (ODEs), have the same structure and identical rate constants but differ with respect to total protein abundances. The authors estimated total abundances of key signaling proteins in the different cell types of interest on the basis of quantitative immunoblotting data and other considerations, including assumptions. Model parameter settings were adjusted to reproduce time-course measurements of phosphorylation (of EpoR, Erk, Akt, and S6). Inputs and perturbations considered in the study included different doses of Epo, kinase inhibitor treatments (affecting Akt and Mek, for example), and overexpression. The cell types considered were hematopoietic in origin: primary CFU-E cells, belonging to the myeloid-erythroid branch, from E13.5 Balb/c mice; a murine IL-3-dependent pro-B cell line (BaF3); and a murine IL-3-independent myeloid cell line (32D). These cell types all express EpoR and respond differently to Epo stimulation; they are commonly used to study EpoR signaling. The authors find that differences in protein abundances across the three cell types largely explain observed differences in their proliferative responses to Epo stimulation. The authors conclude that CFU-E cells activate proliferation primarily through the PI3K/Akt axis, BaF3 cells activate proliferation mainly through the Raf/Mek/Erk axis, and 32D cells use both axes of information flow. These conclusions are based on analysis of the models, including a sensitivity analysis, and experimental tests of model predictions.

The study has some weaknesses. Absolute protein abundances were not comprehensively quantified, which is possible with mass spectrometry-based proteomics (at least for the cell

lines). See Hein et al. (2015) [Cell 163:712-723] and Kulak et al. (2014) [Nat Methods 11:319-324].

We thank the reviewer for this suggestion. We have included the references and in a new experiment applied quantitative mass spectrometry in combination with the "proteomic ruler" method to quantify the absolute protein abundance of key signaling proteins in human erythroid progenitors at the stage of colony-forming unit-erythroid (hCFU-E) cells prepared from three independent, healthy stem-cell donors. Further we determined by quantitative mass spectrometry in combination with the "proteomic ruler" method the abundance of the key signaling proteins in mCFU-E and BaF3-EpoR cells. We show that the results obtained by quantitative mass spectrometry correlated with the values obtained with quantitative immunoblotting. These results are now depicted in the new Fig. 8 and summarized in a new paragraph in the results section (page 21):

"To relate the results obtained with the proteomic ruler method to the previous measurements in mCFU-E, BaF3-EpoR and 32D-EpoR utilizing quantitative immunoblotting and recombinant protein standards, we performed additional mass spectrometric measurements for mCFU-E and BaF3-EpoR cells. As shown in Fig. 8D, for the key signaling components in the two cell types, the number of molecules per cell determined with the different techniques showed a good correlation ($R^2=0.82$), confirming that the snapshot measurement by mass spectrometry yielded reliable results."

And in the discussion section (page 26):

"To determine the abundance of signaling components, we used in our approach quantitative immunoblotting (Schilling *et al*, 2005) and quantitative mass spectrometry in combination with the "proteomic ruler" method that is based on the determination of total protein concentrations relative to the abundance of histones (Wiśniewski *et al*, 2014). The results shown in Fig. 8C demonstrate that the abundance of the signaling components determined by quantitative immunoblotting is very comparable to the results obtained by quantitative proteome-wide mass spectrometric measurements (Hein *et al*, 2015; Kulak *et al*, 2014). Protein abundance in tumor material can be quantified using a label free approach as described above or a super-SILAC approach employing a labeled reference cell line (Zhang *et al*, 2014)."

Models were parameterized on the basis of population-averaged measurements (vs. single-cell measurements). Gene expression, as determined through microarray analysis, was equated with protein expression, which is dubious. Time-series data are correlated, so the data used for calibration of parameter values are not as informative as one might expect from a simple count of data points.

We entirely agree with the reviewer that the analysis of gene expression cannot be simply equated with protein expression. Therefore we utilized for model calibration primarily time- and dose-resolved measurements on key signaling proteins.

The induction of mRNA expression from microarray analysis was only used to identify cell-cycle genes that could be utilized to quantify Epo-induced cell-cycle progression in the different cells. We validated the Epo-induced expression of the selected cell-cycle genes in BaF3-EpoR and mCFU-E by qRT-PCR and confirmed their robust expression after three hours of Epo stimulation. The mRNA expression of the selected cell-cycle genes was used to calculate the cell-cycle indicator. We performed additional PI staining experiments for mCFU-E and BaF3-EpoR cells to confirm the validity of the cell-cycle indicator as a coefficient to quantify the G1-S progression in response to Epo stimulation. The results are now shown in a new Fig. EV5. Further we clarified in the text of the results section the description of the cell-cycle indicator (page 10):

"These results suggested that the quantification of the expression of *cyclinD2*, *cyclinG2* and *p27* might provide a good measure to compare Epo-induced cell-cycle progression in BaF3-EpoR and mCFU-E cells. To summarize the contribution of the cell-cycle activator and the

two **cell-cycle** repressors that counteract each other in controlling cell-cycle progression, we defined a cell-cycle indicator as follows:

$$\frac{[\text{cyclinD2}]}{\sqrt{[\text{cyclinG2}] \times [\text{p27}]}}$$

This coefficient reflects complex regulation of cell-cycle progression in response to Epo stimulation better than individual components (e.g. only *cyclinG2*) and was strongly increased after 3 hours of Epo addition in mCFU-E and BaF3-EpoR cells (Fig. 1F, right panel).

Notably, the cell-cycle indicator was significantly ($p=0.04$) higher in BaF3-EpoR cells (Fig. 1F, right panel), which is in agreement with the results of the propidium iodide staining experiment (Fig. EV5) showing that, whereas mCFU-E cells are already committed to cell-cycle progression, an increasing fraction of BaF3-EpoR cells enters S/G2/M phase in response to stimulation with increasing Epo doses."

To validate the model prediction that the basal expression of DUSP family members is elevated in BaF3-EpoR cells compared to mCFU-E we aimed to experimentally quantify the expression levels of DUSP family members. However, in our quantitative proteome wide mass spectrometric studies of BaF3-EpoR and mCFU-E cells the basal expression levels of DUSP were unfortunately below the detection limit. Therefore we utilized mRNA expression as a proxy to determine the relation of DUSP abundance between DUSP in BaF3-EpoR cells. To better clarify this we adapted the paragraph in the results section accordingly (page 13):

"The analysis of DUSP protein abundance is challenging because multiple isoforms with different functions exist and only very few antibodies, mostly with low specificity, are available. In our proteome-wide quantitative mass spectrometry analysis of unstimulated mCFU-E, hCFU-E and BaF3-EpoR cells, Epo-regulated DUSP family members were below the detection limit. Therefore, we used the mRNA expression levels as proxy, assuming at least some correlation with protein expression."

We agree with the reviewer that the sheer number of the 432 data points we used for model calibration does not yet provide information on how informative the data is. We ensured by the use of different Epo doses ranging from 0.5 to 50 U/ml Epo, overexpression of the negative regulators SHIP1 and PTEN, treatment with inhibitors of AKT, MEK, mTOR and RSK, each at least at four different doses, with AKT and MEK inhibitor also in combination, and time-resolved measurements, that various experimental conditions provided versatile information to parameterize our mathematical model. In the new Fig. EV21 we show that the observables for integrated pS6 and the cell-cycle indicator do not correlate with each other. The χ^2 of 514 for 432 data points indicates that the structure of our integrated mathematical model (Fig. 2A) is sufficient to describe the complexity of the experimental data.

The model has only coarse mechanistic resolution, which is intentional and somewhat justified, but there was not a rigorous reduction of model resolution from fine to coarse, which leaves room for doubt about the validity of the model structure, which is a common concern about biological models. Annotation/justification of the model structure is minimal.

To better emphasize that details on the mathematical model and the systematic model reduction can be found in the Expanded View Information F, we now specified the text as follows (page 11):

"By a systematic model reduction (Expanded View information F.2), we tested the binding rates of the adaptor proteins Gab1/2 (Sun et al, 2008) to the EpoR. We identified that the adapter proteins Gab1/2 may bind either very fast or slow and therefore play a negligible role in the fast equilibrium of receptor-adaptor complex formation. Additionally we decomposed the enzymatic rate constants (e. g., for phosphatases and kinases) into the product of total enzyme concentration and a biochemical rate constant (also called catalytic efficiency, or

turnover, k_{cat}). This decomposition enabled us to quantify the biochemical rate constant as a property of the enzyme, which therefore can be assumed to be independent of a given cell type, whereas the enzyme concentration is cell-type-specific (Expanded View information F). For further details on the coupled ordinary differential equations, the dynamic variables, the parameter estimation as well as their annotation (Tab. EV1) and their sensitivities towards inhibitors see Expanded View information F. The full SBML model is available at the biomodels.org database."

The D2D software package and the model analyses performed with it rely on commercial software (MATLAB), which may be a hindrance to some researchers desiring to repeat the analyses. There are no easy ways to address these weaknesses and they are offset by considerable strengths. Overall the study is impressive. My only advice is that the authors might want to consider adding more cautionary discussion of limitations, which could be helpful for some readers, although the discussion of limitations is acceptable in current form.

We would like to point out that the D2D software package is open-source and free to use: <http://data2dynamics.org/>, and that there are open-source alternatives to MATLAB such as GNU Octave (<http://www.gnu.org/software/octave/>). The provided SBML-compatible xml file also allows uploading of our mathematical model into other open-source alternatives such as CoPaSi (<http://copasi.org/>).

To better point out the limitations of our study, we now mention throughout the manuscript that we refer to proliferation upon Epo stimulation and the impact of inhibitor treatment. We also confirmed the validity of our approaches for the quantification of protein abundance and revised the discussion accordingly (page 26):

"To determine the abundance of signaling components, we used in our approach quantitative immunoblotting (Schilling et al, 2005) and quantitative mass spectrometry in combination with the "proteomic ruler" method that is based on the determination of total protein concentrations relative to the abundance of histones (Wiśniewski et al, 2014). The results shown in Fig. 8C demonstrate that the abundance of the signaling components determined by quantitative immunoblotting is very comparable to the results obtained by quantitative proteome-wide mass spectrometric measurements (Hein et al, 2015; Kulak et al, 2014). Protein abundance in tumor material can be quantified using a label free approach as described above or a super-SILAC approach employing a labeled reference cell line (Zhang et al, 2014)."

An especially innovative aspect of this study is the integration of a mechanistic model with a regression model, which is impressive. I expect to see more examples of this type of approach in the future.

We thank the reviewer for this encouraging statement.

The datasets generated in this study and used in modeling may be valuable to other researchers, especially modelers. I would encourage the authors to provide the data in as many different commonly used formats as possible, to make the data as easily accessible as possible.

We are happy to share our data with the community. We provide the annotated raw files for quantitative and qualitative immunoblotting at the following link: <http://bit.ly/1RPddTD>. As soon as the manuscript is accepted, we will provide the data files freely accessible at the systems biology data and model management platform SEEK: <http://seek.sbepo.de/>. This is now indicated in the materials and methods section (page 31):

"The raw data of all qualitative and quantitative immunoblots of this work can be accessed under the following link: <http://bit.ly/1RPddTD>. Data of quantified immunoblots has also been uploaded through Excmplify (Shi et al, 2013) to the <http://seek.sbepo.de/> SEEK platform (Wolstencroft et al, 2015).

The mass spectrometry proteomics data have been deposited to the ProteomeXchange Consortium via the PRIDE (Vizcaíno *et al*, 2016) partner repository with the dataset identifier PXD004816."

The reviewer can access the proteome data of the hCFU-E cells already now at <http://www.ebi.ac.uk/pride/archive/login> using the login information given below.

Username: reviewer44839@ebi.ac.uk

Password: BX1sL2KY

The authors should acknowledge earlier related modeling work, where cell line-specific measurements of protein abundances were used to obtain cell line-specific models [Stites EC *et al*. (2015) *Biophys J* 108:1819-1829] and where patient-specific data were used to obtain patient-specific models [Fey D *et al*. (2015) *Sci Signal* 8: ra130].

We thank the referee for this advice. We revised the respective paragraph in the discussion section accordingly (page 25, 26):

"The concept of protein abundance determining the utilization of connections and the dynamics of cellular signal transduction (Expanded View information F) is not limited to hematopoietic cell types and can be extended to other cells (Merkle *et al*, 2016). It was shown that the abundance of growth factor receptors correlates with growth factor responses and AKT/ERK bias in diverse breast cancer cell lines (Niepel *et al*, 2014). We show here that the mere abundance of a cytokine receptor such as the EpoR is not sufficient to explain proliferative responses. Whereas mCFU-E and hCFU-cells harbor comparable levels of the EpoR, the abundance of key signaling molecules is very distinct and culminates in major difference in the sensitivity of their Epo-induced proliferative responses towards the AKT and MEK inhibitor. Further refinements might become necessary, as the signaling network topology implemented here involves simplifications and neglects presence of different isoforms such as Gab1 in hCFU-E and mCFU-E cells and Gab2 in BaF3-EpoR and 32D-EpoR cells (Tab. 1, Fig. 8E, Expanded View information F), which might need to be included in a larger context as these low-abundant proteins can be the bottlenecks of signaling (van den Akker *et al*, 2004; Shi *et al*, 2016). However, even the cell-type-specific wiring of feedback loops in signal transduction (Klinger *et al*, 2013; D'Alessandro *et al*, 2015; Stites *et al*, 2015) ultimately depends on protein expression and can thus be captured by the conceptual framework proposed here. By combining the analysis of protein abundance and structure data Kiel *et al*. showed that the abundance of signaling components determined the cell-context specific topology of the ErbB signaling network (Kiel *et al*, 2013). Further, predicting signaling dynamics from (comparatively simple) static measurements of protein abundance may become of practical use for prognosis, as shown for the JNK network in neuroblastoma (Fey *et al*, 2015)."

I think this manuscript is significant because it sheds light on the importance of knowing the characteristic protein abundance profile of a cell type when one is attempting to predict cellular responses to signals and perturbations.

Reviewer #3:

In the presented paper by Adlunt *et al*., the authors quantitatively measure the role of multiple components of the signaling network on cell growth, cell-cycle progression and proliferation in three cell lines of hematopoietic origin.

By measuring the dynamics of numerous signaling proteins they demonstrate that different relative inputs of two cell proliferation signaling pathways (ERK and AKT) are different in different cell lines and can be explained by the difference in signaling protein abundances. By measuring these profiles as well as responses to various stimuli in two cell lines the authors fit a mathematical model that was then able to predict the response of the third cell line to the stimuli based solely on the protein abundance profile measurement in that line.

This part of the work is interesting and pretty convincing.

In the second part of the paper the authors claim that two different processes can play a dominant role in cell proliferation control in different cell types: protein synthesis is rate-limiting for faster cycling cells, while G1-S control is limiting for slower cycling cells. Although this idea seems logical, I think the authors need to provide some stronger data to support this claim. All in all, when revised, this work will be suitable for publication in *Molecular Systems Biology*.

Major points:

1) The "cell-cycle indicator" chosen by the authors, which combines the transcription levels of three cell cycle genes (*cycD2*, *cycG2*, and *p27*), needs a more thorough validation to confirm that it actually reflects the rate of the cell cycle progression. Current justification that these genes demonstrate a strong expression change at 3 hours after EpoR stimulation and therefore can be used to estimate the G1-S transition rate is not sufficient. Comparison with other combinations of known cell cycle genes (such as *e2f1*, *cycE*, ...) would be more informative, as well as some measurements of these parameters in the cells cycling with a different rate (e.g., stimulated with different concentrations of growth factors, or treated with inhibitors).

We realized that we needed to clarify the "cell-cycle indicator". The aim was to establish a measure to quantify Epo-induced cell-cycle progression. We realized after multiple rounds of discussion that it was best to focus on cell-cycle regulated genes strongly responsive to Epo already after three hours of treatment in mCFU-E and BaF3-EpoR cells and here utilizing the ratio of the Epo-induced expression of the selected genes was most informative. We followed the reviewer's advice and inspected the induction of additional genes such as *cyclinE1* (*Ccne1*) and *cyclinE2* (*Ccne2*). However, as depicted in Fig. EV4, the expression of none of these genes was strongly regulated upon Epo stimulation in BaF3-EpoR cells. The mRNA of the *E2f1* gene was not significantly regulated upon Epo stimulation and is therefore not depicted in Fig. EV4. To directly access Epo-stimulated cell-cycle progression in BaF3-EpoR and mCFU-E cells, we performed PI staining experiments (new Fig. EV5) to confirm the validity of the cell-cycle indicator as a coefficient for the timing of G1-S progression (as schematically shown in Fig. EV23A). The PI staining showed that the fraction of BaF3-EpoR cells in the S/G2/M phase of the cell-cycle increased with increasing Epo doses whereas a large fraction of mCFU-E cells was already committed to cell-cycle progression as these cells are already exposed to Epo in the fetal liver. We revised the text of the results section accordingly (page 9, 10):

"These genes are not the only genes involved in the regulation of the cell cycle. However, the results of the quantitative RT-PCR analysis (Fig. 1F) showed that after three hours of Epo stimulation, mRNA induction of *cyclinD2* (*CCND2*) and mRNA repression of *cyclinG2* (*CCNG2*) and *p27* (*CDKN1B*) had comparable fold changes in BaF3-EpoR and mCFU-E cells. On the other hand, *cyclinE1* (*CCNE1*) and *cyclinE2* (*CCNE2*) showed only little regulation in both cell type. These results suggested that the quantification of the expression of *cyclinD2*, *cyclinG2* and *p27* might provide a good measure to compare Epo-induced cell-cycle progression in BaF3-EpoR and mCFU-E cells. To summarize the contribution of the cell-cycle activator and the two cell-cycle repressors that counteract each other in controlling cell-cycle progression, we defined a cell-cycle indicator as follows:

$$\frac{[\textit{cyclinD2}]}{\sqrt{[\textit{cyclinG2}] \times [\textit{p27}]}}$$

This coefficient reflects complex regulation of cell-cycle progression in response to Epo stimulation better than individual components (e.g. only *cyclinG2*) and was strongly increased after 3 hours of Epo addition in mCFU-E and BaF3-EpoR cells (Fig. 1F, right panel).

Notably, the cell-cycle indicator was significantly ($p=0.04$) higher in BaF3-EpoR cells (Fig. 1F, right panel), which is in agreement with the results of the propidium iodide staining

experiment (Fig. EV5) showing that, whereas mCFU-E cells are already committed to cell-cycle progression, an increasing fraction of BaF3-EpoR cells enters S/G2/M phase in response to stimulation with increasing Epo doses."

2) The authors use their mathematical model that was fit based on the data from the two cell lines (CFU-E and BaF3-EpoR) to predict the response of the third cell line (32D-EpoR). However, it is not clear how good their prediction is. E.g., in Fig 4A it would be nice to show besides 32D-EpoR predictions and actual measurements, also the responses of the CFU-E and BaF3-EpoR cells to the same stimulus. This would demonstrate how close the prediction for 32D-EpoR is to the experimental data, compared to the difference between different cell lines. Given how similar the protein expression profiles are between BaF3-EpoR and 32D-EpoR (Table 1), one might expect that the 32D-EpoR response curve could be well approximated simply by the BaF3-EpoR curve. So, the authors at least need to show that their model, which integrates the data from both CFU-E and BaF3-EpoR cell lines, gives a better prediction than BaF3-EpoR data alone.

We thank the reviewer for this suggestion. In the new section Expanded View information L, we now compare simulations for the three different cell types and find that, despite similarities between BaF3-EpoR and 32D-EpoR cells, the cell-type-specific protein abundance results in different dynamic responses of pAKT, ppERK and pS6 upon stimulation with 50 U/ml Epo (Fig. EV16). The χ^2 of comparing these simulations to the experimental data of 32D-EpoR cells indicates that the mathematical model adapted to the protein abundance in 32D-EpoR cells describes the measured dynamics in 32D-EpoR cells for pAKT and ppERK much better than the mathematical model based on the protein abundance in BaF3-EpoR cells (Tab. EV2). The better agreement for pS6 dynamics with the mathematical model based on the protein abundance in BaF3-EpoR cells might be due to the degree of freedom originating from the scaling that was estimated for these simulations to fit the data set. We have revised the results section of the manuscript accordingly (page 14):

"For pAKT, the peak time and signal duration were correctly predicted, while the model overestimated the peak amplitude and steady state of pAKT. Further, given the similarities in the protein abundance of 32D-EpoR and BaF3-EpoR cells, we assumed similarities in the dynamics of pathway activation. However, model simulations in line with experimental data used for model calibration under those conditions (Fig. EV13) indicated that differences in the peak amplitude, signal duration and steady state existed between these two cell types (Expanded View information L)."

3) Authors use the pS6 levels and the cell-cycle indicator measurements to distinguish whether it is the cell growth or the levels of G1-S regulators that is most critical for determining the proliferation rate for different cell lines. However, these two parameters (growth rate and G1-S transition) are not independent as cell size also feeds into the G1-S regulatory network. To show that these parameters can be separated and considered as at least quasi-independent in the context of this study, the authors should test the correlation between the two (e.g., at different levels of Epo stimulation and pathway inhibitors).

We followed the reviewer's advice and show in the new Fig. EV21 that the integrated pS6 response and the cell-cycle indicator are not correlated, which suggests that both processes are at least to some extent independently regulated. We adapted the results part accordingly (page 19):

"To quantitatively connect the integrated pS6 response and the cell-cycle indicator to cell proliferation upon Epo stimulation and inhibitor treatment (Fig. 6A), the proliferation of mCFU-E, BaF3-EpoR and 32D-EpoR cells in response to 5 U/ml Epo and in the absence or presence of 0.005 μ M, 0.05 μ M, 0.5 μ M and 5 μ M of AKT VIII or U0126 was measured. The analysis shown in Expanded View information P (Fig. EV21) revealed that both variables are not correlated and therefore are likely to be regulated independently."

These observations are also in line with the finding that the integrated pS6 response and the cell-cycle indicator cannot be linked equally well to proliferation upon Epo stimulation and single inhibitor treatment shown by multiple linear regression analysis and model selection (Tab. EV5, Fig. EV27, Fig. EV28). We added a clarifying sentence in the results section (page 19):

"Noteworthy, the models with contribution of both, the integrated pS6 response and the cell-cycle indicator, to proliferation were not significantly more informative than the models of individual contributions (Expanded View information P) but predicting proliferation in mCFU-E, BaF3-EpoR and 32D-EpoR cells upon Epo stimulation and inhibitor treatment similarly well (Fig. EV28)."

4) The authors make a strong statement that while slower cycling cells (BaF3-EpoR, 32D-EpoR) are controlled at the level of G1-S progression, faster cycling cells (CFU-E) are controlled by protein synthesis rates. In my opinion, this conclusion is not supported strongly by the data provided. Fig. 6C shows that 2.5-fold change in pS6 levels causes only a ~10% change in proliferation rate in CFU-E cells. The cell-cycle indicator has approximately the same small effect on proliferation for this cell line (Fig. S24). And also, as I already mentioned, the cell-cycle indicator needs some further validation.

We agree with the reviewer that we needed further evidence to support the conclusion that proliferation of slower cycling cells is controlled at the level of G1-S progression. We revised Fig. 1F to show that upon Epo stimulation the cell-cycle indicator is increased to significantly ($p=0.04$) higher levels in BaF3-EpoR cells compared to mCFU-E cells, which is further supported by our new Fig. EV5 as explained above. In addition, we validated our concept with new experiments using human erythroid progenitors at the stage of colony-forming unit-erythroid (hCFU-E) cells prepared from three independent, healthy stem-cell donors. The doubling time of the hCFU-E cells is considerably slower than those of the murine CFU-E cells, as we show in the new Fig. EV31. Applying our concept that proliferation of slower cycling cells is controlled at the level of G1-S progression, we were able to correctly predict that, different from mCFU-E cells, a combined treatment with AKT and MEK inhibitor is much more effective than the individual inhibitors to reduce Epo-induced proliferation of hCFU-E cells as we show in the new Fig. 8F. The concept that the response to Epo stimulation and inhibitor treatment in slower cycling cells is controlled at the level of the cell-cycle indicator is now better explained in the new Expanded View information section T.

The reviewer correctly points out that the inhibitor treatment only mildly reduced Epo-induced proliferation in mCFU-E cells compared to BaF3-EpoR cells and 32D-EpoR cells. The effect of the inhibitor treatment on the cell-cycle indicator (reduced to minimum 41%) is much stronger than the effect on proliferation (reduced to 79%) in mCFU-E cells. However, we would like to point out that in 32D-EpoR, BaF3-EpoR and particularly in hCFU-E cells, Epo-induced proliferation could be reduced to around 50%. We now mention this interesting aspect in the discussion section (page 25):

"We show here that the mere abundance of a cytokine receptor such as the EpoR is not sufficient to explain proliferative responses. Whereas mCFU-E and hCFU-cells harbor comparable levels of the EpoR, the abundance of key signaling molecules is very distinct and culminates in major difference in the sensitivity of their Epo-induced proliferative responses towards the AKT and MEK inhibitor."

5) In the last part of the results section the authors show that their mathematical model predicts the combined effect of MEK and AKT pathway inhibitors on the proliferation of the studied cells. The result looks nice but might be trivial: according to Fig. 6C, U0126 does not have any effect on CFU-E and 32D-EpoR cell lines and only decreases the proliferation rate in BaF3-EpoR cells, while AKT VIII has some effect in all three lines. So, predicting the combined effect of the two drugs one of which does not have an effect on its own looks trivial (CFU-E, 32D-EpoR). So, the only interesting prediction is a cooperative effect of the two inhibitors on BaF3-EpoR. This result would look stronger if the authors could demonstrate an opposite example finding two inhibitors, each of which would affect the cell proliferation rate

but together they would not show a combined effect.

We appreciate the reviewer's comment. We completely revised the Expanded View information section S to resolve the different levels at which the AKT and MEK inhibitors exert cell-type-specific effects. Additionally, as mentioned above, we were able to validate our findings in hCFU-E cells. We determined the protein abundance of signaling components in hCFU-E cells by quantitative mass spectrometry in combination with the "proteomic ruler" method. As explained in a new Expanded View information section T, we derived the sensitivities of the Epo-induced activation of signal transduction towards the inhibitor treatment based on the protein abundance of AKT and MEK in hCFU-E cells. We linked, based on cell size of hCFU-E cells and their Epo-induced proliferation dynamics, the cell-cycle indicator to the impact of the inhibitors on proliferation in response to Epo stimulation. As we show in the new Fig. 8, despite similar levels of the EpoR, is the effect of co-treatment with AKT VIII and U0126 on Epo-induced proliferation in human CFU-E cells, unlike in murine CFU-E cells, much stronger than the effect of the single-inhibitor treatment. We have revised the results section accordingly (page 21, 22):

"To highlight that the protein abundance governs cell-type-specific regulation of Epo-induced proliferation and as a consequence the sensitivity towards inhibitors, we prepared human CFU-E cells from CD34⁺ cells mobilized into the peripheral blood of three healthy donors. By means of mass spectrometry, we quantified 6,925 proteins of which 5,912 proteins were shared among the hCFU-E cells from the three independent donors (Fig. 8A). Next, we applied the "proteomic ruler" method (Wiśniewski et al, 2014) to calculate the copy numbers of individual proteins per cell. To convert these numbers into cytoplasmic concentrations, we measured the average cell size by imaging flow cytometry (Fig. 8B) and calculated the cytoplasmic volume from these data. The average cytoplasmic volume of the hCFU-E of the three donors was comparable but considerably larger than the volume of the mCFU-E cells (Fig. 8C). To relate the results obtained with the proteomic ruler method to the previous measurements in mCFU-E, BaF3-EpoR and 32D-EpoR utilizing quantitative immunoblotting and recombinant protein standards, we performed additional mass spectrometric measurements for mCFU-E and BaF3-EpoR cells. As shown in Fig. 8D, for the key signaling components in the two cell types, the number of molecules per cell determined with the different techniques showed a good correlation ($R^2=0.82$), validating our determinations by quantitative immunoblotting and confirming that the snapshot measurement by mass spectrometry yielded reliable results. Surprisingly, the protein abundance of the key signaling proteins determined for hCFU-E cells showed very low variance between the three independent donors but were highly distinct from the values obtained for mCFU-E cells as well as for the murine cells lines BaF3-EpoR and 32D-EpoR (Fig. 8E). For example Ras, Raf and AKT were present at much lower levels in hCFU-E compared to mCFU-E, BaF3-EpoR and 32-EpoR cells, whereas for PI3 kinase elevated levels were observed in hCFU-E cells. As expected, the levels of the EpoR were comparable in hCFU-E and mCFU-E and elevated to the same extent in Ba3-EpoR and 32D-EpoR cells. To link the signaling layer to proliferation, we applied our concept that larger (see Fig. 8B) and more slowly dividing (Fig. EV30) cells regulate proliferation primarily by the control of G1-S progression (Expanded View information R). Since the measured protein abundance of the key signaling proteins was highly comparable in the hCFU-E cells of the three donors, we used the average concentrations of these proteins to predict with our mathematical model the impact of the AKT inhibitor and MEK inhibitor on Epo-supported cell proliferation of hCFU-E cells. Without any further information (Expanded View information T) the mathematical model predicted that hCFU-E cells are less sensitive to the individual inhibitors but, distinct from mCFU-E cells that are primarily sensitive to the AKT inhibitor, rather require treatment with the combination of both, AKT VIII and U0126, to effectively inhibit Epo-induced proliferation (Fig. 8F). To validate this model prediction experimentally, we quantified the numbers of hCFU-E cells after 96 hours of stimulation with 5 U/ml Epo and single or combined treatment with AKT VIII and U0126. Since hCFU-E cells from donor 3 showed the most robust proliferative response (Fig. EV31), we analyzed proliferation of hCFU-E cells from this donor in biological triplicates (Fig. 8F). Qualitatively the same effects on

proliferation were observed for the hCFU-E cells from donor 1 and donor 2 (Fig. EV32). As shown in Fig. 8E, in line with the model prediction, Epo-induced proliferation of hCFU-E was much more sensitive towards combinatorial treatment with AKT VIII and U0126 compared to the treatment with the individual inhibitors.

These findings showcase that protein abundance can be reliably measured from snapshot data of human material. Based on these data, our integrative mathematical model allows to evaluate combinatorial application of inhibitors in silico and thus may serve to improve the treatment of proliferative disorders such as tumors driven by exacerbated growth-factor signaling."

Minor points:

1) In Fig 1C, the legend mentions cells "stained with Calcein (upper right panel)" that are actually not shown on the figure. Also the legend says that the cell diameter was measured by bright-field microscopy, while in the Results section is said that confocal microscopy was used for this. Was the cytoplasmic volume used for the calculation of protein concentrations measured including or excluding the nucleus? The data from the table in Fig. 1C is already shown in Table 1, so there is no need to duplicate.

We thank the reviewer for these comments. We determined cell diameters and cytoplasmic volumes by imaging flow cytometry and show the fluorescence microscopy pictures just as representative examples. We followed the reviewer's suggestion and removed the information given in Tab. 1 from the revised Fig. 1. The figure legend and the materials and methods section were corrected accordingly (page 31, 42):

"To determine cellular volumes, the cytoplasm was stained with Calcein (eBioscience) and DNA was stained with DRAQ5 (Cell Signaling). Imaging flow cytometry was performed on an Amnis ImageStream^X (Merck Millipore) and data was analyzed with the IDEAS Application v5.0 (Merck Millipore)."

"Size determinations of mCFU-E and BaF3-EpoR cells. Exemplary fluorescence microscopy pictures upon Hoechst staining for nucleus visualization with 60x objective. The bar represents 10 μ m distance (upper panel). Cell diameter was measured by Imaging flow cytometry. Cytoplasm was stained with Calcein, nuclei were stained with DRAQ5. Probability density function of size distribution with indicated mean diameter of mCFU-E and BaF3-EpoR cells. All cells were growth-factor deprived and unstimulated."

2) There is an inconsistency in what concentration of Epo is used for saturating stimulation. In the beginning of the paper it is 50 U/ml, then it becomes 5 U/ml, then for microarray 1 U/ml was used. Please comment on this.

We corrected these inconsistencies and made sure that all stated saturating doses were in fact saturating also in terms of stoichiometric values as we now show in the new **Expanded View information A** for the proliferative response and for the signaling responses in Fig. EV13. We revised the results part accordingly (page 6):

"In a first step the protein abundance and dynamics of phosphorylation of EpoR, AKT and ERK in response to stimulation with a saturating Epo concentration (50 U/ml, see **Expanded View information A**, Fig. EV1, Fig. EV13) was qualitatively assessed."

3) On p.12 the authors conclude that "CFU-E cells signaling to S6 primarily through the AKT axis, BaF3-EpoR cells primarily through the ERK pathway and 32D-EpoR cells using both pathways", but earlier in the same section (Fig. 4B) they have shown that actually 32D-EpoR are very similar to BaF3-EpoR in that influence of the Ras/MEK/ERK cascade has a dominant effect on pS6 with a only a slight impact of AKT pathway.

We followed the reviewer's suggestion and rephrased the text in the results part (page 16):

"Taken together, our results show that the abundances of the network components direct the signal flow differentially through the AKT and Ras/MEK/ERK pathways. In mCFU-E cells signaling to S6 occurs primarily through the AKT axis and in BaF3-EpoR cells primarily through the ERK pathway. In 32D-EpoR cells signaling to S6 is similar to BaF3-EpoR cells, but with slightly higher sensitivity towards AKT."

4) On p.16 the authors compare the experimental and predicted effects of MEK and AKT pathway inhibitors on the cell-cycle indicator levels (Fig. 6B, S17) and conclude that "effects of the inhibitor treatment on the cell-cycle indicator are explained very well by changes in the integrated pAKT and ppERK responses". Please comment on why the model always underestimates the effect of the inhibitors (most obvious - in case of U0126 in 32D-EpoR).

In the Expanded View information O, we explain that, due to the nature of a linear regression with forced intercept = 0, the effect of the highest inhibitor doses cannot be described. We decided against non-linear regression since in that case the system would be over-parameterized and the agreement with the linear model for physiologically more relevant intermediate inhibitor doses was sufficient. We revised the mentioned sentence (page 18,19):

"The measured values of the integrated pAKT response, the integrated ppERK response and the cell-cycle indicator for mCFU-E, BaF3-EpoR and 32D-EpoR cells yielded high correlation, indicating that the effects of the inhibitor treatment on the cell-cycle indicator are explained very well by changes in the integrated pAKT response and the integrated ppERK response with only slight deviation at the highest inhibitor doses (Fig. 6B; Expanded View information O)."

5) When describing the results of the regression analysis of whether the pS6 levels or the cell-cycle indicator better predict the proliferation rate in each cell line (p.17), it's worth mentioning explicitly that the combination of the two was also tested and found less predictive than the best one alone. Does this suggest that one of these two parameters already contains the information about another one (so, they are not independent)? How was the combination of the two parameters exactly calculated?

We thank the reviewer for pointing this out. The combination of both variables, the integrated pS6 response and the cell-cycle indicator, is not less predictive than the individual variables alone but is also not significantly better. We clarified the sentence in the results section accordingly (page 19):

"Noteworthy, the models with contribution of both, the integrated pS6 response and the cell-cycle indicator, to proliferation were not significantly more informative than the models of individual contributions (Expanded View information P) but predicting proliferation in mCFU-E, BaF3-EpoR and 32D-EpoR cells upon Epo stimulation and inhibitor treatment similarly well (Fig. EV28)."

In addition, we show in a new Fig. EV21, that both variables are not correlated. We also mention this in the results section (page 19):

"The analysis shown in Expanded View information P (Fig. EV21) revealed that both variables are not correlated and therefore are likely to be regulated independently."

6) When predicting the combined effect of the two inhibitors on cell proliferation, the authors say that "For CFU-E and 32D-EpoR cells, AKT inhibition was predicted to control cell proliferation in a dose-dependent manner, without combined effect with ERK inhibition" (p. 18). However, actually from Fig. 7B for 32D-EpoR it seems like these the two inhibitors are predicted to have a combined effect, although the U0126 effect is relatively small.

The predicted effect of the inhibitors is now stated more carefully (page 20):

"For mCFU-E and 32D-EpoR cells, AKT inhibition was predicted to control Epo-induced cell proliferation in a dose-dependent manner, without or negligible combined effect of MEK inhibition. Only for BaF3-EpoR cells, the model indicated that Epo-induced cell proliferation is strongly inhibited by increasing doses of both AKT VIII and U0126, resulting in a combined effect of both drugs together (Fig. 7B, upper right panel)."

Thank you again for submitting your work to Molecular Systems Biology. We have now heard back from the three referees who agreed to evaluate your study. The referees are the same that reviewed the previously rejected manuscript MSB-16-6819. As you will see below, reviewers #1 and #3 raise a number of concerns, which preclude the publication of the study in its current form.

Since most of the issues raised are related to the key conclusions of the study they need to be convincingly addressed in a major revision. Please note that our editorial policy in principle allows a single round of major revision.

REFEREE REPORTS

Reviewer #1:

In the revision, the authors addressed many technical issues regarding stoichiometrically consistent Epo treatment of the cells, western blot displays, and some statistical analysis. However, other concerns are simply addressed in words: the 'cell cycle indicator' remains a heuristic that was likely chosen because it gave the authors the answer they wanted. This reviewer appreciates the effort made to add primary human CFU-E cells to address the concern raised about EpoR overexpression in the cell lines. If indeed human CFU-Es behaved like BaF3 cells, it would address the concerns related to ectopic expression. However, visual inspection of the data as shown in Fig. 8F are not convincing and the lack of statistical testing on these new analyses prevents the reader from being able to draw any conclusions about the additional cell type.

A more pervasive, and unresolved, concern relates to overstated claims based on the data as presented. To enumerate:

Conclusions not transparent based on the data shown:

1. "The concentration of ERK was higher in BaF3-EpoR... than mCFU-E cells... whereas the concentration of AKT was comparable" (Page 7): in Figure 1D, both ERK and AKT appear similarly more abundant in BaF3-EpoR cells compared to mCFU-E cells.
2. "the cell-cycle inhibitor was... higher in BaF3-EpoR cells... which is in agreement with the results of the propidium iodide staining experiment..." (Page 9): based on the data shown, the percent of cells in S/G2/M is higher in mCFU-E cells than in BaF3-EpoR cells; the authors should display the data as fold change over unstimulated if they want to indicate EpoR responsiveness. But note: the fold change in Epo-stimulated S/G2/M cells (~1.5-fold for mCFU-E vs. ~7-fold for BaF3-EpoR) correlates poorly with the fold-change in their cell-cycle indicator (~12-fold for mCFU-E vs. ~17-fold for BaF3-EpoR).

Conclusions not supported by data:

1. "The dynamics of ppERK was more sustained in BaF3-EpoR cells than in mCFU-E cells" (Page 8): two data points in Figure 1D suggesting a slightly prolonged activation are ignored in the model fits of Figure 2D where the modeled time courses are close to overlapping.
2. "our mathematical analysis indicates that cell-type-specific processing of Epo stimuli is due to different abundances of the signaling proteins" (Page 12): total overstatement-up to this point in the manuscript, the results only suggest that differences in signal processing can be explained by different abundances in signaling proteins, based on a model whose free parameters were fit to capture those differences in signal processing.
3. "RSK abundance... exhibited a high impact on the integrated pS6 response in BaF3-EpoR and 32D-EpoR cells but not in mCFU-E cells" (Page 15): the authors overexpress RSK in 32D-EpoR cells (Figure 4C), which is a perturbation in the wrong direction because 32D-EpoR cells already express more RSK than mCFU-E cells based on the quantitative data in Table 1. The experiment should have been a RSK knockdown. At a minimum, RSK would need to be overexpressed in mCFU-E cells and show no effect (or less of an effect) in the model and in the experiments.

4. "overexpression of constitutively active Ras qualitatively resulted in a stronger elevation of S6 phosphorylation in BaF3-EpoR than in mCFU-E cells" (Page 16): neither qualitatively nor quantitatively true-in Figure EV17A, the pS6 bands for Ras-G12V overexpression in mCFU-E and BaF3-EpoR are on two different membranes, so quantitative comparisons cannot be made directly; moreover, the quantitative data in Figure EV17B are superimposable (within measurement error) after one takes into account that the y-axes are different scales for mCFU-E and BaF3-EpoR. The inhibitor data in Figure 5 are much stronger in this regard.

5. "the more severe impact of PTEN overexpression compared to SHIP1 overexpression on the Epo-induced proliferation of mCFU-E and BaF3-EpoR cells was correctly predicted by our mathematical model" (Page 20): the Epo EC50 for both modeling and experiments is comparable regardless of whether PTEN or SHIP1 is overexpressed (barring a statistical test that demonstrates otherwise). What changes is the baseline proliferation of mCFU-E cells (captured by the model for PTEN but not SHIP1), propagating to the maximally observed proliferation under conditions of saturating Epo.

6. "the number of molecules per cell determined with the different techniques showed a good correlation ($R^2 = 0.82$), validating our determinations by quantitative immunoblotting and confirming that the snapshot measurement by mass spectrometry yielded reliable results" (Page 21): Figure 8D is on a log-log plot and the slope is roughly one half, meaning that there is a square-root dependence (rather than a linear one) between the immunoblot data and the mass spec data.

7. "Qualitatively the same effects on proliferation were observed for the hCFU-E cells from donor 1 and donor 2" (Page 22): the model predicts comparable sensitivity to the MEK and Akt inhibitors, with maximal inhibition with the combination. Donor 1 is highly sensitive to both inhibitors, with the Akt inhibitor dominating. Donor 2 is resistant to both inhibitors with the exception of the highest doses of Akt inhibitor. There is very little qualitatively similar between these two donors aside from the dominance of the Akt inhibitor, which disagrees with the model predictions.

Inconsistencies within the manuscript:

1. Transcriptome analysis was performed with 5 U/ml Epo (Page 9). Transcriptome analysis was performed with 1 U/ml Epo (Expanded View information A).

In summary, authors should not be given a free pass to conclude whatever they like simply because a manuscript overwhelms the reader with large figures and a deep stack of EV supplements. My understanding is that Molecular Systems Biology publishes strong conclusions supported by strong modeling and strong experiments-two out of three is insufficient.

Reviewer #2:

The authors thoroughly and adequately addressed the concerns expressed in my written critique of the original manuscript.

Reviewer #3:

I liked the manuscript before and it has improved substantially following this revision. This manuscript is now suitable for publication in MSB. However, I ask the authors to perform the following minor revisions.

In my previous point 2, I wrote: "The authors use their mathematical model that was fit based on the data from the two cell lines (CFU-E and BaF3-EpoR) to predict the response of the third cell line (32D-EpoR). However, it is not clear how good their prediction is. E.g., in Fig 4A it would be nice to show besides 32D-EpoR predictions and actual measurements, also the responses of the CFU-E and BaF3-EpoR cells to the same stimulus. This would demonstrate how close the prediction for 32D-EpoR is to the experimental data, compared to the difference between different cell lines. Given how similar the protein expression profiles are between BaF3-EpoR and 32DEpoR (Table 1), one might expect that the 32D-EpoR response curve could be well approximated simply by the BaF3-EpoR curve. So, the authors at least need to show that their model, which integrates the data from both CFU-E and BaF3-EpoR cell lines, gives a better prediction than BaF3-EpoR data alone.

The authors replied: "We thank the reviewer for this suggestion. In the new section Expanded View information L, we now compare simulations for the three different cell types and find that, despite similarities between BaF3-EpoR and 32D-EpoR cells, the cell-type-specific protein abundance results in different dynamic responses of pAKT, ppERK and pS6 upon stimulation with 50 U/ml Epo (Fig. EV16). The χ^2 of comparing these simulations to the experimental data of 32D-EpoR cells indicates that the mathematical model adapted to the protein abundance in 32D-EpoR cells describes the measured dynamics in 32D-EpoR cells for pAKT and ppERK much better than the mathematical model based on the protein abundance in BaF3-EpoR cells (Tab. EV2). The better agreement for pS6 dynamics with the mathematical model based on the protein abundance in BaF3-EpoR cells might be due to the degree of freedom originating from the scaling that was estimated for these simulations to fit the data set. We have revised the results section of the manuscript accordingly (page 14):"

I agree that the curves for the various models are different as shown in EV16, and likely this will impact the model fit. This is a good start to what I was asking for. What I asked was for the authors to compare the fits of these models derived from data from a single cell line, with the model derived from both cell lines in terms of how they predict the measurements of 32D-EpoR. To be perfectly clear, there should be 3 numbers measuring the fit for a model based on CFU-E, a model based on BaF3-EpoR, and a model based on both CFU-E and BaF3-EpoR.

1st Revision - authors' response

25 October 2016

Report continues on next page.

Reviewer #1:

In the revision, the authors addressed many technical issues regarding stoichiometrically consistent Epo treatment of the cells, western blot displays, and some statistical analysis. However, other concerns are simply addressed in words: the 'cell cycle indicator' remains a heuristic that was likely chosen because it gave the authors the answer they wanted. This reviewer appreciates the effort made to add primary human CFU-E cells to address the concern raised about EpoR overexpression in the cell lines. If indeed human CFU-Es behaved like BaF3 cells, it would address the concerns related to ectopic expression. However, visual inspection of the data as shown in Fig. 8F are not convincing and the lack of statistical testing on these new analyses prevents the reader from being able to draw any conclusions about the additional cell type.

A more pervasive, and unresolved, concern relates to overstated claims based on the data as presented. To enumerate:

Conclusions not transparent based on the data shown:

1. "The concentration of ERK was higher in BaF3-EpoR... than mCFU-E cells... whereas the concentration of AKT was comparable" (Page 7): in Figure 1D, both ERK and AKT appear similarly more abundant in BaF3-EpoR cells compared to mCFU-E cells.

We thank the Reviewer for pointing out that the quantification of ERK and AKT protein abundance was difficult to follow. In fact, the qualitative immunoblots shown in Figure 1D used different numbers of cells to obtain clear results with the respective antibodies. These cell numbers entered the quantification and we have now added them directly to Figure 1D for clarity. Specifically, for the abundance of total ERK, the sum of the intensities of the bands for ERK1 and ERK2 were considered (BaF3-EpoR: $1.51 \times 10^7 + 1.39 \times 10^7 = 2.90 \times 10^7$ a.u.; mCFU-E: $4.29 \times 10^6 + 4.09 \times 10^6 = 8.38 \times 10^6$ a.u.), whereas the abundance of total AKT was based on the intensity of the AKT band (BaF3-EpoR: 1.41×10^7 a.u.; mCFU-E: 9.22×10^5 a.u.). Determination of molecules per cell and correction for the difference in cell number and cell volume, as indicated in the revised Figure 1D and Table 1, demonstrated that BaF3-EpoR cells and mCFU-E cells harbored similar concentrations of total AKT, whereas BaF3-EpoR cells contained higher concentrations of total ERK. We have now clarified these calculations in the main text of the manuscript (page 6, second paragraph):

"In a first step the protein abundance and dynamics of phosphorylation of EpoR, AKT and ERK in response to stimulation with a saturating Epo concentration (50 U/ml, see Appendix A, Appendix Figure S1, Appendix Figure S13) was qualitatively assessed for the same amount of BaF3-EpoR and mCFU-E cells. The total expression level and extent of phosphorylation of the EpoR were higher in BaF3-EpoR cells compared to mCFU-E cells (Fig. 1D, top panel). Although the apparent abundance of total AKT was higher in BaF3-EpoR cells, surprisingly the Epo-induced phosphorylation of AKT was higher and more sustained in mCFU-E cells compared to BaF3-EpoR cells (Fig. 1D, middle panel). As an indicator for AKT activation we focused on the analysis of Ser473 phosphorylation that is predictive for full kinase activation (Sarbasov *et al*, 2005;

Alessi *et al.*, 1996; Scheid *et al.*, 2002) since we observed that it correlates with Thr308 phosphorylation in an Epo dose-dependent manner (Appendix Figure S2). For ERK, higher total protein levels were observed for BaF3-EpoR cells. Accordingly, elevated levels of ERK phosphorylation were detected in BaF3-EpoR cells, but overall both cell types exhibited transient ERK phosphorylation dynamics (Fig. 1D, bottom panel)."

(Page 7, first paragraph):

"The accurate quantification of total molecules per cell and the correction for the difference in cellular volume showed that indeed the concentration of total ERK was higher in BaF3-EpoR (2964 ± 166 nM) than in mCFU-E cells (1140 ± 64 nM) whereas the concentration of total AKT was comparable (510 ± 62 nM in BaF3-EpoR cells and 407 ± 16 nM in mCFU-E cells)."

2. "the cell-cycle inhibitor was... higher in BaF3-EpoR cells... which is in agreement with the results of the propidium iodide staining experiment..." (Page 9): based on the data shown, the percent of cells in S/G2/M is higher in mCFU-E cells than in BaF3-EpoR cells; the authors should display the data as fold change over unstimulated if they want to indicate EpoR responsiveness. But note: the fold change in Epo-stimulated S/G2/M cells (~1.5-fold for mCFU-E vs. ~7-fold for BaF3-EpoR) correlates poorly with the fold-change in their cell-cycle indicator (~12-fold for mCFU-E vs. ~17-fold for BaF3-EpoR).

We followed the Reviewer's advice and now present in Appendix Figure S5 the results of the propidium iodide experiment as fold change compared to unstimulated cells. The fold change of cells in the S/G2/M phase of the cell cycle in response to 5 U/ml Epo stimulation is significantly ($p=0.002$) higher in BaF3-EpoR cells (7-fold) compared to mCFU-E cells (1.5-fold) (Appendix Figure S5). We now in addition state the fold change of the cell-cycle indicator and show that the value is significantly ($p=0.04$) higher in BaF3-EpoR cells (16.6-fold) compared to mCFU-E cells (12.2-fold; Fig. 1F, right panel). Hence the cell-cycle indicator clearly shows the same trend as the propidium iodide staining. However, the genes summarized in the cell-cycle indicator are upregulated early (measured at 3 hours after stimulation) whereas S phase entry happens much later (fraction of cells in S/G2/M phase of the cell cycle measured at 11 or 16 hours). As we reported earlier (Mueller *et al.* Mol. Syst. Biol. 11:795, 2015) a linear relationship between the expression of cell-cycle regulatory genes and S-phase entry cannot be expected. Nevertheless, the data faithfully represented the qualitative tendency that BaF3-EpoR cells show a stronger increase in cell cycle-related responses upon Epo stimulation than mCFU-E cells. To better address this issue, we modified Appendix Figure S5 and extensively rephrased the text (page 10, last paragraph, page 11, first paragraph):

"As evidenced in Figure 1F, after 3 hours of Epo addition we observed in BaF3-EpoR and mCFU-E comparatively small changes in the expression of the individual components (e.g. only *cyclinG2*) but a strong increase in the cell cycle indicator, 16-fold for BaF3-EpoR cells and 13-fold for CFU-E cells, respectively. These results underscore that at this early time point the coefficient reflects the complex

regulation of cell-cycle progression in response to Epo stimulation better than any of its components alone.

Notably, the cell-cycle indicator was significantly ($p=0.04$) higher in BaF3-EpoR cells compared to CFU-E cells (Fig. 1F, right panel). In line with this observation, we observed by propidium iodide staining after stimulation with 5 U/ml Epo for 16 hours (BaF3-EpoR) or 11 hours (mCFU-E) that the fold-change of cells in the S/G2/M phase of the cell cycle in response to Epo stimulation was also significantly ($p=0.002$) higher in BaF3-EpoR cells compared to mCFU-E cells (Appendix Figure S5). This result supports our notion of the cell-cycle indicator as an early measure for cell-cycle progression and shows that, whereas mCFU-E cells are already committed to cell-cycle progression, an increasing fraction of BaF3-EpoR cells enters S/G2/M phase in response to stimulation with increasing Epo doses."

Conclusions not supported by data:

1. "The dynamics of ppERK was more sustained in BaF3-EpoR cells than in mCFU-E cells" (Page 8): two data points in Figure 1D suggesting a slightly prolonged activation are ignored in the model fits of Figure 2D where the modeled time courses are close to overlapping.

We believe that the Reviewer refers to Figure 1E. While Figure 1E shows the dynamics of ppERK in both cell types in response to 50 U/ml Epo stimulation (as indicated in the figure legend), Figure 2D shows the dynamics of ERK phosphorylation in both cell types in response to stimulation with only 5 U/ml Epo. The data spline utilized in Figure 1E represents a more sustained phosphorylation of ERK in BaF3-EpoR cells in response to stimulation with 50 U/ml Epo. The model trajectories shown in Figure 2D correctly capture that in response to stimulation with 5 U/ml Epo the phosphorylation of ERK is also slightly more sustained in BaF3-EpoR cells compared to mCFU-E cells. The more sustained dynamics of ppERK in BaF3-EpoR cells for all doses of Epo is even better visible in Appendix Figure S13, upper middle panel. To emphasize this point and address the concern of the Reviewer, we added an additional reference to this Figure in the main text (page 13, first paragraph):

"We found that the distinct signaling dynamics and dose responses to Epo were captured by the mathematical model (Fig. 2B-E; Appendix I, Appendix Figure S13)."

2. "our mathematical analysis indicates that cell-type-specific processing of Epo stimuli is due to different abundances of the signaling proteins" (Page 12): total overstatement-up to this point in the manuscript, the results only suggest that differences in signal processing can be explained by different abundances in signaling proteins, based on a model whose free parameters were fit to capture those differences in signal processing.

We agree with the Reviewer that at this point of the manuscript "cell-type-specific processing of Epo stimuli" refers to "differences in signal processing" and specified this as requested. However, we respectfully disagree with the Reviewer's statement that the "free parameters" can be fitted to "capture those differences in signal processing." As the kinetic parameters of the model are global – that is: they are the

same for all three cell types studied – they cannot explain differences in signaling dynamics between cell types, no matter what their estimated values are. To clarify these issues we modified we modified the conclusion as follows (page 13, second paragraph):

"In summary, our mathematical analysis indicates that differences in signal processing can be explained by different abundance in signaling proteins in mCFU-E and BaF3-EpoR cells, based on a mathematical model with global kinetic parameters."

3. "RSK abundance... exhibited a high impact on the integrated pS6 response in BaF3-EpoR and 32D-EpoR cells but not in mCFU-E cells" (Page 15): the authors overexpress RSK in 32D-EpoR cells (Figure 4C), which is a perturbation in the wrong direction because 32D-EpoR cells already express more RSK than mCFU-E cells based on the quantitative data in Table 1. The experiment should have been a RSK knockdown. At a minimum, RSK would need to be overexpressed in mCFU-E cells and show no effect (or less of an effect) in the model and in the experiments.

As pointed out by the Reviewer, a higher abundance of RSK protein is observed in 32D-EpoR cells compared to mCFU-E cells. Surprisingly, our sensitivity analysis, which takes the protein abundance of the wild-type cells into account, predicted that despite the already high levels of RSK in wild-type 32D-EpoR cells, overexpression of RSK would further enhance the integrated pS6 response upon Epo stimulation. Therefore we focused on the experimental validation of this counterintuitive model prediction (Fig. 4C). To clarify the rationale and to better explain our line of thought, we adapted the text as follows (page 16, second paragraph):

"The sensitivity analysis indicated, for example, that the RSK abundance (Fig. 4B, bottom line) exhibits a high impact on the integrated pS6 response in BaF3-EpoR and 32D-EpoR cells but not in mCFU-E cells. These model-based insights are consistent with the high sensitivities obtained for the Ras/MEK/ERK pathway in the former two cell types, as RSK is downstream of ERK. Although wild-type 32D-EpoR cells already exhibited 2.8-fold higher levels of RSK than mCFU-E cells (Tab. 1), the sensitivity analysis taking this protein abundance into account suggested that RSK overexpression in 32D-EpoR cells would result in an increase of integrated pS6 in response to Epo stimulation. To test this counterintuitive model prediction, RSK was overexpressed in 32D-EpoR cells. Utilizing the amount of RSK experimentally detected in wild-type 32-EpoR cells as well as the amount of RSK present in the cells overexpressing RSK, the mathematical model predicted a major increase in pS6 in response to Epo stimulation whereas pAKT and ppERK remain rather unaffected."

4. "overexpression of constitutively active Ras qualitatively resulted in a stronger elevation of S6 phosphorylation in BaF3-EpoR than in mCFU-E cells" (Page 16): neither qualitatively nor quantitatively true-in Figure EV17A, the pS6 bands for Ras-G12V overexpression in mCFU-E and BaF3-EpoR are on two different membranes, so quantitative comparisons cannot be made directly; moreover, the quantitative data in Figure EV17B are superimposable (within measurement error)

after one takes into account that the y-axes are different scales for mCFU-E and BaF3-EpoR. The inhibitor data in Figure 5 are much stronger in this regard.

To address the concern of the Reviewer, we revised the representation of the quantification in Appendix Figure S17B. Now, we display for mCFU-E and BaF3-EpoR cells the fold change of pS6 at 30 minutes of stimulation with 5 U/ml Epo between wild-type cells and Ras-G12V overexpressing cells. Thereby, a direct comparison of the effect of the overexpression is possible even though these bands were not quantified from the same membranes. The obtained results show that overexpression of constitutively active Ras had a significantly stronger impact ($p=0.04$) on the increase of pS6 at 30 minutes after Epo stimulation relative to wild-type cells in BaF3-EpoR cells compared to mCFU-E cells. On the contrary, the overexpression of PTEN reduced pS6 in mCFU-E cells significantly ($p=0.01$) stronger than in BaF3-EpoR cells. These improvements are now also presented in the revised text (page 16, last paragraph, page 17, first paragraph):

"In further agreement with the sensitivity analysis, the observed experimental overexpression of constitutively active Ras resulted in comparison to wild-type cells in a significantly ($p=0.04$) stronger elevation of Epo-induced S6 phosphorylation in BaF3-EpoR than in mCFU-E cells whereas the overexpression of PTEN significantly ($p=0.01$) diminished Epo-stimulated S6 phosphorylation more strongly in mCFU-E than in BaF3-EpoR cells (Appendix Figure S17)."

5. "the more severe impact of PTEN overexpression compared to SHIP1 overexpression on the Epo-induced proliferation of mCFU-E and BaF3-EpoR cells was correctly predicted by our mathematical model" (Page 20): the Epo EC₅₀ for both modeling and experiments is comparable regardless of whether PTEN or SHIP1 is overexpressed (barring a statistical test that demonstrates otherwise). What changes is the baseline proliferation of mCFU-E cells (captured by the model for PTEN but not SHIP1), propagating to the maximally observed proliferation under conditions of saturating Epo.

To provide additional evidence for the agreement between model prediction and experimental validation, we performed a correlation analysis that is depicted in the new Appendix Figure S22 and shows a good coefficient of determination ($R^2=0.88$). Further we followed the suggestion of the Reviewer and calculated the EC₅₀ for the different conditions in both cell types and indicated these in the modified Fig. 7A. As pointed out by the Reviewer there was no major effect of the overexpression of SHIP1 or PTEN on the EC₅₀ of Epo-induced proliferation in BaF3-EpoR cells. However, in mCFU-E cells a small effect was visible and here the model predicted in line with the experimental evidence that overexpression of PTEN had a larger effect and gave rise to the highest EC₅₀. As observed by the Reviewer the model predicts for low Epo concentrations an elevated baseline proliferation for wild-type and SHIP1-overexpressing mCFU-E cells, which was not detected in the experiment. At low Epo concentrations residual phosphorylation of signaling components is present. However, in the experiments apparently the activation of signal transduction below a certain threshold is not sufficient to elicit proliferation and therefore baseline proliferation is absent. These improvements are now also represented in the revised text (page 21, first paragraph):

"Overall these phenotype predictions by the mathematical model (Fig. 7A, upper panels) were in good agreement ($R^2=0.88$) (Appendix Figure S22) with the experimental data (Fig. 7A, lower panels). In agreement with the experimental data the mathematical model predicted that there was no effect of overexpression of SHIP1 or PTEN on the EC_{50} of Epo-induced proliferation of BaF3-EpoR, whereas in mCFU-E cells a small effect was detectable and overexpression of PTEN consistently gave rise to the highest EC_{50} values. The EC_{50} values estimated for the experimentally measured proliferative responses of the wild-type mCFU-E (0.27 ± 0.05) and BaF3-EpoR (0.68 ± 0.46) cells were in line with our initial observations (see Fig. 1A). At very low Epo concentrations the mathematical model predicted an elevated baseline proliferation for wild-type and SHIP1 overexpressing mCFU-E cells that was not detected in the experiment. At these low Epo concentrations residual phosphorylation of signaling components is detectable and this information was utilized for calibration of the mathematical model. However, in the experiments the activation of signal transduction below a certain threshold apparently was not sufficient to elicit proliferation and therefore baseline proliferation is absent. Yet, in line with the experimental observations the mathematical model predicted that overexpression of PTEN decreased the proliferative response of mCFU-E cells and BAF3-EpoR cells the most."

6. "the number of molecules per cell determined with the different techniques showed a good correlation ($R^2 = 0.82$), validating our determinations by quantitative immunoblotting and confirming that the snapshot measurement by mass spectrometry yielded reliable results" (Page 21): Figure 8D is on a log-log plot and the slope is roughly one half, meaning that there is a square-root dependence (rather than a linear one) between the immunoblot data and the mass spec data.

To improve the representation of our results we now use the same limits for both axes and include as a guide to the eye a diagonal in the log/log plot. Moreover, we now specify the correlation to be rank-based. Accordingly, we adapted Figure 8D, its legend and the main text (page 22, last paragraph):

"As shown in Fig. 8D, for the key signaling components in the two cell types, the number of molecules per cell determined with the different techniques showed a good correlation (Spearman's rank-based coefficient of correlation $\rho=0.88$), validating our determinations by quantitative immunoblotting and confirming that the snapshot measurement by mass spectrometry yielded reliable results."

7. "Qualitatively the same effects on proliferation were observed for the hCFU-E cells from donor 1 and donor 2" (Page 22): the model predicts comparable sensitivity to the MEK and Akt inhibitors, with maximal inhibition with the combination. Donor 1 is highly sensitive to both inhibitors, with the Akt inhibitor dominating. Donor 2 is resistant to both inhibitors with the exception of the highest doses of Akt inhibitor. There is very little qualitatively similar between these two donors aside from the dominance of the Akt inhibitor, which disagrees with the model predictions.

To address the point raised by the Reviewer, we re-cultivated remaining samples of $CD34^+$ cells from donor 1 and donor 3, no material was left from donor 2, and

again observed that the cells from donor 3 proliferated better than the cells from donor 1. As summarized in the new Appendix Figure S34 taking together all experiments we find a significantly higher fold change in the numbers of cells from donor 3 as compared to donor 1 ($p=0.038$) and donor 2 ($p=0.009$) after three days of subcultivation. Based on these observation we chose cells from donor 3 for the validation of our model prediction presented in Figure 8F. To now better show that the effect of the inhibitor treatment is qualitatively comparable between the hCFU-E cells from the individual donors, we repeated the proliferation experiment in the presence of inhibitors and upon Epo stimulation of hCFU-E cells from donor 1 and donor 3. We now show in a 2D scatter plot in the new Appendix Figure S35 that the two experiments, each performed in three biological replicates, are in good agreement regarding the impact of the single and the combined inhibitor treatment on proliferation of the two donors (Pearson's coefficient of correlation $R^2=0.88$). Our mathematical model predicted that whereas Epo-induced proliferation of murine CFU-E cells is solely impaired by the AKT inhibitor, on the contrary Epo-induced proliferation of human CFU-E is not only sensitive towards the AKT inhibitor, but Epo-stimulated proliferation of hCFU-E is also affected by the MEK inhibitor and the combinatorial treatment. To better demonstrate that the mathematical model is able to correctly predict the distinct behavior of the two cell types, we now show a direct comparison of the experimental data obtained for mCFU-E and hCFU-E cells in the new Appendix Figure S36. This figure shows that as predicted by the mathematical model addition of the AKT inhibitor impaired Epo-induced proliferation of mCFU-E and hCFU-E and that there was no significant difference between these responses ($p=0.3$) of both cell types. However, as predicted by the mathematical model (Fig. 8F) the MEK inhibitor exhibited a significantly stronger effect on the Epo-induced proliferation of hCFU-E cells compared to mCFU-E cells ($p=0.03$). Also the combined treatment showed a significantly stronger effect on the proliferation of hCFU-E cells as compared to mCFU-E cells ($p=0.03$).

In sum, based on the different protein abundance of mCFU-E and hCFU-E cells our mathematical model is capable to correctly predict the distinct impact of the inhibitors and their combination on Epo-induced proliferation of the two cell types. To address the Reviewer's concern and better specify our argumentation, we rephrased our statement as follows (page 23, first paragraph, page 24, first paragraph, second paragraph):

"Without any further information (Appendix T) the mathematical model predicted that, distinct from mCFU-E (Fig. 6C, 7B), Epo-induced proliferation of hCFU-E cells was not only reduced by a treatment with the AKT inhibitor but also reduced by the treatment with the MEK inhibitor and the combination thereof (Fig. 8F). To validate this model prediction experimentally, we quantified the numbers of hCFU-E cells after 96 hours of stimulation with 5 U/ml Epo and individual or combined treatment with AKT VIII and U0126. Since hCFU-E cells from donor 3 responded most strongly to Epo, showing a significantly higher fold change in the numbers of cells compared to donor 1 ($p=0.038$) and donor 2 ($p=0.009$) after three days of subcultivation (Appendix Figure S34) and thus a advantageous signal-to-noise ratio, we focused the analyses of proliferation inhibition on hCFU-E cells from this donor in biological triplicates (Fig. 8F). We observed that the Epo-induced proliferation of hCFU-E is impaired upon treatment with both inhibitors individually and their combination. A Pearson's coefficient of correlation $R^2=0.88$

of the results obtained in two independent experiments confirmed the reproducibility of our experimental observations (Appendix Figure S35). When we directly compared the impact of AKT VIII and U0126 and the combination thereof on Epo-induced proliferation of hCFU-E and of mCFU-E, we observed that, in line with the model prediction, the impact of AKT VIII was comparable ($p=0.3$), whereas MEK ($p=0.03$) and the combinatorial inhibitor treatment ($p=0.03$) exhibited significantly larger effects on Epo-promoted proliferation of hCFU-E (Appendix Figure S36).

These findings showcase that protein abundance can be reliably measured from snapshot data of human material. Based on these data, our integrative mathematical model allows to evaluate the impact of inhibitors in silico and thus may serve to improve the treatment of proliferative disorders such as tumors driven by exacerbated growth-factor signaling."

Inconsistencies within the manuscript:

1. Transcriptome analysis was performed with 5 U/ml Epo (Page 9). Transcriptome analysis was performed with 1 U/ml Epo (Expanded View information A).

We thank the Reviewer for pointing out the inconsistency. We utilized 1 U/ml Epo for the transcriptomics analysis of BaF3-EpoR cells and 0.5 U/ml Epo for the mCFU-E cells that we now show in addition (new Appendix Figure S4B). For the confirmation by quantitative RT-PCR the cells were stimulated with 5 U/ml Epo, which is a saturating Epo dose for both cell types (Appendix Figure S1). To better document the establishment of the cell-cycle indicator and correctly state the different doses of Epo employed, we extensively rephrased the text (page 9, last paragraph, page 10, first paragraph):

"To provide a link between signal transduction and cell-cycle progression, transcriptome analysis was performed for up to 18.5 hours after stimulation of BaF3-EpoR cells with 1 U/ml Epo (Appendix Figure S4A) and for up to 24 hours after stimulation of mCFU-E cells with 0.5 U/ml Epo (Appendix Figure S4B). These analyses revealed that in both cell types several cell-cycle regulator genes were differentially expressed upon Epo stimulation (Appendix Figure S4). Prominent among these cell-cycle regulators affected by Epo were the activator Cyclin-D2, and the repressors Cyclin-G2 and p27, all of which jointly control the progression from G1 phase to S phase – the key event for cell-cycle entry (Fang *et al.*, 2007). On the other hand, other genes involved in the regulation of the cell cycle such as *cyclinE1* (*CCNE1*) and *cyclinE2* (*CCNE2*) showed only little regulation in either cell types. To confirm the transcriptomics studies, we examined the selected Epo-responsive cell-cycle regulating genes by quantitative RT-PCR analysis (Fig. 1F) and showed that after three hours of stimulation with 5 U/ml Epo, a saturating Epo dose for proliferation in BaF3-EpoR and mCFU-E cells (Appendix Figure S1), mRNA induction of *cyclinD2* (*CCND2*) and mRNA repression of *cyclinG2* (*CCNG2*) and *p27* (*CDKN1B*) exhibited comparable fold changes in BaF3-EpoR and mCFU-E cells. These results suggested that the quantification of the expression of *cyclinD2*, *cyclinG2* and *p27* might provide an early quantitative measure to compare Epo-induced cell-cycle progression in BaF3-EpoR and mCFU-E cells."

In summary, authors should not be given a free pass to conclude whatever they like simply because a manuscript overwhelms the reader with large figures and a deep

stack of EV supplements. My understanding is that Molecular Systems Biology publishes strong conclusions supported by strong modeling and strong experiments—two out of three is insufficient.

Reviewer #2:

The authors thoroughly and adequately addressed the concerns expressed in my written critique of the original manuscript.

Reviewer #3:

I liked the manuscript before and it has improved substantially following this revision. This manuscript is now suitable for publication in MSB. However, I ask the authors to perform the following minor revisions.

In my previous point 2, I wrote: "The authors use their mathematical model that was fit based on the data from the two cell lines (CFU-E and BaF3-EpoR) to predict the response of the third cell line (32D-EpoR). However, it is not clear how good their prediction is. E.g., in Fig 4A it would be nice to show besides 32D-EpoR predictions and actual measurements, also the responses of the CFU-E and BaF3-EpoR cells to the same stimulus. This would demonstrate how close the prediction for 32D-EpoR is to the experimental data, compared to the difference between different cell lines. Given how similar the protein expression profiles are between BaF3-EpoR and 32DEpoR (Table 1), one might expect that the 32D-EpoR response curve could be well approximated simply by the BaF3-EpoR curve. So, the authors at least need to show that their model, which integrates the data from both CFU-E and BaF3-EpoR cell lines, gives a better prediction than BaF3-EpoR data alone.

The authors replied: "We thank the reviewer for this suggestion. In the new section Expanded View information L, we now compare simulations for the three different cell types and find that, despite similarities between BaF3-EpoR and 32D-EpoR cells, the cell-type-specific protein abundance results in different dynamic responses of pAKT, ppERK and pS6 upon stimulation with 50 U/ml Epo (Fig. EV16). The χ^2 of comparing these simulations to the experimental data of 32D-EpoR cells indicates that the mathematical model adapted to the protein abundance in 32D-EpoR cells describes the measured dynamics in 32D-EpoR cells for pAKT and ppERK much better than the mathematical model based on the protein abundance in BaF3-EpoR cells (Tab. EV2). The better agreement for pS6 dynamics with the mathematical model based on the protein abundance in BaF3-EpoR cells might be due to the degree of freedom originating from the scaling that was estimated for these simulations to fit the data set. We have revised the results section of the manuscript accordingly (page 14):"

I agree that the curves for the various models are different as shown in EV16, and likely this will impact the model fit. This is a good start to what I was asking for. What I asked was for the authors to compare the fits of these models derived from data from a single cell line, with the model derived from both cell lines in terms of how they predict the measurements of 32D-EpoR. To be perfectly clear, there should

be 3 numbers measuring the fit for a model based on CFU-E, a model based on BaF3-EpoR, and a model based on both CFU-E and BaF3-EpoR.

We thank the Reviewer for clarifying the request. Accordingly, we now predicted with a mathematical model that was only calibrated to mCFU-E data and a mathematical model that was only calibrated to BaF3-EpoR data the activation dynamics of AKT, ERK and S6 in 32D-EpoR cells in response to 50 U/ml Epo stimulation. We compared the goodness of the fit of these model predictions with the prediction of the mathematical model calibrated to both, mCFU-E and BaF3-EpoR data. The chi-squared of the model based on only mCFU-E data was the worst with $\chi^2=48\,009$ while the chi-squared of the model based on only BaF3-EpoR data was better with $\chi^2=604$ but still not as good as the prediction of the model based on both, CFU-E and BaF3-EpoR data, with $\chi^2=284$. These results are now shown in Appendix Figure S16B and addressed in Appendix L as well as in the main text (page 15, first paragraph):

"Further we showed that the goodness of fit of a mathematical model calibrated with data from mCFU-E and BaF3-EpoR cells is superior to predict the dynamic activation of AKT, ERK and S6 in response to 50 U/ml Epo stimulation in 32D-EpoR cells when adapted to the cell-type-specific protein abundance compared to mathematical models calibrated with data obtained from mCFU-E cells or BaF3-EpoR cells alone (Appendix L)."

3rd Editorial Decision

11 November 2016

Thank you again for submitting your work to Molecular Systems Biology. We have now evaluated the revised manuscript and your point-by-point response to the referees' comments. While most issues have been satisfactorily addressed, we feel that there are some remaining issues that need to be addressed by figure and text modifications.

In particular:

- Related to point #1 of reviewer #1: Figure 1D shows total protein and phospho-protein levels from mCFU-E and BaF3 cells stimulated with different Epo concentrations (i.e. 2.5 U/ml for mCFU-E vs 5 U/ml for BaF3 cells in the upper two blots and 50 U/ml for both cell lines in the ppERK and ERK immunoblots shown in the bottom). We would like to ask you to explain clearly in the figure legend and the main text why different Epo concentrations have been used.

- Related to the statement on "more sustained dynamics of ppERK in BaF3 vs mCFU-E cells": While from the model predictions there seems to be a trend indicating a (small?) difference in the response, we think that robust conclusions cannot be drawn based on the related data shown in the manuscript (neither from Figure 2D nor from Figure S13). As such, we would recommend toning down the statements suggesting differences in ppERK dynamics predicted by the model.

- Related to point 7 of reviewer #1: Figure S36, showing the difference in the response of mouse and human primary cells to AKT, MEK and AKT+MEK inhibitors should be moved to the main Figure 8, since it contains information that is decisive for supporting a central conclusion of the study. Again, since the difference between the effect of the MEK inhibitor and the combination of AKT+MEK inhibitor seems rather subtle, we think that the related statements need to be toned down.

- A general note: A very large amount of data is shown in the Appendix and, as Reviewer #1 had previously noted, this data is not easily accessible. As such, along the lines of our suggestion to move Figure S36 to the main text, we would recommend moving further figures/panels from the

Appendix to the main text if they convey information essential to support the main conclusions of the study.

We have recently implemented a "model curation service" for papers that contain mathematical models. This is done together with Prof. Jacky Snoep and the FAIRDOM team. In brief, the aim is to enhance reproducibility and add value to papers containing mathematical models. We would ask you to deliver a complete set of corrected models that are appropriately documented (i.e. including a table of parameters) and reproduce the results shown in the figures.

2nd Revision - authors' response

21 November 2016

Thank you for the decision letter of November 11 on our manuscript "Protein abundance of AKT and ERK pathway components governs cell-type-specific regulation of proliferation". We were very pleased to learn that most issues have been satisfactorily addressed.

We are confident that the remaining issues are resolved by our adaptations of figures, main text and appendix.

In particular our changes in the revised manuscript include:

- new Figure 1G is former Appendix Figure S5
- revised Table 1
- new Figure 2F,G are panels of former Appendix Figure S13
- new Figure 4D is former Appendix Figure S17
- new Figure 6C is former Appendix Figure S21
- new Figure 8G is former Appendix Figure S36

In the following, we include a point-by-point response explaining how the particular points have been addressed.

We have now provided SBML files, lists of parameters and a step-by-step documentation to reproduce the signaling simulations for mCFU-E, BaF3-EpoR and 32D-EpoR cells without the use of any commercial software under the following link:

https://powerfolder.dkfz.de/dl/fiE4gMch5pGSRoyXQYZCtYA4/msb7258_adlung_sbml.zip

We hope that you will find the revised manuscript acceptable for publication in *Molecular Systems Biology*.

4th Editorial Decision

6 December 2016

Thank you once again for sending us the revised manuscript and for addressing all remaining issues. We are now satisfied with the modifications made and I am pleased to inform you that your paper has been accepted for publication.

Corresponding Author Name: Thomas Höfer & Ursula Klingmüller

Manuscript Number: MSB-16-7258